# Natural Policy Gradient for Average Reward Non-Stationary Reinforcement Learning

**Neharika Jali, Eshika Pathak**$^*$**, Pranay Sharma**$^\dagger$**, Guannan Qu, Gauri Joshi**
*Carnegie Mellon University*
*{njali, gqu, gaurij}@andrew.cmu.edu, epathak2@illinois.edu, pranaysh@iitb.ac.in*

**Reviewed on OpenReview:** *https://openreview.net/forum?id=hBJYNAYtoo*

## Abstract

We consider the problem of non-stationary reinforcement learning (RL) in the infinite-horizon average-reward setting. We model it by a Markov Decision Process with time-varying rewards and transition probabilities, with a variation budget of $\Delta_T$. Existing non-stationary RL algorithms focus on model-based and model-free value-based methods. Policy-based methods despite their flexibility in practice are not theoretically well understood in non-stationary RL. We propose and analyze the first model-free policy-based algorithm, Non-Stationary Natural Actor-Critic (NS-NAC), a policy gradient method with knowledge of the variation budget, a restart based exploration for change and a novel interpretation of learning rates as adapting factors. Next, eliminating the requirement of apriori knowledge of variation budget $\Delta_T$, we present a bandit-over-RL based parameter-free algorithm BORL-NS-NAC. We present a dynamic regret of $\tilde{\mathcal{O}}(|\mathcal{S}|^{1/2}|\mathcal{A}|^{1/2}\Delta_T^{1/6}T^{5/6})$ for both algorithms under standard assumptions, where $T$ is the time horizon, and $|\mathcal{S}|$, $|\mathcal{A}|$ are the sizes of the state and action spaces. The regret analysis leverages a novel adaptation of the Lyapunov function analysis of NAC to dynamic environments and characterizes the effects of simultaneous updates in policy, value function estimate and changes in the environment.

## 1 Introduction

While Reinforcement Learning has traditionally been studied in stationary environments with time-invariant rewards and state-transitions, this may not always be the case. Consider the examples of a carbon-aware datacenter job scheduler that tracks the dynamic electricity prices and local weather patterns [1] and recommendation systems with evolving user preferences [2]. Time-varying environments are also observed in healthcare [3], ride-sharing [4], and multi-agent systems [5]. Motivated by these applications, we consider the problem of non-stationary reinforcement learning, modeled by a Markov Decision Process with time-varying rewards and transition probabilities, in the infinite horizon average reward setting. While many works consider discounted rewards [3, 6, 7], the more challenging average-reward setting is vital in problems where the importance of rewards does not decay with time [8, 9, 10, 11]. The key challenges for an agent operating in a dynamic environment are learning an optimal behavior policy that varies with the environment, devising an efficient exploration strategy, and effectively incorporating the acquired information into its behavior.

Current algorithms designed for non-stationary MDPs in the average reward setting can be classified broadly into model-based and model-free value-based methods. Model-based solutions incorporate sliding windows, forgetting factors, and confidence interval management mechanisms into UCRL [12, 13, 14, 15]. Model-free value-based methods assimilate restarts and optimism into Q-Learning [16, 17] and LSVI [18, 19]. A significant gap in literature is absence of model-free policy-based techniques for time-varying environments.

---

$^*$Currently at University of Illinois Urbana-Champaign
$^\dagger$Currently at C-MInDS, Indian Insitute of Technology Bombay

The inherent flexibility of policy-based algorithms makes them suitable for continuous state-action spaces, efficient parameterization in high-dimensional state-action spaces, and enables effective exploration through stochastic policy learning [20].

**Our Contributions.** We tackle the problem of non-stationary reinforcement learning in the challenging infinite-horizon average reward setting in the following manner.

1. We propose and analyze Non-Stationary Natural Actor-Critic (NS-NAC), a policy gradient method with knowledge of the variation budget, a restart based exploration for change and a novel interpretation of learning rates as adapting factors. To the best of our knowledge, this is the first model-free policy-based method for time-varying environments.

2. We present a bandit-over-RL based parameter-free algorithm BORL-NS-NAC that eliminates the need for knowing the variation budget $\Delta_T$ apriori.

3. We present a $\tilde{\mathcal{O}}\left(|\mathcal{S}|^{1/2}|\mathcal{A}|^{1/2}\Delta_T^{1/6}T^{5/6}\right)$ dynamic regret bound for both algorithms under standard assumptions including uniform ergodicity where $T$ is the time horizon, $\Delta_T$ represents the variation budget of rewards and transition probabilities, $|\mathcal{S}||\mathcal{A}|$ is the size of the state-action space and $\tilde{\mathcal{O}}(\cdot)$ hides logarithmic factors. The regret analysis leverages a novel adaptation of the Lyapunov function analysis of NAC to dynamic environments and characterizes the effects of simultaneous updates in policy, value function estimate and changes in the environment.

## 2 Related Work

**Non-Stationary RL.** Solutions to the non-stationary RL problem can be categorized into passive and active methods. Active algorithms are designed to actively detect changes in the environment in contrast to passive ones which implicitly adapt to new environments without distinct recognition of the change. While we focus our attention on passive techniques with dynamic regret as the performance metric in this work, a comprehensive survey can be found in [21] and [22]. Model-based solutions in the infinite horizon average reward setting incorporate into UCRL a sliding window or a forgetting factor for piecewise stationary MDPs [14], variation aware restarts [13] and a bandit based tuning of sliding window and confidence intervals [12] for gradual or abrupt changes constrained by a variation budget.

In the episodic setting, model-free value based methods assimilate restarts and optimism into Q-Learning [16], LSVI [18, 19] and sliding window and optimistic confidence set based exploration into a value function approximated learning [17]. Further, in the episodic setting, [23] proposes strategically pausing learning as an effective solution to non-stationarity with forecasts of the future. [24] proposed an algorithm agnostic black-box approach that finds a non-stationary equivalent to optimal regret stationary MDP algorithms. [12, 16] also present parameter-free non-stationary RL algorithms that leverage the bandit-over-RL framework to adaptively tune algorithm without knowledge of the variation budget. Further, [16] presents an information theoretic lower bound on the dynamic regret and [25] captures the complexity of updating value functions with any change. We note the distinction between the scope of this work and the body of research on adversarial MDPs which often allow for only changes in rewards, study the static regret and work with full information feedback instead of bandit feedback. See Section A for a table of comparison of regret bounds.

**Non-Stationary Bandits.** A precursor to non-stationary RL, the multi-armed bandit problem with time-varying rewards was first proposed in [26]. Solutions include UCB with a sliding window or a discounting factor [26], UCB with adaptive blocks of exploration and exploitation [27], Restart-Exp3 [27], Thompson Sampling with a discounting factor [28] and bandit based sliding window tuning [29]. Further, while most existing works assume arbitrarily (constrained by variation budget) changing reward distributions, [30] achieves an improved regret when the reward distributions change smoothly. Recent work by [31] points out ambiguities in the definition of non-stationary bandits and how the dynamic regret performance metric causes over-exploration, and [32] proposes, predictive sampling, an algorithm that deprioritizes acquiring information that loses usefulness quickly.

**Policy Gradient Algorithms for Stationary RL.** [33] presents the first finite time convergence of the average reward two timescale Advantage Actor-Critic (A2C) to a stationary point. [34] further improved its rate by leveraging a single timescale algorithm. Convergence to global optima of A2C was analyzed in [35, 36] which use a two loop structure with the inner loop critic estimation. Further, [37] combines a mirror descent update with experience replay and characterized global convergence. Natural Policy Gradient (NPG) was analyzed in the discounted reward case in [38, 39] and with entropy regularization in [40]. NPG in the average reward setting with exact gradients was characterized in [41]. Most relevant to our work is the Natural Actor Critic (NAC) analyzed for the discounted reward case in [42] and average reward setting with (compatible) function approximation in [43].

## 3 Problem Setting

**Notation.** Standard typeface (e.g., $s$) denote scalars and bold typeface (e.g., $\mathbf{r}$, $\mathbf{A}$) denote vectors and matrices. $\|\cdot\|_\infty$ denotes the infinity norm and $\|\cdot\|_2$ denotes the 2-norm of vectors and matrices. Given two probability measures $P$ and $Q$, $d_{TV}(P, Q) = \frac{1}{2}\int_{\mathcal{X}}|P(dx) - Q(dx)|$ is the total variation distance between $P$ and $Q$, while $D_{\mathrm{KL}}(P\|Q) = \int_{\mathcal{X}} P(dx)\log\frac{P(dx)}{Q(dx)}$ is the KL-divergence. For two sequences $\{a_n\}$ and $\{b_n\}$, $a_n = \mathcal{O}(b_n)$ represents the existence of an absolute constant $C$ such that $a_n \leq Cb_n$. Further $\tilde{\mathcal{O}}$ is used to hide logarithmic factors. $|\mathcal{S}|$ denotes the cardinality of a set $\mathcal{S}$. Given a positive integer $T$, $[T]$ denotes the set $\{0, 1, 2, \cdots, T-1\}$.

### 3.1 Preliminaries: Stationary RL

**Markov Decision Process.** Reinforcement learning tasks can be modeled as discrete-time Markov Decision Processes (MDPs). An MDP is represented as $\mathcal{M} = (\mathcal{S}, \mathcal{A}, \mathbf{P}, \mathbf{r})$ where $\mathcal{S}$ and $\mathcal{A}$ are, respectively, finite sets of states and actions, $\mathbf{P} \in \mathbb{R}^{|\mathcal{S}||\mathcal{A}|\times|\mathcal{S}|}$ is the transition probability matrix, with $P(s'|s, a) \in [0, 1]$, for $s, s' \in \mathcal{S}, a \in \mathcal{A}$, and $\mathbf{r} \in \mathbb{R}^{|\mathcal{S}||\mathcal{A}|}$ is the reward vector with individual entries $\{r(s, a)\}$ bounded in magnitude by constant $U_R > 0$. An agent in state $s$ takes an action $a \sim \pi(\cdot|s)$ according to a policy $\boldsymbol{\pi}$, where for each state $s$, $\pi(\cdot|s)$ is a probability distribution over the action space. The agent then receives a reward $r(s, a)$ and transitions to the next state $s' \sim P(\cdot|s, a)$. We denote the *policy* by $\boldsymbol{\pi} \in \mathbb{R}^{|\mathcal{S}||\mathcal{A}|}$, which concatenates $\{\pi(\cdot|s)\}_s$. In a stationary MDP, the transition probabilities $\mathbf{P}$ and the rewards $\mathbf{r}$ are *time-invariant*.

**Average Reward and Value Functions.** In this work, we consider the average reward setting, essential in modeling problems where the importance of rewards does not decay with time [9, 11]. The time averaged reward of an ergodic Markov chain following policy $\boldsymbol{\pi}$ converges to

$$J^{\boldsymbol{\pi}} := \lim_{T\to\infty}\frac{\mathbb{E}\left[\sum_{t=0}^{T-1}r(s_t, a_t)\right]}{T} = \mathbb{E}_{s\sim d^{\boldsymbol{\pi},\mathbf{P}}(\cdot), a\sim\pi(\cdot|s)}\left[r(s, a)\right],$$

where $d^{\boldsymbol{\pi},\mathbf{P}}$ is the stationary distribution over states induced by policy $\boldsymbol{\pi}$ and transition probabilities $\mathbf{P}$. The *relative* state-value function defines overall reward accumulated when starting from state $s$ as

$$V^{\boldsymbol{\pi}}(s) := \mathbb{E}\left[\sum_{t=0}^{\infty}(r(s_t, a_t) - J^{\boldsymbol{\pi}})\,\Big|\,s_0 = s\right],$$

where the expectation is over the trajectory rolled out by $a_t \sim \pi(\cdot|s_t)$ and $s_{t+1} \sim P(\cdot|s_t, a_t)$. Similarly, the relative state-action value function defines the overall reward accumulated by policy $\boldsymbol{\pi}$ when starting from state $s$ and action $a$ as

$$Q^{\boldsymbol{\pi}}(s, a) := \mathbb{E}\left[\sum_{t=0}^{\infty}(r(s_t, a_t) - J^{\boldsymbol{\pi}})\,|\,s_0 = s, a_0 = a\right].$$

**Natural Actor-Critic.** The goal of an agent is to find a policy that maximizes the average reward

$$\boldsymbol{\pi}^\star = \max_{\boldsymbol{\pi}} J^{\boldsymbol{\pi}} = \max_{\boldsymbol{\pi}} \mathbb{E}_{s\sim d^{\boldsymbol{\pi},\mathbf{P}}(\cdot), a\sim\pi(\cdot|s)}\left[r(s, a)\right].$$

Here, we consider the actor-critic class of policy-based algorithms. While actor-only methods are at a disadvantage due to inefficient use of samples and high variance and critic-only methods are at a risk of the divergence from the optimal policy, actor-critic methods provide the best of both worlds [33]. An actor-critic algorithm learns the policy and the value function simultaneously by gradient methods. Further, the natural actor-critic leverages the second-order method of natural gradient to establish guarantees of global optimality [44, 42]. The *actor* updates the policy $\boldsymbol{\pi_\theta}$ parameterized by $\boldsymbol{\theta}$ by performing a natural gradient ascent [45] step

$$\boldsymbol{\theta} \leftarrow \boldsymbol{\theta} + \beta F_{\boldsymbol{\pi_\theta}}^{-1} \nabla J^{\boldsymbol{\pi_\theta}}, \quad \text{where} \quad F_{\boldsymbol{\pi_\theta}} := \mathbb{E}_{s \sim d^{\boldsymbol{\pi_\theta}}, \mathbf{P}(\cdot), a \sim \boldsymbol{\pi_\theta}(\cdot|s)} \left[ \nabla \log \boldsymbol{\pi_\theta}(a|s) \left( \nabla \log \boldsymbol{\pi_\theta}(a|s) \right)^\top \right].$$

$F_{\boldsymbol{\pi_\theta}}$ is called the Fisher Information matrix. The gradient of the average reward is given by the Policy Gradient Theorem [20, Section 13.2] as

$$\nabla J^{\boldsymbol{\pi_\theta}} = \mathbb{E}_{s \sim d^{\boldsymbol{\pi_\theta}}, \mathbf{P}(\cdot), a \sim \boldsymbol{\pi_\theta}(\cdot|s)} \left[ Q^{\boldsymbol{\pi_\theta}}(s, a) \nabla \log \boldsymbol{\pi_\theta}(a|s) \right].$$

The *critic* enables an approximate policy gradient computation by estimating the Q-Value function $Q^{\boldsymbol{\pi}}(s, a)$ using TD-learning as

$$Q(s, a) \leftarrow Q(s, a) + \alpha \left[ r(s, a) - \eta + Q(s', a') - Q(s, a) \right],$$

where $s' \sim P(\cdot|s, a)$, $a' \sim \pi(\cdot|s')$, and $\eta$ is an estimate of the average reward $J^{\boldsymbol{\pi}}$.

## 3.2 Non-Stationary RL

In this work, we study reinforcement learning with *time-varying environments*. The MDP is modeled by a sequence of environments $\mathcal{M} = \{\mathcal{M}_t = (\mathcal{S}, \mathcal{A}, \mathbf{P}_t, \mathbf{r}_t)\}_{t=0}^{T-1}$, with time-varying rewards $\{\mathbf{r}_t\}$ and transition probabilities $\{\mathbf{P}_t\}$. At each time $t$, the agent in state $s_t$ takes action $a_t$, receives a reward $r_t(s_t, a_t)$, and transitions to the next state $s_{t+1} \sim P_t(\cdot|s_t, a_t)$. The cumulative change in the reward and transition probabilities is quantified in terms of *variation budgets* $\Delta_{R,T}$ and $\Delta_{P,T}$ as

$$\Delta_{R,T} = \sum_{t=0}^{T-1} \|\mathbf{r}_{t+1} - \mathbf{r}_t\|_\infty, \quad \Delta_{P,T} = \sum_{t=0}^{T-1} \|\mathbf{P}_{t+1} - \mathbf{P}_t\|_\infty, \quad \Delta_T = \Delta_{R,T} + \Delta_{P,T}. \tag{1}$$

Note that while the overall budgets $\Delta_{R,T}, \Delta_{P,T}$ may be used as inputs by the agent, the variations at a given time $t$, $\|\mathbf{r}_{t+1} - \mathbf{r}_t\|_\infty$ and $\|\mathbf{P}_{t+1} - \mathbf{P}_t\|_\infty$, are unknown.

We denote the long-term average reward obtained by following policy $\boldsymbol{\pi}_t$ in the environment $\mathcal{M}_t$ by

$$J_t^{\boldsymbol{\pi}_t} = \mathbb{E}_{s \sim d^{\boldsymbol{\pi}_t}, \mathbf{P}_t(\cdot), a \sim \pi(\cdot|s)} \left[ r_t(s, a) \right].$$

Further, the state and state-action value functions at time $t$ are solutions to the Bellman equations

$$V_t^{\boldsymbol{\pi}_t}(s) = \sum_{a \in \mathcal{A}} \pi(a|s) Q_t^{\boldsymbol{\pi}_t}(s, a) \quad \text{and} \quad Q_t^{\boldsymbol{\pi}_t}(s, a) = r_t(s, a) - J_t^{\boldsymbol{\pi}_t} + \sum_{s' \in \mathcal{S}} P_t(s'|s, a) V_t^{\boldsymbol{\pi}_t}(s').$$

The set of solutions to the Bellman equations above is $\mathbf{Q}_t^{\boldsymbol{\pi}_t} = \{\mathbf{Q}_{t,E}^{\boldsymbol{\pi}_t} + c\mathbf{1} | \mathbf{Q}_{t,E}^{\boldsymbol{\pi}_t} \in E, c \in \mathbb{R}\}$ where $E$ is the subspace orthogonal to the all ones vector and $\mathbf{Q}_{t,E}^{\boldsymbol{\pi}_t}$ is the unique solution in $E$ [46].

**Dynamic Regret.** The goal of the agent is to maximize the time-averaged reward $\sum_{t=0}^{T-1} r_t(s_t, a_t)/T$. We measure performance using an equivalent metric called the *dynamic regret* defined as

$$\text{Dyn-Reg}(\mathcal{M}, T) := \mathbb{E} \left[ \sum_{t=0}^{T-1} J_t^{\boldsymbol{\pi}_t^\star} - r_t(s_t, a_t) \right], \tag{2}$$

where $\boldsymbol{\pi}_t^\star = \arg\max_{\boldsymbol{\pi}} J_t^{\boldsymbol{\pi}}$ is the optimal policy in the environment $\mathcal{M}_t = (\mathcal{S}, \mathcal{A}, \mathbf{P}_t, \mathbf{r}_t)$ at time $t$. The optimal average reward $J_t^{\boldsymbol{\pi}_t^\star}$ associated with $\boldsymbol{\pi}_t^\star$ can be computed by solving the linear program (27) described in Section D.4. The model of change and notion of dynamic regret considered here are standard in the

non-stationary RL literature [12, 47, 18, 16, 17]. Note that it is more challenging to analyze and practical than static regret, which compares the cumulative reward collected by an agent against that of a single stationary optimal policy [48, 19].

**Challenges due to Non-Stationarity.** When running policy-gradient methods in stationary RL, the policy evolves to efficiently learn a fixed environment $(\mathbf{P}, \mathbf{r})$. However, in non-stationary case, the environment $(\mathbf{P}_t, \mathbf{r}_t)$ also changes over time. Therefore, the agent chases a moving target, namely, the *time-varying optimal policy* $\boldsymbol{\pi}_t^\star$, resulting in the following unique challenges.

- *Explore-for-Change vs Exploit:* The agent needs to explore more aggressively than in the stationary setting to adapt to the changing dynamics. As an example, a sub-optimal action at the current timestep may become optimal at a later timestep, necessitating re-exploration. This is in sharp contrast to stationary RL, where sub-optimal actions are picked less often as time progresses.

- *Forgetting Old Environments:* The policy and value function estimates must evolve quickly lest they might become irrelevant when the environment changes significantly. However, observations are noisy and the agent needs to collect multiple samples to obtain confident estimates. Hence, an agent has to carefully balance the rate of forgetting the old environment versus learning a new one.

## 4 Algorithm: NS-NAC

In this section, we present Non-Stationary Natural Actor-Critic (NS-NAC), a two-timescale natural policy gradient method with a restart based exploration for change and step-sizes designed to carefully balance the rate of forgetting the old environment and adapting to a new one. Note that we use the variation budget $\Delta_T$ as an input to NS-NAC here. Since the variation budget may not always be known, we present a parameter-free algorithm BORL-NS-NAC in Section 6 that does not require this.

---

**Algorithm 1** Non-Stationary Natural Actor-Critic (NS-NAC)

---

1: **Input** time horizon $T$; variation budgets $\Delta_{R,T}, \Delta_{P,T}$; projection radius $R_Q$
2: **Set** step-sizes of actor $\beta$, critic $\alpha$, average reward $\gamma$ as function of $\Delta_{R,T}, \Delta_{P,T}$; number of restarts $N$ of length $H = \lfloor \frac{T}{N} \rfloor$ as function of $\Delta_{R,T}, \Delta_{P,T}$; time $t = 0$; $s_0 \sim$ some starting distribution
3: **for** $n = 0, 1, 2, \ldots, N-1$ **do**
4:     Set policy $\pi_t(a|s) = \frac{1}{|\mathcal{A}|}$, value function $Q_t(s,a) = 0 \ \forall s, a$, average reward estimate $\eta_t = 0$
5:     Take action $a_t \sim \pi_t(\cdot|s_t)$
6:     **for** $h = 0, 1, 2, \ldots, H-1$ **do**
7:         Observe reward $r_t(s_t, a_t)$, next state $s_{t+1} \sim P_t(\cdot|s_t, a_t)$, take action $a_{t+1} \sim \pi_t(\cdot|s_{t+1})$
8:         $\eta_{t+1} \leftarrow \eta_t + \gamma \left( r_t(s_t, a_t) - \eta_t \right)$
9:         $Q_{t+1}(s_t, a_t) \leftarrow \Pi_{R_Q} \left[ Q_t(s_t, a_t) + \alpha \left( r_t(s_t, a_t) - \eta_t + Q_t(s_{t+1}, a_{t+1}) - Q_t(s_t, a_t) \right) \right]$
10:        $\pi_{t+1}(a|s) \leftarrow \frac{\pi_t(a|s) \exp(\beta Q_t(s,a))}{\sum_{a' \in \mathcal{A}} \pi_t(a'|s) \exp(\beta Q_t(s,a'))}, \forall s, a$
11:        $t \leftarrow t + 1$
12:     **end for**
13: **end for**

---

The NS-NAC algorithm seeks to maximize the total reward received over the time horizon $T$, given the variation budgets $\Delta_{R,T}$ and $\Delta_{P,T}$. At timestep $t$, $\boldsymbol{\pi}_t$ denotes the softmax parameterized tabular policy with parameters $\boldsymbol{\theta}_t \in \mathbb{R}^{|\mathcal{S}||\mathcal{A}|}$ where $\pi_t(a|s) \geq 0 \ \forall a \in \mathcal{A}$ and $\sum_a \pi_t(a|s) = 1 \ \forall s \in \mathcal{S}$ and represented as

$$\boldsymbol{\pi}_t = \frac{\exp[\boldsymbol{\theta}_t]_{s,a}}{\sum_{a' \in \mathcal{A}} \exp[\boldsymbol{\theta}_t]_{s,a'}}.$$

$\boldsymbol{\pi}_t^\star = \arg\max_{\boldsymbol{\pi}} J_t^{\boldsymbol{\pi}}$ is the optimal policy in the environment $\mathcal{M}_t$. The estimate of the tabular state-action value function $\mathbf{Q}_t^{\boldsymbol{\pi}_t}$ is denoted by $\mathbf{Q}_t \in \mathbb{R}^{|\mathcal{S}||\mathcal{A}|}$. $\eta_t$ denotes the estimate of the average reward $J_t^{\boldsymbol{\pi}_t}$.

NS-NAC divides total horizon $T$ into $N$ segments of length $H = \lfloor T/N \rfloor$ each. At the beginning of each segment, the algorithm restarts the NAC sub-routine (line 4), thereby ensuring that the algorithm sufficiently

*explores for change.* Next, at each time-step $t = nH + h \ \forall n \in [N], h \in [H]$, the *actor* (slower timescale) takes a natural gradient ascent step [38] towards optimal policy in environment $\mathcal{M}_t$ as

$$\boldsymbol{\theta}_{t+1} \leftarrow \boldsymbol{\theta}_t + \beta F_{\boldsymbol{\pi}_t}^{-1} \mathbb{E}_{s,a}\left[Q_t^{\boldsymbol{\pi}_t}(s,a)\nabla \log \pi_t(a|s)\right] \quad \Longrightarrow \quad \pi_{t+1}(a|s) \leftarrow \frac{\pi_t(a|s)\exp(\beta Q_t^{\boldsymbol{\pi}_t}(s,a))}{\sum_{a' \in \mathcal{A}} \pi_t(a'|s)\exp(\beta Q_t^{\boldsymbol{\pi}_t}(s,a'))} \forall s,a.$$

In the absence of knowledge of the exact natural gradient, the actor uses an estimate of the value function to update the policy with tabular softmax parameterization as in line 10.

The *critic* (faster timescale) estimates the tabular state-action value function of the current policy $\boldsymbol{\pi}_t$ as $\mathbf{Q}_t$ using TD-Learning with step-size $\alpha$ (line 9). The projection step in line 9 is defined as $\Pi_{R_Q}\left[\mathbf{x}\right] := \arg\min_{\|\mathbf{y}\|_2 \leq R_Q} \|\mathbf{x} - \mathbf{y}\|_2$ (see Lemma 1 and following discussion on choice of $R_Q$). Further, the average reward estimate $\eta_t$ is updated with step-size $\gamma$ (line 8). Using a two timescale technique with $\alpha \gg \beta$, NS-NAC thus enables the actor to chase the moving target $\boldsymbol{\pi}_t^\star$ facilitated by the critic updates of the value function estimates which adapt to the changed data distribution. In the stationary RL case, this change in data distribution is induced solely by the evolving actor policy, while in non-stationary RL, the time-varying environment $(\mathbf{P}_t, \mathbf{r}_t)$ further exacerbates it. Further, as Theorem 1 suggests, a careful selection of the step-sizes as a function of the variation budgets enables NS-NAC to balance the rate of *forgetting the old environment* versus learning a new one.

**Function Approximation.** While we consider the tabular formulation here for the ease of presentation, NS-NAC can also be extended to the function approximation setting. Further details in Section E.

## 5 Regret Analysis: NS-NAC

In this section, we set up notation and assumptions and establish an upper bound on the dynamic regret of NS-NAC. We further present a sketch of the proof in Section 7.

### 5.1 Assumptions

**Notation.** We denote an observation $O_t = (s_t, a_t, s_{t+1}, a_{t+1})$. If $d^{\boldsymbol{\pi}_t, \mathbf{P}_t}(\cdot)$ is the stationary distribution induced over the states, we define the matrices $\mathbf{A}(O_t), \bar{\mathbf{A}}^{\boldsymbol{\pi}_t, \mathbf{P}_t} \in \mathbb{R}^{|\mathcal{S}||\mathcal{A}| \times |\mathcal{S}||\mathcal{A}|}$ as

$$\mathbf{A}(O_t)_{i,j} = \begin{cases} -1, & \text{if } (s_t, a_t) \neq (s_{t+1}, a_{t+1}), i = j = (s_t, a_t) \\ 1, & \text{if } (s_t, a_t) \neq (s_{t+1}, a_{t+1}), i = (s_t, a_t), j = (s_{t+1}, a_{t+1}) \\ 0, & \text{else} \end{cases}$$

$$\bar{\mathbf{A}}^{\boldsymbol{\pi}_t, \mathbf{P}_t} = \mathbb{E}_{s \sim d^{\boldsymbol{\pi}_t, \mathbf{P}_t}(\cdot), a \sim \boldsymbol{\pi}_t(\cdot|s), s' \sim \mathbf{P}_t(\cdot|s,a), a' \sim \boldsymbol{\pi}_t(\cdot|s')}\left[\mathbf{A}(s,a,s',a')\right].$$

If $D^{\boldsymbol{\pi}_t, \mathbf{P}_t} = diag\left(d^{\boldsymbol{\pi}_t, \mathbf{P}_t}(s)\pi_t(a|s)\right)$ and $\mathbf{1}$ is the all ones vector, then the TD limiting point [42] satisfies

$$\mathbf{D}^{\boldsymbol{\pi_t}, \mathbf{P_t}}\left(\mathbf{r}_t - J_t^{\boldsymbol{\pi}_t}\mathbf{1}\right) + \bar{\mathbf{A}}^{\boldsymbol{\pi}_t, \mathbf{P}_t}\mathbf{Q}_t^{\boldsymbol{\pi}_t} = 0. \tag{3}$$

**Assumption 1** (Uniform Ergodicity, [33, 43]). A Markov chain generated by implementing policy $\boldsymbol{\pi}$ and transition probabilities $\mathbf{P}$ is called uniformly ergodic, if there exists $m > 0$ and $\rho \in (0, 1)$ such that

$$d_{TV}\left(P(s_\tau \in \cdot|s_0 = s), d^{\boldsymbol{\pi}, \mathbf{P}}\right) \leq m\rho^\tau \ \forall \tau \geq 0, s \in \mathcal{S},$$

where $d^{\boldsymbol{\pi}, \mathbf{P}}$ is the stationary distribution induced over the states. We assume Markov chains induced by all potential policies $\boldsymbol{\pi}_t$ in all environments $\mathbf{P}_t$, $t \in [T]$, are uniformly ergodic with parameters $m, \rho$. Further, if $\boldsymbol{\pi}_t^\star$ denotes the optimal policy for the environment $\mathcal{M}_t = (\mathcal{S}, \mathcal{A}, \mathbf{P}_t, \mathbf{r}_t)$, there exists $C > 0$ such that

$$C = \inf_{s,t,t',\boldsymbol{\pi}} \frac{d^{\boldsymbol{\pi}, \mathbf{P}_{t'}}(s)}{d^{\boldsymbol{\pi}_t^\star, \mathbf{P}_t}(s)} > 0.$$

**Lemma 1** (Lemma 2, [46]). *Under Assumption 1, for all potential policies $\boldsymbol{\pi}_t$ in all environments $\mathbf{P}_t$, $t \in [T]$, $\bar{\mathbf{A}}^{\boldsymbol{\pi}_t, \mathbf{P}_t}$ is negative semi-definite. Define its maximum non-zero eigenvalue as $-\lambda$.*

Assumption 1 is standard in literature [41, 33, 49]. Also note, we set the projection radius $R_Q = 2U_R\lambda^{-1}$ in line 9 because $\|\left(\bar{\mathbf{A}}^{\boldsymbol{\pi}_t, \mathbf{P}_t}\right)^\dagger\|_2 \leq \lambda^{-1}$ where $\dagger$ represents the pseudo-inverse.

### 5.2 Bounds on Regret

**Theorem 1.** *If Assumption 1 is satisfied, the step-sizes are chosen as $0 < \alpha, \beta, \gamma < 1/2$ and number of restarts as $0 < N < T$ in Algorithm 1, then we have*

$$Dyn\text{-}Reg(\mathcal{M}, T) = \mathbb{E}\left[\sum_{t=0}^{T-1} J_t^{\pi_t^\star} - r_t(s_t, a_t)\right] \leq \underbrace{\tilde{\mathcal{O}}\left(\frac{N}{\beta}\right) + \tilde{\mathcal{O}}\left(\sqrt{\frac{NT}{\alpha}}\right)}_{\text{effect of initialization}} + \underbrace{\tilde{\mathcal{O}}\left(\frac{\beta T}{\alpha}\right) + \tilde{\mathcal{O}}\left(T\sqrt{\beta}\right)}_{\substack{\text{cumulative change} \\ \text{in policy over horizon } T}}$$

$$+ \underbrace{\tilde{\mathcal{O}}\left(\frac{\beta T}{\gamma}\right) + \tilde{\mathcal{O}}\left(T\sqrt{\gamma}\right) + \tilde{\mathcal{O}}\left(\sqrt{\frac{NT}{\gamma}}\right)}_{\text{error in average reward estimate at critic}} + \underbrace{\tilde{\mathcal{O}}\left(T\sqrt{\alpha}\right)}_{\substack{\text{cumulative change} \\ \text{in critic estimates}}} + \underbrace{\tilde{\mathcal{O}}\left(\frac{\Delta_T T}{N}\right) + \tilde{\mathcal{O}}\left(\frac{\Delta_T^{1/3}T^{2/3}}{\sqrt{\alpha}} + \frac{\Delta_T^{1/3}T^{2/3}}{\sqrt{\gamma}}\right)}_{\text{error due to non-stationarity}}, \tag{4}$$

*where $\Delta_T = \Delta_{R,T} + \Delta_{P,T}$, $\tilde{\mathcal{O}}(\cdot)$ hides the constants and logarithmic dependence on the time horizon $T$. Choosing optimal $\alpha^\star = \gamma^\star = \left(\frac{\Delta_T}{T}\right)^{1/3}$, $\beta^\star = \left(\frac{\Delta_T}{T}\right)^{1/2}$ and $N^\star = \Delta_T^{5/6}T^{1/6}$, the resulting regret (with explicit dependence on the size of the state-action space $|\mathcal{S}|, |\mathcal{A}|$) is*

$$Dyn\text{-}Reg(\mathcal{M}, T) \leq \tilde{\mathcal{O}}\left(|\mathcal{S}|^{1/2}|\mathcal{A}|^{1/2}\Delta_T^{1/6}T^{5/6}\right). \tag{5}$$

We provide a sketch of the proof in Section 7 and the full proof in Section D. Further, extension of NS-NAC to the *function approximation* setting can be found in Section E.

**Effect of Non-Stationarity.** The variation budget $\Delta_T$ (1) represents the extent of non-stationarity of the environment. In Theorem 1, as the variation budget increases, so do the optimal choice of step-sizes and number of restarts, and the regret incurred (5). This observation is consistent with the intuition that in a rapidly changing environment, the algorithm must adapt quickly and explore more (hence, larger step-sizes and more restarts). However, as a result, the algorithm cannot exploit its current policy and value-function estimates, which soon become outdated (hence, higher regret). Also, in environments with larger state/action spaces, the agent requires proportionately more samples to detect changes and learn a good policy.

*Remark.* Define the mixing time [33] as $\tau_{mix} := \min\{i \geq 0 | m\rho^{i-1} \leq \min\{\alpha, \beta\}\}$ with $m, \rho$ as in Assumption 1. The explicit dependence of the upper bound, tracked in Section D, on $\tau_{mix}$ and $\lambda$ can be stated as

$$\text{Dyn-Reg}(\mathcal{M}, T) \leq \tilde{\mathcal{O}}\left(|\mathcal{S}|^{1/2}|\mathcal{A}|^{1/2}\frac{\tau_{mix}}{\lambda}\Delta_T^{1/6}T^{5/6}\right).$$

We choose not to include this explicit dependence in Theorem 1 to maintain focus on the dependence of regret on the variation budget and prevent confusion by direct comparison to the diameter $D$ of an MDP.

**Theorem 2** ([16], Proposition 1)**.** *For any learning algorithm, there exists a non-stationary MDP such that the dynamic regret of the algorithm is at least $\Omega(|\mathcal{S}|^{1/3}|\mathcal{A}|^{1/3}D^{2/3}\Delta_T^{1/3}T^{2/3})$ where $D$ is the diameter.*

**Gap between Bounds.** To the best of our knowledge, this is the first bound on dynamic regret for model-free policy-based algorithm in the infinite horizon average reward setting. Observe that the infinite horizon setting (only one sample per environment available) is harder than the episodic setting (environment remains stationary during the episode) and necessitates a single loop algorithm with the policy being updated at every timestep. We conjecture that the gap between the bounds results from a slack in the analysis of the underlying Natural Actor-Critic (NAC) algorithm. The best-known regret bounds for NAC for an infinite horizon *stationary* MDP in the (compatible) function approximation setting with a two timescale algorithm is $\tilde{\mathcal{O}}(T^{3/4})$ [42]. Analysis of the actor involves the norm of the critic estimation error $\|\mathbf{Q}_t - \mathbf{Q}_t^{\pi_t}\|$ (Proposition 1) whereas guarantees for critic establish a bound on norm-squared of the error $\|\mathbf{Q}_t - \mathbf{Q}_t^{\pi_t}\|^2$ (Proposition 2). This mismatch, which underlies the sub-optimality of the current best stationary infinite horizon NAC analysis, is exacerbated in non-stationary environments resulting in the gap between the upper and the lower bounds. [1] Note that mismatch of the value function estimation error between actor and critic doesn't occur in the analysis of model-based methods which use Hoeffding style high probability bounds.

---

[1]The term characterizing the difference in value functions at consecutive timesteps $\|\mathbf{Q}_{t+1}^{\pi_{t+1}} - \mathbf{Q}_t^{\pi_t}\|$ is the cause for the bottleneck $\tilde{\mathcal{O}}\left(\Delta_T^{1/3}T^{2/3}\left(\frac{1}{\sqrt{\alpha}} + \frac{1}{\sqrt{\gamma}}\right)\right)$ term (see $I_4, I_5, I_6$ in Proposition 2).

## 6  Unknown Variation Budgets: BORL-NS-NAC

NS-NAC required knowledge of the variation budget to achieve sub-linear regret by choosing the optimal step-sizes and number of restarts as described in Theorem 1 which is not always possible in practice. To overcome this limitation, in this section, we propose a parameter-free algorithm that adaptively learns the variation budget when it is unknown a priori. Inspired by the bandit-over-RL (BORL) framework in [16, 12], we present BORL-NS-NAC that does not require prior knowledge of the variation budget $\Delta_T$ yet achieves sub-linear dynamic regret. Further, utilizing the EXP3.P analysis from [50], we present a regret upper bound.

---

**Algorithm 2** Bandit-over-RL Non-Stationary Natural Actor-Critic (BORL-NS-NAC)

---

1: **Input** time horizon $T$, projection radius $R_Q$
2: **Initialize** $u_{0,j} = 0, p_{0,j} = \frac{1}{\lceil \ln T \rceil} \ \forall j \in [\lceil \ln T \rceil]$, epoch length $W$
3: **Set** $\xi = 0.95\sqrt{\frac{\lceil \ln T \rceil}{\lceil \ln T \rceil \lceil T/W \rceil}}, \sigma = \sqrt{\frac{\lceil \ln T \rceil}{\lceil \ln T \rceil \lceil T/W \rceil}}, \zeta = 1.05\sqrt{\frac{\lceil \ln T \rceil \lceil \ln T \rceil}{\lceil T/W \rceil}}$
4: **for** $i = 0, 1, \ldots, \lfloor T/W \rfloor$ **do**
5:     Sample $j_i \sim p_i$ where $p_{i,j} = (1 - \zeta)\frac{\exp(\xi u_{i,j})}{\sum_j \exp(\xi u_{i,j})} + \frac{\zeta}{\lceil \ln T \rceil}$
6:     Set step-sizes $\beta_i = \left(\frac{T^{j_i/\lfloor \ln T \rfloor}}{T}\right)^{1/2}, \alpha_i = \gamma_i = \left(\frac{T^{j_i/\lfloor \ln T \rfloor}}{T}\right)^{1/3}$ and restarts $N_i = W\left(\frac{T^{j_i/\lfloor \ln T \rfloor}}{T}\right)^{5/6}$
7:     Run NS-NAC (Algorithm 1) for $W$ time-steps and observe cumulative reward $R_{i,j_i} = \sum_{t=iW}^{(i+1)W-1} r_t(s_t, a_t)$
8:     Update posterior as $u_{i+1,j} = u_{i,j} + \frac{\sigma + \mathbb{I}_{j=j_i} \cdot R_{i,j_i}/W}{p_{i,j}}$
9: **end for**

---

BORL-NS-NAC works by leveraging the adversarial bandit framework to tune the variation budget dependent parameters, i.e. step-sizes and number of restarts, in NS-NAC and hedges against changes in rewards and transition probabilities. Algorithm 2 runs the EXP3.P algorithm [50] over $\lceil T/W \rceil$ epochs with NS-NAC as a sub-routine in each epoch. In each epoch, an arm of the bandit is pulled to choose the parameters of the sub-routine and the cumulative rewards received are used to update the posterior. The space of all possible parameters is discretized and the arms of the bandit are considered to be $\mathcal{T} = \{T^0, T^{1/\lfloor \ln T \rfloor}, T^{2/\lfloor \ln T \rfloor}, \ldots, T\}$. In each epoch $i$, arm $j_i$ is pulled/sampled from the distribution $p_i$ (line 5) as

$$p_{i,j} = (1 - \zeta)\frac{\exp(\xi u_{i,j})}{\sum_j \exp(\xi u_{i,j})} + \frac{\zeta}{\lceil \ln T \rceil},$$

and step-sizes and number of restarts for the NS-NAC sub-routine as chosen as a function of $j_i$ as described in line 6. The cumulative reward $R_{i,j_i}$ observed in epoch $i$ is used to update the posterior (line 8) as

$$u_{i+1,j} = u_{i,j} + \frac{\sigma + \mathbb{I}_{j=j_i} \cdot R_{i,j_i}/W}{p_{i,j}}.$$

We now present an upper bound on the dynamic regret with the proof adapted from [16, 12] which present parameter-free non-stationary model-free value-based and model-based algorithms respectively. While we defer the proof to Section F, we would like to highlight the additional cost of $\tilde{\mathcal{O}}\left(W\sqrt{\ln T \cdot \frac{T}{W}}\right)$ incurred by running the EXP3.P algorithm to hedge against the unknown variation budgets and an extra constant factor in the other terms. Note that for the optimal choice of parameters as discussed below, this additional regret term continues to be of the same order as the other terms $\tilde{\mathcal{O}}\left(\Delta_T^{1/6}T^{5/6}\right)$ and hence BORL-NS-NAC continues to have the same order of regret as NS-NAC.

**Theorem 3.** *If Assumption 1 is satisfied, the time horizon $T$ is divided into epochs of length $W = \mathcal{O}(T^{2/3})$ in Algorithm 2, then we have for any $j^\dagger \in \{0, 1, \cdots, \lfloor \ln T \rfloor\}$*

$$Dyn\text{-}Reg(\mathcal{M}, T) \leq \underbrace{\tilde{\mathcal{O}}\left(W\sqrt{\ln T \cdot \frac{T}{W}}\right)}_{\text{cost of hedging by EXP3.P}} + \underbrace{\tilde{\mathcal{O}}\left(\frac{TN^\dagger}{W\beta^\dagger}\right) + \tilde{\mathcal{O}}\left(\frac{T}{W}\sqrt{\frac{N^\dagger W}{\alpha^\dagger}}\right)}_{\text{effect of initialization}} + \underbrace{\tilde{\mathcal{O}}\left(\frac{\beta^\dagger T}{\alpha^\dagger}\right) + \tilde{\mathcal{O}}\left(T\sqrt{\beta^\dagger}\right)}_{\substack{\text{cumulative change} \\ \text{in policy over horizon } T}} \quad (6)$$

$$+ \underbrace{\tilde{\mathcal{O}}\left(\frac{\beta^\dagger T}{\gamma^\dagger}\right) + \tilde{\mathcal{O}}\left(T\sqrt{\gamma^\dagger}\right) + \tilde{\mathcal{O}}\left(\frac{T}{W}\sqrt{\frac{N^\dagger W}{\gamma}}\right)}_{\text{error in average reward estimate at critic}} + \underbrace{\tilde{\mathcal{O}}\left(T\sqrt{\alpha^\dagger}\right)}_{\substack{\text{cumulative change} \\ \text{in critic estimates}}} + \underbrace{\tilde{\mathcal{O}}\left(\frac{\Delta_T W}{N^\dagger}\right) + \tilde{\mathcal{O}}\left(\frac{\Delta_T^{1/3}T^{2/3}}{\sqrt{\alpha^\dagger}} + \frac{\Delta_T^{1/3}T^{2/3}}{\sqrt{\gamma^\dagger}}\right)}_{\text{error due to non-stationarity}},$$

*where $\Delta_T = \Delta_{R,T} + \Delta_{P,T}$, $\alpha^\dagger = \gamma^\dagger = \left(\frac{T^{j^\dagger/\lfloor \ln T \rfloor}}{T}\right)^{1/3}, \beta^\dagger = \left(\frac{T^{j^\dagger/\lfloor \ln T \rfloor}}{T}\right)^{1/2}, N^\dagger = W\left(\frac{T^{j^\dagger/\lfloor \ln T \rfloor}}{T}\right)^{5/6}$ and $\tilde{\mathcal{O}}(\cdot)$ hides constants and logarithmic dependence on time horizon $T$. Choosing optimal value of $j^\dagger$ and resulting optimal parameters as $\alpha^\star = \gamma^\star = \left(\frac{\Delta_T}{T}\right)^{1/3}, \beta^\star = \left(\frac{\Delta_T}{T}\right)^{1/2}$ and $\frac{N^\star T}{W} = \Delta_T^{5/6}T^{1/6}$, we upper bound the regret as*

$$Dyn\text{-}Reg(\mathcal{M}, T) \leq \tilde{\mathcal{O}}\left(|\mathcal{S}|^{1/2}|\mathcal{A}|^{1/2}\Delta_T^{1/6}T^{5/6}\right). \quad (7)$$

## 7    Proof Sketch of Theorem 1

We now present a sketch of the proof of Theorem 1 that presents an upper bound on regret of NS-NAC and address the following theoretical challenges that non-stationarity poses. (a) Stationary environment NAC analyses use the KL-divergence to the optimal policy as a Lyapunov function. What is an appropriate function for dynamic environments where the optimal policy varies with time? (b) How do the simultaneously varying environment and evolving policy affect the estimation of the average reward and state-action value function? (c) How do the time-varying transition probabilities affect the martingale-based argument used to analyze the Markovian noise?

**Regret Decomposition.**    We start by decomposing as

$$\text{Dyn-Reg}(\mathcal{M}, T) = \sum_{t=0}^{T-1} \underbrace{\mathbb{E}\left[J_t^{\boldsymbol{\pi}_t^\star} - J_t^{\boldsymbol{\pi}_t}\right]}_{I_1: \substack{\text{difference of optimal versus} \\ \text{actual average reward}}} + \underbrace{\mathbb{E}\left[J_t^{\boldsymbol{\pi}_t} - r_t(s_t, a_t)\right]}_{I_2: \substack{\text{difference of actual versus} \\ \text{instantaneous reward}}}, \quad (8)$$

where $I_1$ measures the performance difference between the average reward of the actual policy $\boldsymbol{\pi}_t$ at time $t$ relative to the optimal policy $\boldsymbol{\pi}_t^\star$. The second term $I_2$ analyzes the gap between the average reward and the actual rewards received due to the stochasticity of the Markovian sampling process.

**Actor (Proposition 1).**    We first bound $I_1$ in (8) by adapting the Natural Policy Gradient analysis for average-reward stationary MDPs in [41] to non-stationary environments. NPG in the stationary case is analyzed by characterizing the drift of the policy towards the optimal policy using an appropriate Lyapunov function. In non-stationary case we innovatively separate out and analyze the change in the environment from the drift of the policy as follows. We start by dividing the total horizon $T$ into $N$ restarted segments of length $H$ each and split $I_1$ as

$$I_1 = \mathbb{E}\left[\sum_{n=0}^{N-1}\sum_{h=0}^{H-1} \underbrace{\left(J_{nH+h}^{\boldsymbol{\pi}_{nH+h}^\star} - J_{nH}^{\boldsymbol{\pi}_{nH}^\star}\right)}_{I_3: \substack{\text{optimal avg. reward} \\ \text{across two environments}}} + \underbrace{\left(J_{nH}^{\boldsymbol{\pi}_{nH}^\star} - J_{nH}^{\boldsymbol{\pi}_{nH+h}}\right)}_{I_4: \substack{\text{avg. reward} \\ \text{sub-optimality}}} + \underbrace{\left(J_{nH}^{\boldsymbol{\pi}_{nH+h}} - J_{nH+h}^{\boldsymbol{\pi}_{nH+h}}\right)}_{I_5: \substack{\text{avg. reward with same} \\ \text{policy in two environments}}}\right].$$

We benchmark policies learned in each segment $n \in [N]$ against the optimal average reward at the initial timestep $nH$ i.e. $J_{nH}^{\boldsymbol{\pi}_{nH}^\star}$. We bound $I_4$ by mirror descent style analysis for each segment $n$ with $t =$

$\{nH, \ldots, (n+1)H - 1\}$ by the Lyapunov function adapted to non-stationarity as

$$W(\boldsymbol{\pi}_t) = \sum_s d^{\boldsymbol{\pi}^\star_{nH}, \mathbf{P}_{nH}}(s) D_{\mathrm{KL}}(\boldsymbol{\pi}^\star_{nH}(\cdot|s)\|\boldsymbol{\pi}_t(\cdot|s)).$$

In addition, since NS-NAC does not have access to the exact value functions $\mathbf{Q}^{\boldsymbol{\pi}_t}_t$, $I_4$ also depends on the critic estimation error $\|\mathbf{Q}^{\boldsymbol{\pi}_t}_t - \mathbf{Q}_t\|_\infty$.

We analyze the change in the environment next. We bound $I_3$, the difference in optimal average rewards in two different environments, in terms of corresponding changes in the environment $\|\mathbf{r}_{nH+h} - \mathbf{r}_{nH}\|_\infty$ and $\|\mathbf{P}_{nH+h} - \mathbf{P}_{nH}\|_\infty$ (Lemma 5) by a clever use of the linear programming formulation of an MDP. Similarly, we deftly bound $I_5$, the difference in average rewards when following the same policy $\boldsymbol{\pi}_{nH+h}$ in two different environments, in terms of the environment change (Lemma 6). The number of restarts $N$ balances exploration-for-change and learning a good policy and we optimize it in Theorem 1 to minimize regret.

**Critic (Proposition 2).** We bound the critic estimation error $\boldsymbol{\psi}_t = \Pi_E\left[\mathbf{Q}_t - \mathbf{Q}^{\boldsymbol{\pi}_t}_t\right]$ [2] for each restarted segment $n \in [N]$ where $t = \{nH, \ldots, (n+1)H - 1\}$ by adapting the critic analysis used in stationary MDPs [33, 42, 46] to non-stationary environments. If $O_t = (s_t, a_t, s_{t+1}, a_{t+1})$, we can decompose the error as

$$\|\boldsymbol{\psi}_{t+1}\|_2^2 \lesssim (1-\alpha)\|\boldsymbol{\psi}_t\|_2^2 + \alpha \underbrace{\boldsymbol{\psi}_t^\top \left[ (\mathbf{r}_t(O_t) - \mathbf{J}^{\boldsymbol{\pi}_t}_t(O_t) + \mathbf{A}(O_t)\mathbf{Q}^{\boldsymbol{\pi}_t}_t) + \left(\mathbf{A}(O_t) - \bar{\mathbf{A}}^{\boldsymbol{\pi}_t, \mathbf{P}_t}\right) \boldsymbol{\psi}_t\right]}_{I_6:\text{error due to Markov noise}}$$

$$+ \alpha \underbrace{(\mathbf{J}^{\boldsymbol{\pi}_t}_t(O_t) - \boldsymbol{\eta}_t(O_t))^2}_{I_7:\text{avg. reward estimation error}} + \frac{1}{\alpha}\underbrace{\|\Pi_E\left[\mathbf{Q}^{\boldsymbol{\pi}_t}_t - \mathbf{Q}^{\boldsymbol{\pi}_{t+1}}_{t+1}\right]\|_2^2}_{I_8:\text{value function drift}} + \alpha^2 \underbrace{\|\mathbf{r}_t(O_t) - \boldsymbol{\eta}_t(O_t) + \mathbf{A}(O_t)\mathbf{Q}_t\|_2^2}_{I_9:\text{variance term}}. \quad (9)$$

$I_6$ is the error induced by the Markovian noise which is analyzed leveraging the auxiliary Markov chain described below. $I_7$ describes the error due to an inaccurate estimation of the average reward which is bounded below. $I_8$, the change in the true value function is caused by drifting policies and environments, and can be neatly bounded in terms of the change in policy, rewards and transition probabilities (Lemma 9). Finally, $I_{10}$ is the variance term.

**Bound on Markovian Noise.** For each restarted segment $n \in [N]$, consider time indices $(n+1)H > t > \tau > nH$. Consider the *auxiliary Markov chain* starting from $s_{t-\tau}$ constructed by conditioning on $\mathcal{F}_{t-\tau} = \{s_{t-\tau}, \boldsymbol{\pi}_{t-\tau-1}, \mathbf{P}_{t-\tau}\}$ and rolling out by applying $\boldsymbol{\pi}_{t-\tau-1}, \mathbf{P}_{t-\tau}$ as

$$s_{t-\tau} \xrightarrow{\boldsymbol{\pi}_{t-\tau-1}} a_{t-\tau} \xrightarrow{\mathbf{P}_{t-\tau}} \tilde{s}_{t-\tau+1} \xrightarrow{\boldsymbol{\pi}_{t-\tau-1}} \tilde{a}_{t-\tau+1} \xrightarrow{\cdots} \tilde{s}_t \xrightarrow{\boldsymbol{\pi}_{t-\tau-1}} \tilde{a}_t \xrightarrow{\mathbf{P}_{t-\tau}} \tilde{s}_{t+1} \xrightarrow{\boldsymbol{\pi}_{t-\tau-1}} \tilde{a}_{t+1}.$$

Recall that the *original Markov chain* is

$$s_{t-\tau} \xrightarrow{\boldsymbol{\pi}_{t-\tau-1}} a_{t-\tau} \xrightarrow{\mathbf{P}_{t-\tau}} s_{t-\tau+1} \xrightarrow{\boldsymbol{\pi}_{t-\tau}} a_{t-\tau+1} \xrightarrow{\cdots} s_t \xrightarrow{\boldsymbol{\pi}_{t-1}} a_t \xrightarrow{\mathbf{P}_t} s_{t+1} \xrightarrow{\boldsymbol{\pi}_t} a_{t+1}.$$

This enables us to characterize properties of original Markov chain compared to the auxiliary chain as $d_{TV}(P(O_t \in \cdot|\mathcal{F}_{t-\tau}), P(\tilde{O}_t \in \cdot|\mathcal{F}_{t-\tau}))$ where $\tilde{O}_t = (\tilde{s}_t, \tilde{a}_t, \tilde{s}_{t+1}, \tilde{a}_{t+1})$. We do this by bounding the effects of drifting policies and transition probabilities in the original chain and leveraging uniform ergodicity in the auxiliary chain. While prior works use auxiliary Markov chains for stationary environments [49, 33, 43], ours is the first adaptation to a non-stationary environment. Observe that the time-varying transition probabilities $\mathbf{P}_t$ add an extra layer of complexity, unlike the stationary case where only the policy changes over time.

**Average Reward Estimation Error (Proposition 4).** To bound $I_7$ in (9), i.e., the error in the average reward estimate $\phi_t = \boldsymbol{\eta}_t - J^{\boldsymbol{\pi}_t}_t$, we can decompose the error as

$$\phi_{t+1}^2 \lesssim (1-\gamma)\phi_t^2 + \underbrace{\gamma(r_t(O_t) - J^{\boldsymbol{\pi}_t}_t)^2}_{I_{10}:\text{error due to Markov noise}} + \frac{1}{\gamma}\underbrace{(J^{\boldsymbol{\pi}_t}_t - J^{\boldsymbol{\pi}_{t+1}}_{t+1})^2}_{I_{11}:\text{avg reward at consecutive timesteps}} + \underbrace{\gamma^2 (r_t(O_t) - \boldsymbol{\eta}_t)^2}_{I_{12}:\text{variance term}}. [6]$$

---

[2] $\Pi_E[\mathbf{x}] = \arg\min_{\mathbf{y} \in E} \|\mathbf{x} - \mathbf{y}\|_2$ is the projection to $E$, the subspace orthogonal to the all ones vector $\mathbf{1}$.

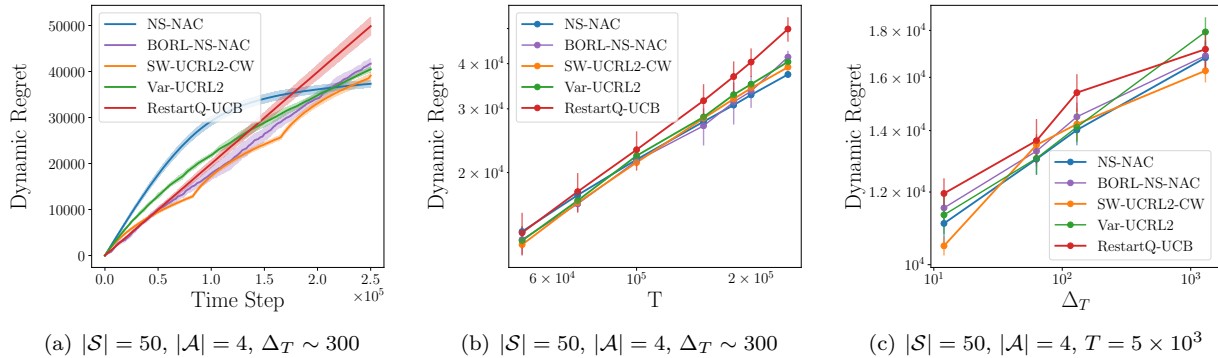

(a) $|\mathcal{S}| = 50$, $|\mathcal{A}| = 4$, $\Delta_T \sim 300$     (b) $|\mathcal{S}| = 50$, $|\mathcal{A}| = 4$, $\Delta_T \sim 300$     (c) $|\mathcal{S}| = 50$, $|\mathcal{A}| = 4$, $T = 5 \times 10^3$

Figure 1: Performance of NS-NAC and baseline algorithms across various settings. (a) Dynamic regret for a single instance with $T = 25 \times 10^4$ steps. Log-log plots showing the effect of varying: (b) time horizon $T$, and (c) variation budget $\Delta_T$.

$I_{10}$ is analyzed using the auxiliary Markov chain construction. $I_{11}$ quantifies the difference in average rewards at consecutive timesteps, and is neatly bounded in Lemma 6 in terms of the corresponding changes in policies, rewards, and transition probabilities. $I_{12}$ is again the variance term.

Finally, $I_2$ in (8) characterizes the difference between the average reward and the instantaneous reward at any time, and is analyzed in Proposition 3 using the auxiliary Markov chain to bound the bias occurring due to Markovian sampling. This concludes the proof sketch.

## 8 Simulations

We empirically evaluate the performance of our algorithms on a synthetic non-stationary MDP (see Section G), comparing it with three baseline algorithms: SW-UCRL2-CW [51], Var-UCRL2 [13], and RestartQ-UCB [16]. SW-UCRL2-CW is a model-based algorithm that adapts to non-stationarity by maintaining a sliding window of recent observations, applying extended value iteration, and adjusting confidence intervals to track changing dynamics. Var-UCRL2, also model-based, adjusts its confidence intervals dynamically based on the observed variations in rewards and transitions. Model-free RestartQ-UCB periodically restarts Q-learning and resets its upper confidence bounds to adapt to non-stationarity. While there is a gap between our theoretical regret analysis and those of the baseline methods, we empirically observe in Figure 1 that NS-NAC and BORL-NS-NAC strongly match their performance achieving sub-linear dynamic regret across all settings. Further, we observe in Figure 1(b) and Figure 1(c) the sub-linear effect of varying time-horizon $T$ and variation budget $\Delta_T$ on the dynamic regret. See Section G for more experimental analysis.

## 9 Conclusion

We consider the problem of non-stationary reinforcement learning in the infinite-horizon average-reward setting and model it as an MDP with time-varying rewards and transition probabilities. We propose and analyze the first model-free policy-based algorithm, Non-Stationary Natural Actor-Critic. A two-timescale natural policy gradient based method, NS-NAC utilizes restarts to explore for change and learning rates as adapting factors to balance forgetting old and learning new environments. Further, we present a bandit-over-RL based parameter-free algorithm BORL-NS-NAC that does not require prior knowledge of the variation budget and adaptively tunes step-sizes and number of restarts. Both algorithms achieve a sub-linear dynamic regret, thus, theoretically validating policy gradient methods often used in practice in continual non-stationary RL.

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

## 10   Appendix

## Contents

# A    Additional Related Work

Table 1: Regret comparison across Non-Stationary and Stationary RL algorithms with variation budget $\Delta_T$, time horizon $T$, episode length $H$, size of the state-action space $|\mathcal{S}|$, $|\mathcal{A}|$, maximum diameter of MDP $D$, dimension of feature space $d$ and dynamic Bellman Eluder dimension $\tilde{d}$.

| Setting | Algorithm | Regret | Model Free | Policy Based |
|---|---|---|---|---|
| | Lower Bound | $\Omega\left(|\mathcal{S}|^{\frac{1}{3}}|\mathcal{A}|^{\frac{1}{3}}D^{\frac{2}{3}}\Delta_T^{\frac{1}{3}}T^{\frac{2}{3}}\right)$ | - | - |
| | [15] | $\tilde{\mathcal{O}}\left(|\mathcal{S}||\mathcal{A}|^{\frac{1}{2}}DL^{\frac{1}{3}}T^{\frac{2}{3}}\right)$ | $\times$ | - |
| Non-Stationary | [14] | $\tilde{\mathcal{O}}\left(|\mathcal{S}|^{\frac{2}{3}}|\mathcal{A}|^{\frac{1}{3}}D^{\frac{2}{3}}L^{\frac{1}{3}}T^{\frac{2}{3}}\right)$ | $\times$ | - |
| Infinite Horizon | [13] | $\tilde{\mathcal{O}}\left(|\mathcal{S}||\mathcal{A}|^{\frac{1}{2}}D\Delta_T^{\frac{1}{3}}T^{\frac{2}{3}}\right)$ | $\times$ | - |
| Average Reward | [12] | $\tilde{\mathcal{O}}\left(|\mathcal{S}|^{\frac{2}{3}}|\mathcal{A}|^{\frac{1}{2}}D\Delta_T^{\frac{1}{4}}T^{\frac{3}{4}}\right)$ | $\times$ | - |
| | [24] | $\tilde{\mathcal{O}}\left(\Delta_T^{\frac{1}{3}}T^{\frac{2}{3}}\right)$ | $\times$ | - |
| | This Work | $\tilde{\mathcal{O}}\left(|\mathcal{S}|^{\frac{1}{2}}|\mathcal{A}|^{\frac{1}{2}}\Delta_T^{\frac{1}{6}}T^{\frac{5}{6}}\right)$ | $\checkmark$ | $\checkmark$ |
| | Lower Bound | $\Omega\left(|\mathcal{S}|^{\frac{1}{3}}|\mathcal{A}|^{\frac{1}{3}}\Delta_T^{\frac{1}{3}}H^{\frac{2}{3}}T^{\frac{2}{3}}\right)$ | - | - |
| | [53] | $\tilde{\mathcal{O}}\left(|\mathcal{S}||\mathcal{A}|^{\frac{1}{2}}\Delta_T^{\frac{1}{3}}H^{\frac{4}{3}}T^{\frac{2}{3}}\right)$ | $\checkmark$ | $\times$ |
| Non-Stationary | [24] | $\tilde{\mathcal{O}}\left(\Delta_T^{\frac{1}{3}}T^{\frac{2}{3}}\right)$ | $\checkmark$ | $\times$ |
| Episodic | [17] | $\tilde{\mathcal{O}}\left(\tilde{d}^{\frac{1}{2}}H^2T^{\frac{1}{2}}\right)$ | $\checkmark$ | $\times$ |
| | [16] | $\tilde{\mathcal{O}}\left(|\mathcal{S}|^{\frac{1}{3}}|\mathcal{A}|^{\frac{1}{3}}\Delta_T^{\frac{1}{3}}HT^{\frac{2}{3}}\right)$ | $\checkmark$ | $\times$ |
| Non-Stationary | [18] | $\tilde{\mathcal{O}}\left(d^{\frac{4}{3}}\Delta_T^{\frac{1}{3}}H^{\frac{4}{3}}T^{\frac{2}{3}}\right)$ | $\checkmark$ | $\times$ |
| Episodic Linear MDP | [19] | $\tilde{\mathcal{O}}\left(d^{\frac{5}{4}}\Delta_T^{\frac{1}{4}}H^{\frac{5}{4}}T^{\frac{3}{4}}\right)$ | $\checkmark$ | $\times$ |
| Stationary Infinite Horizon Discounted Reward | [42] | $\tilde{\mathcal{O}}\left(T^{\frac{5}{6}}\right)$ | $\checkmark$ | $\checkmark$ |
| Stationary Infinite Horizon Average Reward | [43] | $\tilde{\mathcal{O}}\left(T^{\frac{2}{3}}\right)$ | $\checkmark$ | $\checkmark$ |

## B Notation

**Variation Budgets**

$$\Delta_{R,T} = \sum_{t=0}^{T-1} \|\mathbf{r}_{t+1} - \mathbf{r}_t\|_\infty; \Delta_{R,t-\tau+1,t} = \sum_{i=t-\tau+1}^{t} \|\mathbf{r}_i - \mathbf{r}_{i-1}\|_\infty,$$

$$\Delta_{P,T} = \sum_{t=0}^{T-1} \|\mathbf{P}_{t+1} - \mathbf{P}_t\|_\infty; \Delta_{P,t-\tau+1,t} = \sum_{i=t-\tau+1}^{t} \|\mathbf{P}_i - \mathbf{P}_{i-1}\|_\infty,$$

$$\Delta_T = \Delta_{R,T} + \Delta_{P,T}.$$

The critic update (line 9 in Algorithm 1) can be defined in vector form using the following notation. Note that we use a one-to-one mapping $f : \mathcal{S} \times \mathcal{A} \to \{1, 2, \ldots, |\mathcal{S}||\mathcal{A}|\}$, to map state-action pairs $(s, a) \in \mathcal{S} \times \mathcal{A}$ to vector/matrix entries. However, for ease of notation, we denote the index of each entry by $(s, a)$, instead of the more accurate $f(s, a)$.

$$O_t = (s_t, a_t, s_{t+1}, a_{t+1})$$

$$\mathbf{r}_t(O_t) = [0; \cdots ; 0; r_t(s_t, a_t); 0; \cdots ; 0]^\top \in \mathbb{R}^{|\mathcal{S}||\mathcal{A}|}$$

$$\eta_t(O_t) = [0; \cdots ; 0; \eta_t; 0; \cdots ; 0]^\top \in \mathbb{R}^{|\mathcal{S}||\mathcal{A}|}$$

$$\mathbf{J}_t^{\boldsymbol{\pi}}(O_t) = [0; \cdots ; 0; J_t^{\boldsymbol{\pi}}; 0; \cdots ; 0]^\top \in \mathbb{R}^{|\mathcal{S}||\mathcal{A}|}$$

$$\mathbf{A}(O) \in \mathbb{R}^{|\mathcal{S}||\mathcal{A}| \times |\mathcal{S}||\mathcal{A}|} \quad \text{such that}$$

$$\mathbf{A}(O)_{i,j} = \mathbf{A}(s, a, s', a')_{i,j} = \begin{cases} -1 & \text{if } (s,a) \neq (s',a'), i = j = (s,a) \\ 1 & \text{if } (s,a) \neq (s',a'), i = (s,a), j = (s',a') \\ 0 & \text{else} \end{cases}$$

As a result, we get the critic update

$$\mathbf{Q}_{t+1} = \prod_{R_Q} [\mathbf{Q}_t + \alpha \left( \mathbf{r}_t(O_t) - \eta_t(O_t) + \mathbf{A}(O_t)\mathbf{Q}_t \right)].$$

For the purpose of analysis, we define the following quantities.

$$\bar{\mathbf{A}}^{\boldsymbol{\pi},\mathbf{P}} = \mathbb{E}_{s \sim d^{\boldsymbol{\pi},\mathbf{P}}(\cdot), a \sim \boldsymbol{\pi}(\cdot|s), s' \sim \mathbf{P}(\cdot|s,a), a' \sim \boldsymbol{\pi}(\cdot|s')} \left[ \mathbf{A}(s, a, s', a') \right]$$

$$\mathbf{Q}^{\boldsymbol{\pi},\mathbf{P},\mathbf{r}} = \mathbf{Q} \quad \text{associated with} \quad \boldsymbol{\pi}, \mathbf{P}, \mathbf{r}$$

$$J^{\boldsymbol{\pi},\mathbf{P},\mathbf{r}} = \sum_s d^{\boldsymbol{\pi},\mathbf{P}}(s) \sum_a \boldsymbol{\pi}(a|s) r(s, a)$$

$$\Pi_E[\mathbf{x}] = \arg\min_{\mathbf{y} \in E} \|\mathbf{x} - \mathbf{y}\|_2 \text{ where } E \text{ is the subspace orthogonal to the all ones vector } \mathbf{1}$$

$$\boldsymbol{\psi}_t = \Pi_E \left[ \mathbf{Q}_t - \mathbf{Q}_t^{\boldsymbol{\pi}_t} \right] \qquad \text{(Error in the value-function estimate)}$$

$$\Gamma(\boldsymbol{\pi}, \mathbf{P}, \mathbf{r}, \psi, O) = \boldsymbol{\psi}^\top \left( \mathbf{r}(O) - \mathbf{J}^{\boldsymbol{\pi},\mathbf{P},\mathbf{r}}(O) + \mathbf{A}(O)\mathbf{Q}^{\boldsymbol{\pi},\mathbf{P},\mathbf{r}} \right) + \boldsymbol{\psi}^\top \left( \mathbf{A}(O) - \bar{\mathbf{A}}^{\boldsymbol{\pi},\mathbf{P}} \right) \boldsymbol{\psi}$$

$$\phi_t = \eta_t - J_t^{\boldsymbol{\pi}_t} \qquad \text{(Error in the average reward estimate)}$$

$$\Lambda(\boldsymbol{\pi}, \mathbf{P}, \mathbf{r}, \eta, O) = (\eta - J^{\boldsymbol{\pi},\mathbf{P},\mathbf{r}})(r(s, a) - J^{\boldsymbol{\pi},\mathbf{P},\mathbf{r}})$$

Given time indices $t > \tau > 0$, consider the *auxiliary Markov chain* starting from $s_{t-\tau}$ constructed by conditioning on $s_{t-\tau}, \boldsymbol{\pi}_{t-\tau-1}, \mathbf{P}_{t-\tau}$ and rolling out by applying $\boldsymbol{\pi}_{t-\tau-1}, \mathbf{P}_{t-\tau}$ as

$$s_{t-\tau} \xrightarrow{\boldsymbol{\pi}_{t-\tau-1}} a_{t-\tau} \xrightarrow{\mathbf{P}_{t-\tau}} \tilde{s}_{t-\tau+1} \xrightarrow{\boldsymbol{\pi}_{t-\tau-1}} \tilde{a}_{t-\tau+1} \xrightarrow{\mathbf{P}_{t-\tau}} \ldots \tilde{s}_t \xrightarrow{\boldsymbol{\pi}_{t-\tau-1}} \tilde{a}_t \xrightarrow{\mathbf{P}_{t-\tau}} \tilde{s}_{t+1} \xrightarrow{\boldsymbol{\pi}_{t-\tau-1}} \tilde{a}_{t+1}.$$

Recall that the *original Markov chain* is

$$s_{t-\tau} \xrightarrow{\boldsymbol{\pi}_{t-\tau-1}} a_{t-\tau} \xrightarrow{\mathbf{P}_{t-\tau}} s_{t-\tau+1} \xrightarrow{\boldsymbol{\pi}_{t-\tau}} a_{t-\tau+1} \xrightarrow{\mathbf{P}_{t-\tau+1}} \ldots s_t \xrightarrow{\boldsymbol{\pi}_{t-1}} a_t \xrightarrow{\mathbf{P}_t} s_{t+1} \xrightarrow{\boldsymbol{\pi}_t} a_{t+1}.$$

## C  Symbol Reference

| Constant | First Appearance |
|---|---|
| $U_R$ | Section 3.1 |
| $U_Q$ | Lemma 24 |
| $N, H$ | Algorithm 1 |
| $C = \inf\limits_{s,t,t',\boldsymbol{\pi}} \dfrac{d^{\boldsymbol{\pi}, \mathbf{P}_{t'}}(s)}{d^{\boldsymbol{\pi}^\star, \mathbf{P}_t}(s)}$ | Assumption 1 |
| $m, \rho$ | Assumption 1 |
| $M = \lceil \log_\rho m^{-1} \rceil + \frac{1}{1-\rho}$ | Lemma 22 |
| $\lambda$ | Lemma 1 |
| $W_1 = (3G_R^2)^{1/3}(4U_Q^2)^{2/3}$ | Proposition 2 |
| $W_2 = (3G_P^2)^{1/3}(4U_Q^2)^{2/3}$ | Proposition 2 |
| $D_1 = L_{\boldsymbol{\pi}} B_2 + 4U_R \sqrt{|\mathcal{S}||\mathcal{A}|} B_2 + 4U_R$ | Proposition 3 |
| $D_2 = 4U_R + L_P$ | Proposition 3 |
| $D_3 = 4U_R F_6 + 8U_R^2$ | Proposition 4 |
| $D_4 = 9L_{\boldsymbol{\pi}}^2 B_2^2$ | Proposition 4 |
| $W_3 = (3)^{1/3}(4U_R^2)^{2/3}$ | Proposition 4 |
| $W_4 = (3L_P^2)^{1/3}(4U_R^2)^{2/3}$ | Proposition 4 |
| $B_1 = 2\sqrt{|\mathcal{A}|} U_Q^2$ | Lemma 2 |
| $B_2 = U_Q$ | Lemma 3 |
| $B_3 = (F_{1\boldsymbol{\pi}} + G_{\boldsymbol{\pi}} + F_3 \sqrt{|\mathcal{S}||\mathcal{A}|} + F_4) B_2$ | Lemma 10 |
| $B_4 = F_2(2U_R + 2U_Q)$ | Lemma 10 |
| $B_5 = F_2 G_R$ | Lemma 10 |
| $B_6 = F_{1\mathbf{P}} + F_2 G_P + F_3$ | Lemma 10 |
| $B_7 = (F_5 L_{\boldsymbol{\pi}} + F_7 \sqrt{|\mathcal{S}||\mathcal{A}|} + F_8) B_2$ | Lemma 11 |
| $B_8 = F_7 + F_5 L_P$ | Lemma 11 |
| $L_{\boldsymbol{\pi}} = 4U_R(M+1)\sqrt{|\mathcal{S}||\mathcal{A}|}$ | Lemma 6 |
| $L_P = 4U_R M$ | Lemma 6 |
| $G_{\boldsymbol{\pi}} = 2U_Q \sqrt{|\mathcal{S}||\mathcal{A}|}$ | Lemma 7 |
| $G_R = 2\lambda^{-1} \sqrt{|\mathcal{S}||\mathcal{A}|}$ | Lemma 8 |
| $G_P = (\lambda^{-1} L_P + 4U_R \lambda^{-1} M + 4U_R \lambda^{-2}(M+1))\sqrt{|\mathcal{S}||\mathcal{A}|}$ | Lemma 8 |
| $F_{1\boldsymbol{\pi}} = 2U_Q L_{\boldsymbol{\pi}} + 4U_Q G_{\boldsymbol{\pi}} + 8U_Q^2(M+2)|\mathcal{S}||\mathcal{A}|$ | Lemma 13 |
| $F_{1P} = 2U_Q L_P + 4U_Q G_P + 8U_Q^2(M+1)\sqrt{|\mathcal{S}||\mathcal{A}|}$ | Lemma 13 |
| $F_2 = 2U_R + 18U_Q$ | Lemma 14 |
| $F_3 = 16U_R U_Q + 24U_Q^2 \sqrt{|\mathcal{S}||\mathcal{A}|}$ | Lemma 15 |
| $F_4 = 8U_R U_Q + 24U_Q^2 \sqrt{|\mathcal{S}||\mathcal{A}|}$ | Lemma 16 |
| $F_5 = 4U_R$ | Lemma 18 |
| $F_6 = 2U_R$ | Lemma 19 |
| $F_7 = 8U_R^2$ | Lemma 20 |
| $F_8 = 8U_R^2$ | Lemma 21 |

## D  Regret Analysis: NS-NAC

**Theorem 4.** *If Assumption 1 is satisfied and the step-sizes are chosen as $0 < \alpha, \beta, \gamma, \epsilon < 1/2$ and number of restarts as $0 < N < T$ in Algorithm 1, then we have*

$$Dyn\text{-}Reg(\mathcal{M}, T) = \mathbb{E}\left[\sum_{t=0}^{T-1} J_t^{\pi_t^\star} - r_t(s_t, a_t)\right] \leq \underbrace{\tilde{\mathcal{O}}\left(\frac{N}{\beta}\right) + \tilde{\mathcal{O}}\left(\sqrt{\frac{NT}{\alpha}}\right)}_{\text{effect of initialization}} + \underbrace{\tilde{\mathcal{O}}\left(\frac{\beta T}{\alpha}\right) + \tilde{\mathcal{O}}\left(T\sqrt{\beta}\right)}_{\substack{\text{cumulative change} \\ \text{in policy over horizon } T}}$$

$$+ \underbrace{\tilde{\mathcal{O}}\left(\frac{\beta T}{\gamma}\right) + \tilde{\mathcal{O}}\left(T\sqrt{\gamma}\right) + \tilde{\mathcal{O}}\left(\sqrt{\frac{NT}{\gamma}}\right)}_{\text{error in average reward estimate at critic}} + \underbrace{\tilde{\mathcal{O}}\left(T\sqrt{\alpha}\right)}_{\substack{\text{cumulative change} \\ \text{in critic estimates}}} + \underbrace{\tilde{\mathcal{O}}\left(\frac{\Delta_T T}{N}\right) + \tilde{\mathcal{O}}\left(\frac{\Delta_T^{1/3} T^{2/3}}{\sqrt{\alpha}} + \frac{\Delta_T^{1/3} T^{2/3}}{\sqrt{\gamma}}\right)}_{\text{error due to non-stationarity}},$$

*where $\Delta_T = \Delta_{R,T} + \Delta_{P,T}$, $\tilde{\mathcal{O}}(\cdot)$ hides the constants and logarithmic dependence on the time horizon $T$. Choosing optimal $\alpha^\star = \gamma^\star = \left(\frac{\Delta_T}{T}\right)^{1/3}$, $\beta^\star = \left(\frac{\Delta_T}{T}\right)^{1/2}$ and $N^\star = \Delta_T^{5/6} T^{1/6}$, the resulting regret (with explicit dependence on the size of the state-action space $|\mathcal{S}|, |\mathcal{A}|$) is*

$$Dyn\text{-}Reg(\mathcal{M}, T) \leq \tilde{\mathcal{O}}\left(|\mathcal{S}|^{1/2} |\mathcal{A}|^{1/2} \Delta_T^{1/6} T^{5/6}\right).$$

*Proof.* Recall that Algorithm 1 divides the total time horizon $T$ into $N$ segments of length $H = \lfloor \frac{T}{N} \rfloor$.

$$\mathbb{E}\left[\sum_{t=0}^{T-1} J_t^{\pi_t^\star} - r_t(s_t, a_t)\right] = \mathbb{E}\left[\sum_{t=0}^{T-1} J_t^{\pi_t^\star} - J_t^{\pi_t}\right] + \mathbb{E}\left[\sum_{t=0}^{T-1} J_t^{\pi_t} - r_t(s_t, a_t)\right]$$

$$\overset{(a)}{\leq} \tilde{\mathcal{O}}\left(\frac{\Delta_T T}{N}\right) + \tilde{\mathcal{O}}\left(\frac{N}{\beta}\right) + \tilde{\mathcal{O}}\left(\beta T\right) + 2\sum_{n=0}^{N-1}\sum_{h=0}^{H-1} \mathbb{E}\left[\|\Pi_E\left[\mathbf{Q}_{nH+h}^{\pi_{nH+h}} - \mathbf{Q}_{nH+h}\right]\|_\infty\right]$$

$$+ \mathbb{E}\left[\sum_{t=0}^{T-1} J_t^{\pi_t} - r_t(s_t, a_t)\right]$$

$$\leq \tilde{\mathcal{O}}\left(\frac{\Delta_T T}{N}\right) + \tilde{\mathcal{O}}\left(\frac{N}{\beta}\right) + \tilde{\mathcal{O}}\left(\beta T\right) + 2\sum_{n=0}^{N-1} H^{1/2}\left(\sum_{h=0}^{H-1} \mathbb{E}\left[\|\Pi_E\left[\mathbf{Q}_{nH+h}^{\pi_{nH+h}} - \mathbf{Q}_{nH+h}\right]\|_2^2\right]\right)^{1/2}$$

$$+ \mathbb{E}\left[\sum_{t=0}^{T-1} J_t^{\pi_t} - r_t(s_t, a_t)\right]$$

$$\overset{(b)}{\leq} \tilde{\mathcal{O}}\left(\frac{\Delta_T T}{N}\right) + \tilde{\mathcal{O}}\left(\frac{N}{\beta}\right) + \tilde{\mathcal{O}}\left(\beta T\right) + 2\sum_{n=0}^{N-1}\left[\tilde{\mathcal{O}}\left(\sqrt{H}\right) + \tilde{\mathcal{O}}\left(\sqrt{\frac{H}{\alpha}}\right) + \tilde{\mathcal{O}}\left(\sqrt{\alpha}H\right) + \tilde{\mathcal{O}}\left(\frac{\beta H}{\alpha}\right)\right.$$

$$+ \tilde{\mathcal{O}}\left(\sqrt{\beta}H\right) + \tilde{\mathcal{O}}\left(\frac{\beta H}{\gamma}\right) + \tilde{\mathcal{O}}\left(\sqrt{\gamma}H\right) + \tilde{\mathcal{O}}\left(\sqrt{\frac{H}{\gamma}}\right) + \tilde{\mathcal{O}}\left(\frac{\Delta_{nH,(n+1)H}^{1/3} H^{2/3}}{\sqrt{\alpha}}\right)$$

$$\left. + \tilde{\mathcal{O}}\left(\frac{\Delta_{nH,(n+1)H}^{1/3} H^{2/3}}{\sqrt{\gamma}}\right)\right] + \mathbb{E}\left[\sum_{t=\tau_T}^{T-1} J_t^{\pi_t} - r_t(s_t, a_t)\right]$$

$$\overset{(c)}{\leq} \tilde{\mathcal{O}}\left(\frac{\Delta_T T}{N}\right) + \tilde{\mathcal{O}}\left(\frac{N}{\beta}\right) + \tilde{\mathcal{O}}\left(\beta T\right) + \tilde{\mathcal{O}}\left(N\sqrt{H}\right) + \tilde{\mathcal{O}}\left(\sqrt{\frac{NT}{\alpha}}\right) + \tilde{\mathcal{O}}\left(T\sqrt{\alpha_{,,}}\right) + \tilde{\mathcal{O}}\left(\frac{\beta T}{\alpha}\right)$$

$$+ \tilde{\mathcal{O}}\left(T\sqrt{\beta}\right) + \tilde{\mathcal{O}}\left(\frac{\beta T}{\gamma}\right) + \tilde{\mathcal{O}}\left(T\sqrt{\gamma}\right) + \tilde{\mathcal{O}}\left(\sqrt{\frac{NT}{\gamma}}\right) + \tilde{\mathcal{O}}\left(\frac{\Delta_T^{1/3} T^{2/3}}{\sqrt{\alpha}}\right)$$

$$+ \tilde{\mathcal{O}}\left(\frac{\Delta_T^{1/3} T^{2/3}}{\sqrt{\gamma}}\right) + \tilde{\mathcal{O}}(1) + \tilde{\mathcal{O}}\left(\beta T\right) + \tilde{\mathcal{O}}\left(\Delta_{P,T}\right),$$

where $(a)$ is due to Proposition 1, $(b)$ is by Proposition 2 and $\Delta_{nH,(n+1)H} = \Delta_{R,nH,(n+1)H} + \Delta_{P,nH,(n+1)H}$, $(c)$ is by Jensen's inequality, $\Delta_T = \Delta_{R,T} + \Delta_{P,T}$ and Proposition 3. We further have $\tau_H = \mathcal{O}(\log T)$. Note that $\tilde{\mathcal{O}}(\cdot)$ hides constants and logarithmic terms. $\qquad\square$

### D.1 Actor

The next result bounds the performance difference, measured by the average reward, between the optimal policies $\boldsymbol{\pi}_t^\star$ and the current policy $\boldsymbol{\pi}_t$.

**Proposition 1.** *If Assumption 1 holds, we have*

$$
\mathbb{E}\left[\sum_{t=0}^{T-1} J_t^{\boldsymbol{\pi}_t^\star} - J_t^{\boldsymbol{\pi}_t}\right] \leq \underbrace{\left(2 + 2G_R + \frac{1}{C}\right)\frac{T\Delta_{R,T}}{N} + \left(U_Q + L_P + 2G_P + \frac{L_P}{C}\right)\frac{T\Delta_{P,T}}{N}}_{\text{error due to non-stationarity}}
$$

$$
+ \underbrace{2\sum_{n=0}^{N-1}\sum_{h=0}^{H-1}\mathbb{E}\left[\|\Pi_E\left[\mathbf{Q}_{nH+h}^{\boldsymbol{\pi}_{nH+h}} - \mathbf{Q}_{nH+h}\right]\|_\infty\right]}_{\text{critic estimation error}} + \underbrace{N \cdot \frac{\log|\mathcal{A}|}{\beta}}_{\substack{N\times\text{bias of} \\ \text{initialization}}} + \underbrace{\frac{B_1\beta T}{C}}_{\substack{\text{cumulative} \\ \text{change in policy}}} + \underbrace{\frac{U_R}{C}}_{\text{constant}},
$$

*where $\Delta_{R,T} = \sum_{t=0}^{T-1} \|\mathbf{r}_{r+1} - \mathbf{r}_t\|_\infty$, $\Delta_{P,T} = \sum_{t=0}^{T-1} \|\mathbf{P}_{t+1} - \mathbf{P}_t\|_\infty$, $C$ is defined in Assumption 1, $T$ is the total time horizon and $N$ is the number of restarts. The remaining constants are defined in Section C.*

*Proof.* Recall that Algorithm 1 divides the total time horizon $T$ into $N$ segments of length $H = \lfloor \frac{T}{N} \rfloor$. In each segment (indexed by $n \in [N]$), we use $J_{nH}^{\boldsymbol{\pi}_{nH}^\star}$ as an anchor against which to compare the performance of the learned policies.

$$
\mathbb{E}\left[\sum_{t=0}^{T-1} J_t^{\boldsymbol{\pi}_t^\star} - J_t^{\boldsymbol{\pi}_t}\right] \leq \mathbb{E}\left[\sum_{n=0}^{N-1}\sum_{h=0}^{H-1}\left(J_{nH+h}^{\boldsymbol{\pi}_{nH+h}^\star} - J_{nH}^{\boldsymbol{\pi}_{nH}^\star}\right) + \left(J_{nH}^{\boldsymbol{\pi}_{nH}^\star} - J_{nH+h}^{\boldsymbol{\pi}_{nH+h}}\right) + \left(J_{nH+h}^{\boldsymbol{\pi}_{nH+h}} - J_{nH+h}^{\boldsymbol{\pi}_{nH+h}}\right)\right]
$$

$$
\stackrel{(a)}{\leq} \mathbb{E}\left[\sum_{n=0}^{N-1}\sum_{h=0}^{H-1}\left(2\|\mathbf{r}_{nH+h} - \mathbf{r}_{nH}\|_\infty + (U_Q + L_P)\|\mathbf{P}_{nH+h} - \mathbf{P}_{nH}\|_\infty\right) + \left(J_{nH}^{\boldsymbol{\pi}_{nH}^\star} - J_{nH+h}^{\boldsymbol{\pi}_{nH+h}}\right)\right]
$$

$$
\stackrel{(b)}{\leq} \sum_{n=0}^{N-1}\sum_{h=0}^{H-1} H\left(2\|\mathbf{r}_{nH+h} - \mathbf{r}_{nH+h-1}\|_\infty + (U_Q + L_P)\|\mathbf{P}_{nH+h} - \mathbf{P}_{nH+h-1}\|_\infty\right) + \left(J_{nH}^{\boldsymbol{\pi}_{nH}^\star} - J_{nH+h}^{\boldsymbol{\pi}_{nH+h}}\right)
$$

$$
\leq H\left(2\Delta_{R,T} + (U_Q + L_P)\Delta_{P,T}\right) + \mathbb{E}\left[\sum_{n=0}^{N-1}\sum_{h=0}^{H-1} J_{nH}^{\boldsymbol{\pi}_{nH}^\star} - J_{nH+h}^{\boldsymbol{\pi}_{nH+h}}\right], \tag{10}
$$

where $(a)$ is by Lemma 5 and Lemma 6 and $(b)$ is by triangle inequality. We now bound the last term as

$$
\sum_{n=0}^{N-1}\sum_{h=0}^{H-1} J_{nH}^{\boldsymbol{\pi}_{nH}^\star} - J_{nH+h}^{\boldsymbol{\pi}_{nH+h}}
$$

$$
\stackrel{(c)}{=} \sum_{n=0}^{N-1}\sum_{h=0}^{H-1}\frac{1}{\beta}\sum_s\sum_a d^{\boldsymbol{\pi}_{nH}^\star, \mathbf{P}_{nH}}(s)\boldsymbol{\pi}_{nH}^\star(a|s)\left[\beta Q_{nH+h}^{\boldsymbol{\pi}_{nH+h}}(s,a) - \beta V_{nH}^{\boldsymbol{\pi}_{nH+h}}(s)\right]
$$

$$
= \sum_{n,h,s,a}\frac{d^{\boldsymbol{\pi}_{nH}^\star, \mathbf{P}_{nH}}(s)\boldsymbol{\pi}_{nH}^\star(a|s)}{\beta}\left[\beta Q_{nH+h}^{\boldsymbol{\pi}_{nH+h}}(s,a) - \beta V_{nH+h}^{\boldsymbol{\pi}_{nH+h}}(s) + \beta Q_{nH+h}(s,a) - \beta Q_{nH+h}(s,a)\right]
$$

$$
+ \sum_{n,h,s,a}\frac{d^{\boldsymbol{\pi}_{nH}^\star, \mathbf{P}_{nH}}(s)\boldsymbol{\pi}_{nH}^\star(a|s)}{\beta}\left[\beta Q_{nH}^{\boldsymbol{\pi}_{nH+h}}(s,a) - \beta Q_{nH+h}^{\boldsymbol{\pi}_{nH+h}}(s,a) + \beta V_{nH+h}^{\boldsymbol{\pi}_{nH+h}}(s) - \beta V_{nH}^{\boldsymbol{\pi}_{nH+h}}(s)\right]
$$

$$
= \sum_{n,h,s,a}\frac{d^{\boldsymbol{\pi}_{nH}^\star, \mathbf{P}_{nH}}(s)\boldsymbol{\pi}_{nH}^\star(a|s)}{\beta}\left[\beta Q_{nH+h}^{\boldsymbol{\pi}_{nH+h}}(s,a) - \beta V_{nH+h}^{\boldsymbol{\pi}_{nH+h}}(s) + \beta Q_{nH+h}(s,a) - \beta Q_{nH+h}(s,a)\right]
$$

$$+ \sum_n \sum_h 2\|\mathbf{Q}_{nH}^{\boldsymbol{\pi}_{nH+h}} - \mathbf{Q}_{nH+h}^{\boldsymbol{\pi}_{nH+h}}\|_\infty$$

$$\overset{(d)}{\leq} \sum_{n,h,s,a} \frac{d^{\boldsymbol{\pi}_{nH}^\star, \mathbf{P}_{nH}}(s)\boldsymbol{\pi}_{nH}^\star(a|s)}{\beta} \left[\beta Q_{nH+h}^{\boldsymbol{\pi}_{nH+h}}(s,a) - \beta V_{nH+h}^{\boldsymbol{\pi}_{nH+h}}(s) + \beta Q_{nH+h}(s,a) - \beta Q_{nH+h}(s,a)\right]$$

$$+ \sum_{n=0}^{N-1} \sum_{h=0}^{H-1} 2G_R\|\mathbf{r}_{nH} - \mathbf{r}_{nH+h}\|_\infty + 2G_P\|\mathbf{P}_{nH} - \mathbf{P}_{nH+h}\|_\infty$$

$$\overset{(e)}{=} \sum_n \sum_h \frac{1}{\beta} \sum_s \sum_a d^{\boldsymbol{\pi}_{nH}^\star, \mathbf{P}_{nH}}(s)\boldsymbol{\pi}_{nH}^\star(a|s) \Bigg[\underbrace{\log Z_{nH+h}(s) - \beta V_{nH+h}^{\boldsymbol{\pi}_{nH+h}}(s)}_{I_1}\Bigg]$$

$$+ \sum_{n,h,s,a} \frac{d^{\boldsymbol{\pi}_{nH}^\star, \mathbf{P}_{nH}}(s)\boldsymbol{\pi}_{nH}^\star(a|s)}{\beta} \Bigg[\underbrace{\log \frac{\pi_{nH+h+1}(a|s)}{\pi_{nH+h}(a|s)}}_{I_2} + \underbrace{\beta Q_{nH+h}^{\boldsymbol{\pi}_{nH+h}}(s,a) - \beta Q_{nH+h}(s,a)}_{I_3}\Bigg]$$

$$+ (H-1)(2G_R\Delta_{R,T} + 2G_P\Delta_{P,T}) \tag{11}$$

where $(c)$ follows from the Performance Difference Lemma 4, $(d)$ follows from Lemma 8 and $(e)$ from the actor update equation (line 10 in Algorithm 1) and $Z_t(s) = \sum_{a'\in\mathcal{A}} \pi_t(a'|s)\exp(\beta Q_t(s,a'))$. Next, we bound each of $I_1, I_2, I_3$. Using Lemma 2, we have

$$I_1 = \sum_n \sum_h \sum_s d^{\boldsymbol{\pi}_{nH}^\star, \mathbf{P}_{nH}}(s) \left[\frac{\log Z_{nH+h}(s)}{\beta} - V_{nH+h}^{\boldsymbol{\pi}_{nH+h}}(s)\right] \underbrace{\sum_a \pi_{nH}^\star(a|s)}_{=1}$$

$$\leq \sum_n \sum_h \left[\frac{J_{nH+h+1}^{\boldsymbol{\pi}_{nH+h+1}} - J_{nH+h}^{\boldsymbol{\pi}_{nH+h}}}{C} + \|\mathbf{Q}_{nH+h}^{\boldsymbol{\pi}_{nH+h}} - \mathbf{Q}_{nH+h}\|_\infty + \frac{B_1\beta}{C}\right.$$

$$\left. + \frac{\|\mathbf{r}_{nH+h+1} - \mathbf{r}_{nH+h}\|_\infty}{C} + \frac{L_P\|\mathbf{P}_{nH+h+1} - \mathbf{P}_{nH+h}\|_\infty}{C}\right]. \tag{12}$$

Next, we establish a bound on $I_2$ as

$$I_2 = \frac{1}{\beta} \sum_n \sum_h \sum_s \sum_a d^{\boldsymbol{\pi}_{nH}^\star, \mathbf{P}_{nH}}(s)\boldsymbol{\pi}_{nH}^\star(a|s) \log \frac{\pi_{nH+h+1}(a|s)}{\pi_{nH+h}(a|s)}$$

$$\leq \frac{1}{\beta} \sum_{n=0} \sum_h \sum_s d^{\boldsymbol{\pi}_{nH}^\star, \mathbf{P}_{nH}}(s) \left[D_{\mathrm{KL}}\left(\boldsymbol{\pi}_{nH}^\star(\cdot|s)\|\boldsymbol{\pi}_{nH+h}(\cdot|s)\right) - D_{\mathrm{KL}}\left(\boldsymbol{\pi}_{nH}^\star(\cdot|s)\|\boldsymbol{\pi}_{nH+h+1}(\cdot|s)\right)\right]$$

$$= \frac{1}{\beta} \sum_n \sum_s d^{\boldsymbol{\pi}_{nH}^\star, \mathbf{P}_{nH}}(s) \left[D_{\mathrm{KL}}\left(\boldsymbol{\pi}_{nH}^\star(\cdot|s)\|\boldsymbol{\pi}_{nH}(\cdot|s)\right) - D_{\mathrm{KL}}\left(\boldsymbol{\pi}_{nH}^\star(\cdot|s)\|\boldsymbol{\pi}_{(n+1)H}(\cdot|s)\right)\right]$$

$$\overset{(f)}{\leq} \frac{1}{\beta} \sum_n \sum_s d^{\boldsymbol{\pi}_{nH}^\star, \mathbf{P}_{nH}}(s) D_{\mathrm{KL}}\left(\boldsymbol{\pi}_{nH}^\star(\cdot|s)\|\boldsymbol{\pi}_{nH}(\cdot|s)\right)$$

$$\overset{(g)}{\leq} \frac{1}{\beta} \sum_n \sum_s d^{\boldsymbol{\pi}_{nH}^\star, \mathbf{P}_{nH}}(s) \log \frac{|\mathcal{A}|}{1} \leq \frac{N\log|\mathcal{A}|}{\beta} \tag{13}$$

where $(f)$ is because of non-negativity of KL-divergence and $(g)$ is due to the restart in line 4 of Algorithm 1. Lastly, $I_3$ can be bounded as

$$I_3 = \sum_n \sum_h \sum_s \sum_a d^{\boldsymbol{\pi}_{nH}^\star, \mathbf{P}_{nH}}(s)\boldsymbol{\pi}_{nH}^\star(a|s) \left[Q_{nH+h}^{\boldsymbol{\pi}_{nH+h}}(s,a) - Q_{nH+h}(s,a)\right]$$

$$\leq \sum_n \sum_h \|\mathbf{Q}_{nH+h}^{\boldsymbol{\pi}_{nH+h}} - \mathbf{Q}_{nH+h}\|_\infty. \tag{14}$$

We substitute the bounds on $I_1, I_2, I_3$ from (12)-(14) in (11) and then combine with (10). Recall that the set of solutions to the Bellman equations is $\mathbf{Q}_t^{\boldsymbol{\pi}_t} = \{\mathbf{Q}_{t,E}^{\boldsymbol{\pi}_t} + c\mathbf{1} | \mathbf{Q}_{t,E}^{\boldsymbol{\pi}_t} \in E, c \in \mathbb{R}\}$ where $E$ is the subspace

orthogonal to the all ones vector and $\mathbf{Q}_{t,E}^{\boldsymbol{\pi}_t}$ is the unique solution in $E$ [46]. Finally, we use the equivalence $\|\mathbf{Q}_t^{\boldsymbol{\pi}_t} - \mathbf{Q}_t\|_\infty = \|\Pi_E\left[\mathbf{Q}_t^{\boldsymbol{\pi}_t} - \mathbf{Q}_t\right]\|_\infty$ to get the result. $\qquad\square$

### D.2 Critic

In this section, we characterize the error in the critic estimation.

**Proposition 2.** *For any $n \in [N]$, if Assumption 1 is satisfied and $0 < \gamma < 1/2$, then we have*

$$\sum_{t=nH+\tau_H}^{nH+H-1} \mathbb{E}\left[\|\Pi_E\left[\mathbf{Q}_t - \mathbf{Q}_t^{\boldsymbol{\pi}_t}\right]\|_2^2\right]$$

$$\leq \underbrace{\tilde{\mathcal{O}}\left(\frac{1}{\alpha}\right)}_{\substack{\text{effect of}\\\text{initialization}}} + \underbrace{\tilde{\mathcal{O}}\left(\frac{\Delta_{R,nH,(n+1)H}^{2/3}H^{1/3}}{\alpha}\right) + \tilde{\mathcal{O}}\left(\frac{\Delta_{P,nH,(n+1)H}^{2/3}H^{1/3}}{\alpha}\right)}_{\text{error due to non-stationarity}}$$

$$+ \underbrace{\tilde{\mathcal{O}}\left(\gamma H\right) + \tilde{\mathcal{O}}\left(\frac{1}{\gamma}\right) + \tilde{\mathcal{O}}\left(\frac{\beta^2 H}{\gamma^2}\right) + \tilde{\mathcal{O}}\left(\frac{\Delta_{R,nH,(n+1)H}^{2/3}H^{1/3}}{\gamma}\right) + \tilde{\mathcal{O}}\left(\frac{\Delta_{P,nH,(n+1)H}^{2/3}H^{1/3}}{\gamma}\right)}_{\text{error in average reward estimate at critic}}$$

$$+ \underbrace{\tilde{\mathcal{O}}\left(\beta T\right) + \tilde{\mathcal{O}}\left(\frac{\beta^2 H}{\alpha^2}\right)}_{\substack{\text{cumulative change}\\\text{in policy over horizon } T}} + \underbrace{\tilde{\mathcal{O}}\left(\alpha H\right)}_{\substack{\text{cumulative change}\\\text{in critic estimates}}} ,$$

*where $\tilde{\mathcal{O}}(\cdot)$ hides constants and logarithmic terms which can be found in Equation (17) and $\Delta_{R,nH,(n+1)H} = \sum_{t=nH}^{(n+1)H}\|\mathbf{r}_{t+1} - \mathbf{r}_t\|_\infty$, and $\Delta_{P,nH,(n+1)H} = \sum_{t=nH}^{(n+1)H}\|\mathbf{P}_{t+1} - \mathbf{P}_t\|_\infty$.*

*Proof.* Recall that $\boldsymbol{\psi}_t = \Pi_E\left[\mathbf{Q}_t - \mathbf{Q}_t^{\boldsymbol{\pi}_t}\right]$, $E$ is the subspace orthogonal to the all ones vector $\mathbf{1}$ and the critic update equation (line 9 in Algorithm 1) can be expressed in vector form as $\mathbf{Q}_{t+1} = \Pi_{R_Q}\left[\mathbf{Q}_t + \alpha\left(\mathbf{r}_t(O_t) - \boldsymbol{\eta}_t(O_t) + \mathbf{A}(O_t)\mathbf{Q}_t\right)\right]$. Recall the notations $\mathbf{r}_t, \eta_t, \mathbf{A}(O_t), \bar{\mathbf{A}}^{\boldsymbol{\pi}_t,\mathbf{P}_t}, \mathbf{J}_t(O_t), \Gamma(\cdot), \phi_t$ from Section B. We therefore have

$$\|\boldsymbol{\psi}_{t+1}\|_2^2 = \|\Pi_E\left[\mathbf{Q}_{t+1} - \mathbf{Q}_{t+1}^{\boldsymbol{\pi}_{t+1}}\right]\|_2^2$$

$$\leq \|\Pi_E\left[\mathbf{Q}_t + \alpha\left(\mathbf{r}_t(O_t) - \boldsymbol{\eta}_t(O_t) + \mathbf{A}(O_t)\mathbf{Q}_t\right) - \mathbf{Q}_{t+1}^{\boldsymbol{\pi}_{t+1}}\right]\|_2^2$$

$$= \|\Pi_E\left[\boldsymbol{\psi}_t + \alpha\left(\mathbf{r}_t(O_t) - \boldsymbol{\eta}_t(O_t) + \mathbf{A}(O_t)\mathbf{Q}_t\right) + \mathbf{Q}_t^{\boldsymbol{\pi}_t} - \mathbf{Q}_{t+1}^{\boldsymbol{\pi}_{t+1}}\right]\|_2^2$$

$$\leq \|\boldsymbol{\psi}_t\|_2^2 + 2\alpha\boldsymbol{\psi}_t^\top\left(\mathbf{r}_t(O_t) - \boldsymbol{\eta}_t(O_t)\right) + \mathbf{A}(O_t)\mathbf{Q}_t$$
$$\quad + 2\boldsymbol{\psi}_t^\top\Pi_E\left[\mathbf{Q}_t^{\boldsymbol{\pi}_t} - \mathbf{Q}_{t+1}^{\boldsymbol{\pi}_{t+1}}\right] + 2\alpha^2\|\mathbf{r}_t(O_t) - \boldsymbol{\eta}_t(O_t) + \mathbf{A}(O_t)\mathbf{Q}_t\|_2^2 + 2\|\Pi_E\left[\mathbf{Q}_t^{\boldsymbol{\pi}_t} - \mathbf{Q}_{t+1}^{\boldsymbol{\pi}_{t+1}}\right]\|_2^2$$

$$\leq \|\boldsymbol{\psi}_t\|_2^2 + 2\alpha\boldsymbol{\psi}_t^\top\left(\mathbf{r}_t(O_t) - \boldsymbol{\eta}_t(O_t) + \mathbf{A}(O_t)\mathbf{Q}_t - \bar{\mathbf{A}}^{\boldsymbol{\pi}_t,\mathbf{P}_t}\boldsymbol{\psi}_t\right) + 2\alpha\boldsymbol{\psi}_t^\top\bar{\mathbf{A}}^{\boldsymbol{\pi}_t,\mathbf{P}_t}\boldsymbol{\psi}_t$$
$$\quad + 2\boldsymbol{\psi}_t^\top\Pi_E\left[\mathbf{Q}_t^{\boldsymbol{\pi}_t} - \mathbf{Q}_{t+1}^{\boldsymbol{\pi}_{t+1}}\right] + 2\alpha^2\|\mathbf{r}_t(O_t) - \boldsymbol{\eta}_t(O_t) + \mathbf{A}(O_t)\mathbf{Q}_t\|_2^2 + 2\|\Pi_E\left[\mathbf{Q}_t^{\boldsymbol{\pi}_t} - \mathbf{Q}_{t+1}^{\boldsymbol{\pi}_{t+1}}\right]\|_2^2$$

$$\leq \|\boldsymbol{\psi}_t\|_2^2 + 2\alpha\boldsymbol{\psi}_t^\top\left(\mathbf{r}_t(O_t) - \boldsymbol{\eta}_t(O_t) + \mathbf{A}(O_t)\mathbf{Q}_t^{\boldsymbol{\pi}_t}\right) + 2\alpha\boldsymbol{\psi}_t^\top\left(\mathbf{A}(O_t) - \bar{\mathbf{A}}^{\boldsymbol{\pi}_t,\mathbf{P}_t}\right)\boldsymbol{\psi}_t + 2\alpha\boldsymbol{\psi}_t^\top\bar{\mathbf{A}}^{\boldsymbol{\pi}_t,\mathbf{P}_t}\boldsymbol{\psi}_t$$
$$\quad + 2\boldsymbol{\psi}_t^\top\Pi_E\left[\mathbf{Q}_t^{\boldsymbol{\pi}_t} - \mathbf{Q}_{t+1}^{\boldsymbol{\pi}_{t+1}}\right] + 2\alpha^2\|\mathbf{r}_t(O_t) - \boldsymbol{\eta}_t(O_t) + \mathbf{A}(O_t)\mathbf{Q}_t\|_2^2 + 2\|\Pi_E\left[\mathbf{Q}_t^{\boldsymbol{\pi}_t} - \mathbf{Q}_{t+1}^{\boldsymbol{\pi}_{t+1}}\right]\|_2^2$$

$$\leq \|\boldsymbol{\psi}_t\|_2^2 + 2\alpha\Gamma(\boldsymbol{\pi}_t,\mathbf{P}_t,\mathbf{r}_t,\boldsymbol{\psi}_t,O_t) + 2\alpha\boldsymbol{\psi}_t^\top\left(\mathbf{J}_t^{\boldsymbol{\pi}_t}(O_t) - \eta_t(O_t)\right) + 2\alpha\boldsymbol{\psi}_t^\top\bar{\mathbf{A}}^{\boldsymbol{\pi}_t,\mathbf{P}_t}\boldsymbol{\psi}_t$$
$$\quad + 2\boldsymbol{\psi}_t^\top\Pi_E\left[\mathbf{Q}_t^{\boldsymbol{\pi}_t} - \mathbf{Q}_{t+1}^{\boldsymbol{\pi}_{t+1}}\right] + 2\alpha^2\|\mathbf{r}_t(O_t) - \boldsymbol{\eta}_t(O_t) + \mathbf{A}(O_t)\mathbf{Q}_t\|_2^2 + 2\|\Pi_E\left[\mathbf{Q}_t^{\boldsymbol{\pi}_t} - \mathbf{Q}_{t+1}^{\boldsymbol{\pi}_{t+1}}\right]\|_2^2$$

$$\overset{(a)}{\leq} \|\boldsymbol{\psi}_t\|_2^2 + 2\alpha\Gamma(\boldsymbol{\pi}_t,\mathbf{P}_t,\mathbf{r}_t,\boldsymbol{\psi}_t,O_t) + 2\alpha\|\boldsymbol{\psi}_t\|_2\|\mathbf{J}_t^{\boldsymbol{\pi}_t}(O_t) - \eta_t(O_t)\|_2 + 2\alpha\boldsymbol{\psi}_t^\top\bar{\mathbf{A}}^{\boldsymbol{\pi}_t,\mathbf{P}_t}\boldsymbol{\psi}_t$$
$$\quad + 2\|\boldsymbol{\psi}_t\|_2\|\Pi_E\left[\mathbf{Q}_t^{\boldsymbol{\pi}_t} - \mathbf{Q}_{t+1}^{\boldsymbol{\pi}_{t+1}}\right]\|_2 + 2\alpha^2\|\mathbf{r}_t(O_t) - \boldsymbol{\eta}_t(O_t) + \mathbf{A}(O_t)\mathbf{Q}_t\|_2^2 + 2\|\Pi_E\left[\mathbf{Q}_t^{\boldsymbol{\pi}_t} - \mathbf{Q}_{t+1}^{\boldsymbol{\pi}_{t+1}}\right]\|_2^2$$

$$\overset{(b)}{\leq} \|\boldsymbol{\psi}_t\|_2^2 + 2\alpha\Gamma(\boldsymbol{\pi}_t,\mathbf{P}_t,\mathbf{r}_t,\boldsymbol{\psi}_t,O_t) + 2\alpha\|\boldsymbol{\psi}_t\|_2|J_t^{\boldsymbol{\pi}_t} - \eta_t| - 2\alpha\lambda\|\boldsymbol{\psi}_t\|_2^2$$
$$\quad + 2\|\boldsymbol{\psi}_t\|_2\|\Pi_E\left[\mathbf{Q}_t^{\boldsymbol{\pi}_t} - \mathbf{Q}_{t+1}^{\boldsymbol{\pi}_{t+1}}\right]\|_2 + 2\alpha^2\|\mathbf{r}_t(O_t) - \boldsymbol{\eta}_t(O_t) + \mathbf{A}(O_t)\mathbf{Q}_t\|_2^2 + 2\|\Pi_E\left[\mathbf{Q}_t^{\boldsymbol{\pi}_t} - \mathbf{Q}_{t+1}^{\boldsymbol{\pi}_{t+1}}\right]\|_2^2$$

$$\leq (1 - 2\alpha\lambda)\|\boldsymbol{\psi}_t\|_2^2 + 2\alpha\Gamma(\boldsymbol{\pi}_t, \mathbf{P}_t, \mathbf{r}_t, \boldsymbol{\psi}_t, O_t) + 2\alpha\|\boldsymbol{\psi}_t\|_2|J_t^{\boldsymbol{\pi}_t} - \eta_t|$$
$$+ 2\|\boldsymbol{\psi}_t\|_2\|\Pi_E\left[\mathbf{Q}_t^{\boldsymbol{\pi}_t} - \mathbf{Q}_{t+1}^{\boldsymbol{\pi}_{t+1}}\right]\|_2 + 2\alpha^2(2U_R + 2U_Q)^2 + 2\|\Pi_E\left[\mathbf{Q}_t^{\boldsymbol{\pi}_t} - \mathbf{Q}_{t+1}^{\boldsymbol{\pi}_{t+1}}\right]\|_2^2$$
$$\leq (1 - 2\alpha\lambda)\|\boldsymbol{\psi}_t\|_2^2 + 2\alpha\Gamma(\boldsymbol{\pi}_t, \mathbf{P}_t, \mathbf{r}_t, \boldsymbol{\psi}_t, O_t) + 2\alpha\|\boldsymbol{\psi}_t\|_2|J_t^{\boldsymbol{\pi}_t} - \eta_t| + 2\|\boldsymbol{\psi}_t\|_2\|\Pi_E\left[\mathbf{Q}_t^{\boldsymbol{\pi}_t} - \mathbf{Q}_{t+1}^{\boldsymbol{\pi}_t}\right]\|_2$$
$$+ 2\|\boldsymbol{\psi}_t\|_2\|\Pi_E\left[\mathbf{Q}_{t+1}^{\boldsymbol{\pi}_t} - \mathbf{Q}_{t+1}^{\boldsymbol{\pi}_{t+1}}\right]\|_2 + 2\alpha^2(2U_R + 2U_Q)^2 + 2\|\Pi_E\left[\mathbf{Q}_t^{\boldsymbol{\pi}_t} - \mathbf{Q}_{t+1}^{\boldsymbol{\pi}_{t+1}}\right]\|_2^2,$$

where $(a)$ is due to Cauchy-Schwarz inequality, $(b)$ follows from $\boldsymbol{\psi}_t \in E$ and Lemma 1.

Taking expectation, rearranging the terms, setting $\tau = \tau_H = \min\{i \geq 0 | m\rho^{i-1} \leq \min\{\beta, \alpha\}\}$ and summing over time, we have

$$\sum_{t=nH+\tau_H}^{nH+H-1} \lambda\mathbb{E}\left[\|\boldsymbol{\psi}_t\|_2^2\right]$$

$$\leq \underbrace{\sum_{t=nH+\tau_H}^{nH+H-1} \frac{\mathbb{E}[\|\boldsymbol{\psi}_t\|_2^2 - \|\boldsymbol{\psi}_{t+1}\|_2^2]}{2\alpha}}_{I_1} + \underbrace{\sum_{t=nH+\tau_H}^{nH+H-1} \mathbb{E}\left[\Gamma(\boldsymbol{\pi}_t, \mathbf{P}_t, \mathbf{r}_t, \boldsymbol{\psi}_t, O_t)\right]}_{I_2} + \underbrace{\sum_{t=nH+\tau_H}^{nH+H-1} \mathbb{E}\left[|\phi_t|\|\boldsymbol{\psi}_t\|_2\right]}_{I_3}$$

$$+ \underbrace{\sum_{t=nH+\tau_H}^{nH+H-1} \frac{\mathbb{E}\left[\|\boldsymbol{\psi}_t\|_2\|\Pi_E\left[\mathbf{Q}_t^{\boldsymbol{\pi}_t} - \mathbf{Q}_{t+1}^{\boldsymbol{\pi}_t}\right]\|_2\right]}{\alpha}}_{I_4} + \underbrace{\sum_{tnH+\tau_H}^{nH+H-1} \frac{\mathbb{E}\left[\|\boldsymbol{\psi}_t\|_2\|\Pi_E\left[\mathbf{Q}_{t+1}^{\boldsymbol{\pi}_t} - \mathbf{Q}_{t+1}^{\boldsymbol{\pi}_{t+1}}\right]\|_2\right]}{\alpha}}_{I_5}$$

$$+ \alpha(2U_R + 2U_Q)^2(T - \tau_T) + \underbrace{\sum_{t=nH+\tau_H}^{nH+H-1} \frac{\mathbb{E}\left[\|\Pi_E\left[\mathbf{Q}_t^{\boldsymbol{\pi}_t} - \mathbf{Q}_{t+1}^{\boldsymbol{\pi}_{t+1}}\right]\|_2^2\right]}{\alpha}}_{I_6}. \tag{15}$$

We now bound each of the terms starting with the first term as

$$I_1 = \frac{\mathbb{E}[\|\boldsymbol{\psi}_{nH+\tau_H}\|_2^2 - \|\boldsymbol{\psi}_{nH+H}\|_2^2]}{2\alpha} \leq \frac{2U_Q^2}{\alpha}.$$

By Lemma 10, we have

$$I_2 \leq \sum_{t=nH+\tau_H}^{nH+H-1} B_3\beta(\tau_H + 1)^2 + B_4\alpha\tau_H + B_5\Delta_{R,t-nH-\tau_H+1,t} + B_6\tau_H\Delta_{P,t-nH-\tau_H+1,t}$$
$$\leq B_3\beta(\tau_H + 1)^2(H - \tau_H) + B_4\alpha\tau_H(H - \tau_H) + B_5\tau_H\Delta_{R,nH,(n+1)H} + B_6\tau_H^2\Delta_{P,nH,(n+1)H}.$$

By the Cauchy-Schwarz inequality, we have

$$I_3 \leq \sum_{t=nH+\tau_H}^{nH+H-1} \sqrt{\mathbb{E}[\phi_t^2]}\sqrt{\mathbb{E}[\|\boldsymbol{\psi}_t\|_2^2]} \leq \left(\sum_{t=nH+\tau_H}^{nH+H-1} \mathbb{E}[\phi_t^2]\right)^{1/2}\left(\sum_{t=nH+\tau_H}^{nH+H-1} \mathbb{E}[\|\boldsymbol{\psi}_t\|_2^2]\right)^{1/2},$$

where $\sum_{t=nH+\tau_H}^{nH+H-1} \mathbb{E}[\phi_t^2]$ can be further bounded using Proposition 4.

Using Lemma 8, we have

$$I_4 \leq \frac{2U_Q}{\alpha}\sum_{t=nH+\tau_H}^{nH+H-1} \|\mathbf{Q}_t^{\boldsymbol{\pi}_t} - \mathbf{Q}_{t+1}^{\boldsymbol{\pi}_t}\|_2 \leq \frac{2U_Q}{\alpha}\sum_{t=nH+\tau_H}^{nH+H-1} G_R\|\mathbf{r}_{t+1} - \mathbf{r}_t\|_\infty + G_P\|\mathbf{P}_{t+1} - \mathbf{P}_t\|_\infty$$
$$\leq \frac{2U_Q}{\alpha}(G_R\Delta_{R,nH,(n+1)H} + G_P\Delta_{P,nH,(n+1)H}).$$

Using the Cauchy-Schwarz inequality and Lemma 7, we have

$$
\begin{aligned}
I_5 &\leq \left( \sum_{t=nH+\tau_H}^{nH+H-1} \frac{\mathbb{E}[\|\Pi_E \left[ \mathbf{Q}_{t+1}^{\boldsymbol{\pi}_t} - \mathbf{Q}_{t+1}^{\boldsymbol{\pi}_{t+1}} \right] \|_2^2]}{\alpha^2} \right)^{1/2} \left( \sum_{t=nH+\tau_H}^{nH+H-1} \mathbb{E}[\|\boldsymbol{\psi}_t\|_2^2] \right)^{1/2} \\
&\leq \left( \frac{G_{\boldsymbol{\pi}}^2 B_2^2 \beta^2 H}{\alpha^2} \right)^{1/2} \left( \sum_{t=nH+\tau_H}^{nH+H-1} \mathbb{E}[\|\boldsymbol{\psi}_t\|_2^2] \right)^{1/2} .
\end{aligned}
$$

We now the final term $I_6$ as follows. For timesteps with small changes in the environment, we use Lemma 9, and for timesteps with large changes in the environment, we use a naive upper bound. Define the set of timesteps $\mathcal{T}_Q := \{t : \|\mathbf{r}_{t+1} - \mathbf{r}_t\|_\infty \leq \delta_R, \|\mathbf{P}_{t+1} - \mathbf{P}_t\|_\infty \leq \delta_P\}$.

$$
\begin{aligned}
I_6 &= \sum_{t=nH+\tau_H}^{nH+H-1} \frac{\mathbb{E} \left[ \|\Pi_E \left[ \mathbf{Q}_t^{\boldsymbol{\pi}_t} - \mathbf{Q}_{t+1}^{\boldsymbol{\pi}_{t+1}} \right] \|_2^2 \right]}{\alpha} \overset{(c)}{\leq} \sum_{t \in \mathcal{T}_Q} \frac{\mathbb{E} \left[ \|\Pi_E \left[ \mathbf{Q}_t^{\boldsymbol{\pi}_t} - \mathbf{Q}_{t+1}^{\boldsymbol{\pi}_{t+1}} \right] \|_2^2 \right]}{\alpha} + \sum_{t \notin \mathcal{T}_Q} \frac{4U_Q^2}{\alpha} \\
&\overset{(d)}{\leq} \sum_{t \in \mathcal{T}_Q} \frac{3G_R^2 \delta_R^2}{\alpha} + \frac{3G_P^2 \delta_P^2}{\alpha} + \frac{3G_{\boldsymbol{\pi}}^2 B_2^2 \beta^2}{\alpha} + \sum_{t \notin \mathcal{T}_Q} \frac{4U_Q^2}{\alpha} \\
&\overset{(e)}{\leq} \frac{3G_R^2 \delta_R^2 H}{\alpha} + \frac{3G_P^2 \delta_P^2 H}{\alpha} + \frac{3G_{\boldsymbol{\pi}}^2 B_2^2 \beta^2 H}{\alpha} + \frac{4U_Q^2 \Delta_{R,nH,(n+1)H}}{\alpha \delta_R} + \frac{4U_Q^2 \Delta_{P,nH,(n+1)H}}{\alpha \delta_P} \\
&\overset{(f)}{\leq} \frac{W_1 \Delta_{R,nH,(n+1)H}^{2/3} H^{1/3}}{\alpha} + \frac{W_2 \Delta_{P,nH,(n+1)H}^{2/3} H^{1/3}}{\alpha} + \frac{3G_{\boldsymbol{\pi}}^2 B_2^2 \beta^2 H}{\alpha}
\end{aligned} \tag{16}
$$

where $(c)$ follows from the Lemma 24, $(d)$ follows from Lemma 9 and $(e)$ is obtained by choosing $\delta_R = \left( \frac{4U_Q^2 \Delta_{R,nH,(n+1)H}}{3G_R^2 H} \right)^{1/3}$ and $\delta_P = \left( \frac{4U_Q^2 \Delta_{P,nH,(n+1)H}}{3G_P^2 H} \right)^{1/3}$ and defining $W_1 = (3G_R^2)^{1/3}(4U_Q^2)^{2/3}$, $W_2 = (3G_P^2)^{1/3}(4U_Q^2)^{2/3}$.

We substitute the bounds on $I_1, \ldots, I_6$ (using Proposition 4) into (15) and use the squaring trick from Section C.3 in [33]. The above equation is of the form, $X \leq Y + Z\sqrt{X}$. Completing the squares and rearranging, we get $X \leq 2Y + Z^2$. Hence, we get the final result as

$$
\begin{aligned}
&\sum_{t=nH+\tau_H}^{nH+H-1} \mathbb{E} \left[ \|\boldsymbol{\psi}_t\|_2^2 \right] \\
&\leq \frac{4U_Q^2}{\alpha\lambda} + \frac{2B_3 \beta (\tau_H + 1)^2 H}{\lambda} + \frac{2\alpha(B_4 + 8U_R^2 + 8U_Q^2)\tau_H H}{\lambda} + \frac{2B_5 \tau_H \Delta_{R,nH,(n+1)H}}{\lambda} \\
&\quad + \frac{2B_6 \tau_H^2 \Delta_{P,nH,(n+1)H}}{\lambda} + \frac{8U_R^2}{\gamma\lambda^2} + \frac{4B_7 \beta (\tau_H + 1)^2 H}{\lambda^2} + \frac{2D_3 \gamma \tau_H H}{\lambda^2} \\
&\quad + \frac{4B_8 (\tau_H + 1)^2 \Delta_{P,nH,(n+1)H}}{\lambda^2} + \frac{2D_4 \beta^2 H}{\gamma^2 \lambda^2} + \frac{8W_3 \Delta_{R,nH,(n+1)H}^{2/3} H^{1/3}}{\gamma\lambda^2} \\
&\quad + \frac{8W_4 \Delta_{P,nH,(n+1)H}^{2/3} H^{1/3}}{\gamma\lambda^2} + \frac{4U_Q G_R \Delta_{R,nH,(n+1)H}}{\alpha\lambda} + \frac{4U_Q G_P \Delta_{P,nH,(n+1)H}}{\alpha\lambda} + \frac{G_{\boldsymbol{\pi}}^2 B_2^2 \beta^2 H}{\alpha^2 \lambda^2} \\
&\quad + \frac{2W_1 \Delta_{R,nH,(n+1)H}^{2/3} H^{1/3}}{\alpha\lambda} + \frac{2W_2 \Delta_{P,nH,(n+1)H}^{2/3} H^{1/3}}{\alpha\lambda} + \frac{6G_{\boldsymbol{\pi}}^2 B_2^2 \beta^2 H}{\alpha\lambda} \\
&\leq \tilde{\mathcal{O}} \left( \frac{1}{\alpha} \right) + \tilde{\mathcal{O}} (\beta H) + \tilde{\mathcal{O}} (\alpha H) + \tilde{\mathcal{O}} \left( \Delta_{R,nH,(n+1)H} \right) + \tilde{\mathcal{O}} \left( \Delta_{P,nH,(n+1)H} \right) + \tilde{\mathcal{O}} (\gamma H) + \tilde{\mathcal{O}} \left( \frac{1}{\gamma} \right) \\
&\quad + \tilde{\mathcal{O}} \left( \frac{\beta^2 H}{\gamma^2} \right) + \tilde{\mathcal{O}} \left( \frac{\Delta_{R,nH,(n+1)H}^{2/3} H^{1/3}}{\gamma} \right) + \tilde{\mathcal{O}} \left( \frac{\Delta_{P,nH,(n+1)H}^{2/3} H^{1/3}}{\gamma} \right)
\end{aligned} \tag{17}
$$

$$+ \tilde{\mathcal{O}} \left( \frac{\Delta_{R,nH,(n+1)H}^{2/3} H^{1/3}}{\alpha} \right) + \tilde{\mathcal{O}} \left( \frac{\Delta_{P,nH,(n+1)H}^{2/3} H^{1/3}}{\alpha} \right) + \tilde{\mathcal{O}} \left( \frac{\beta^2 H}{\alpha^2} \right),$$

where $\tilde{\mathcal{O}}(\cdot)$ hides constants and logarithmic terms.

□

### D.3 Average Reward Estimation

In this section, we first analyze the gap between the average rewards and the rewards accumulate by NS-NAC in Proposition 3. We then characterize the error in the average reward estimation in Proposition 4.

**Proposition 3.** *For any $n \in [N]$, if Assumption 1 is satisfied, then the following holds true*

$$\sum_{t=nH+\tau_H}^{nH+H-1} \mathbb{E}\left[ J_t^{\boldsymbol{\pi}_t} - r_t(s_t, a_t) \right] \leq D_1 \beta (\tau_H + 1)^2 (H - \tau_H) + D_2 (\tau_H + 1)^2 \Delta_{P,nH,(n+1)H}$$

*where $D_1 = L_{\boldsymbol{\pi}} B_2 + 4U_R \sqrt{|\mathcal{S}||\mathcal{A}|} B_2 + 4U_R$, $D_2 = 4U_R + L_P$ and $\Delta_{P,nH,(n+1)H} = \sum_{t=nH}^{(n+1)H} \|\mathbf{P}_{t+1} - \mathbf{P}_t\|_{\infty}$.*

*Proof.* Given time indices $t > \tau > 0$, recall the auxiliary Markov chain starting from $s_{t-\tau}$ constructed by conditioning on $s_{t-\tau}, \boldsymbol{\pi}_{t-\tau-1}, \mathbf{P}_{t-\tau}$ and rolling out by applying $\boldsymbol{\pi}_{t-\tau-1}, \mathbf{P}_{t-\tau}$ as

$$s_{t-\tau} \xrightarrow{\boldsymbol{\pi}_{t-\tau-1}} a_{t-\tau} \xrightarrow{\mathbf{P}_{t-\tau}} \tilde{s}_{t-\tau+1} \xrightarrow{\boldsymbol{\pi}_{t-\tau-1}} \tilde{a}_{t-\tau+1} \xrightarrow{\mathbf{P}_{t-\tau}} \ldots \tilde{s}_t \xrightarrow{\boldsymbol{\pi}_{t-\tau-1}} \tilde{a}_t \xrightarrow{\mathbf{P}_{t-\tau}} \tilde{s}_{t+1} \xrightarrow{\boldsymbol{\pi}_{t-\tau-1}} \tilde{a}_{t+1}.$$

Also, recall that the original Markov chain is

$$s_{t-\tau} \xrightarrow{\boldsymbol{\pi}_{t-\tau-1}} a_{t-\tau} \xrightarrow{\mathbf{P}_{t-\tau}} s_{t-\tau+1} \xrightarrow{\boldsymbol{\pi}_{t-\tau}} a_{t-\tau+1} \xrightarrow{\mathbf{P}_{t-\tau+1}} \ldots s_t \xrightarrow{\boldsymbol{\pi}_{t-1}} a_t \xrightarrow{\mathbf{P}_t} s_{t+1} \xrightarrow{\boldsymbol{\pi}_t} a_{t+1}.$$

Further, recall $J^{\boldsymbol{\pi}_{t-\tau-1}, \mathbf{P}_{t-\tau}, \mathbf{r}_t} := \sum_{s,a} d^{\boldsymbol{\pi}_{t-\tau-1}, \mathbf{P}_{t-\tau}}(s) \boldsymbol{\pi}_{t-\tau-1}(a|s) r_t(s,a)$.

We start by decomposing the term as

$$\mathbb{E}\left[ J_t^{\boldsymbol{\pi}_t} - r_t(s_t, a_t) \right] \tag{18}$$
$$= \underbrace{\mathbb{E}\left[ J_t^{\boldsymbol{\pi}_t} - J^{\boldsymbol{\pi}_{t-\tau-1}, \mathbf{P}_{t-\tau}, \mathbf{r}_t} \right]}_{I_1} + \underbrace{\mathbb{E}\left[ r_t(\tilde{s}_t, \tilde{a}_t) - r_t(s_t, a_t) \right]}_{I_2} + \underbrace{\mathbb{E}\left[ J^{\boldsymbol{\pi}_{t-\tau-1}, \mathbf{P}_{t-\tau}, \mathbf{r}_t} - r_t(\tilde{s}_t, \tilde{a}_t) \right]}_{I_3}.$$

$$\tag{19}$$

Note that $I_1$ is the difference in the average rewards between the two policies $\boldsymbol{\pi}_t, \boldsymbol{\pi}_{t-\tau-1}$ in two different environments $(\mathbf{P}_t, \mathbf{r}_t)$ and $(\mathbf{P}_{t-\tau}, \mathbf{r}_t)$ that share the same reward function. Hence, using Lemma 6 and Lemma 3 successively, we get

$$I_1 \leq \mathbb{E}\left[ L_{\boldsymbol{\pi}} \|\boldsymbol{\pi}_t - \boldsymbol{\pi}_{t-\tau-1}\|_2 + L_P \|\mathbf{P}_t - \mathbf{P}_{t-\tau}\|_{\infty} \right]$$
$$\leq \mathbb{E}\left[ L_{\boldsymbol{\pi}} \sum_{i=t-\tau}^{t} \|\boldsymbol{\pi}_i - \boldsymbol{\pi}_{i-1}\|_2 + L_P \sum_{i=t-\tau+1}^{t} \|\mathbf{P}_i - \mathbf{P}_{i-1}\|_{\infty} \right]$$
$$\leq L_{\boldsymbol{\pi}} B_2 \beta (\tau + 1) + L_P \Delta_{P,t-\tau+1,t}, \tag{20}$$

where $\Delta_{P,t-\tau+1,t} = \sum_{i=t-\tau+1}^{t} \|\mathbf{P}_i - \mathbf{P}_{i-1}\|_{\infty}$.

For $I_2$, by Lemma 23 and Lemma 12 successively, we get

$$I_2 \leq 2U_R \cdot 2d_{TV}\left( P((s_t, a_t) \in \cdot | \mathcal{F}_{t-\tau}), P((\tilde{s}_t, \tilde{a}_t) \in \cdot | \mathcal{F}_{t-\tau}) \right)$$
$$\leq 4U_R \sqrt{|\mathcal{S}||\mathcal{A}|} \mathbb{E}\left[ \sum_{i=t-\tau}^{t} \|\boldsymbol{\pi}_i - \boldsymbol{\pi}_{t-\tau-1}\|_2 \Big| \mathcal{F}_{t-\tau} \right] + 4U_R \sum_{i=t-\tau}^{t} \|\mathbf{P}_i - \mathbf{P}_{t-\tau}\|_{\infty}$$

$$\leq 4U_R\sqrt{|\mathcal{S}||\mathcal{A}|}B_2\beta(\tau+1)^2 + 4U_R\tau\Delta_{P,t-\tau+1,t}. \tag{21}$$

Finally, we bound $I_3$ using Lemma 17 as

$$I_3 \leq 4U_R m\rho^\tau. \tag{22}$$

Plugging the bounds on $I_1, I_2, I_3$ into Equation (19) and setting $\tau = \tau_H = \min\{i \geq 0 | m\rho^{i-1} \leq \min\{\beta, \alpha\}\}$,

$$\sum_{t=nH+\tau_H}^{nH+H-1} \mathbb{E}\left[J_t^{\boldsymbol{\pi}_t} - r_t(s_t, a_t)\right] \leq \sum_{t=nH+\tau_H}^{nH+H-1} L_{\boldsymbol{\pi}}B_2\beta(\tau_H+1) + L_P\Delta_{P,t-nH-\tau_H+1,t}$$
$$+ 4U_R\sqrt{|\mathcal{S}||\mathcal{A}|}B_2\beta(\tau_H+1)^2 + 4U_R\tau_H\Delta_{P,t-nH-\tau_H+1,t} + 4U_Rm\rho^{\tau_H}$$
$$\leq (L_{\boldsymbol{\pi}} + 4U_R\sqrt{|\mathcal{S}||\mathcal{A}|})B_2\beta(\tau_H+1)^2(H-\tau_H) + (4U_R + L_P)(\tau_H+1)^2\Delta_{P,nH,(n+1)H}$$
$$+ 4U_R\beta(H-\tau_H).$$

$\square$

**Proposition 4.** *For any $n \in [N]$, if Assumption 1 holds and $0 < \gamma < 1/2$, then we have the following*

$$\sum_{t=nH+\tau_H}^{nH+H-1} \mathbb{E}\left[(J_t^{\boldsymbol{\pi}_t} - \eta_t)^2\right] \leq \frac{4U_R^2}{\gamma} + 2B_7\beta(\tau_H+1)^2H + D_3\gamma\tau_H H + 2B_8(\tau_H+1)^2\Delta_{P,nH,(n+1)H}$$

$$+ \frac{D_4\beta^2 H}{\gamma^2} + \frac{4W_3\Delta_{R,nH,(n+1)H}^{2/3}H^{1/3}}{\gamma} + \frac{4W_4\Delta_{P,nH,(n+1)H}^{2/3}H^{1/3}}{\gamma}$$

*where $D_3 = 4U_R F_6 + 8U_R^2$, $D_4 = 9L_{\boldsymbol{\pi}}^2 B_2^2$, $\Delta_{R,nH,(n+1)H} = \sum_{t=nH}^{(n+1)H} \|\mathbf{r}_{t+1} - \mathbf{r}_t\|_\infty$, $\Delta_{P,nH,(n+1)H} = \sum_{t=nH}^{(n+1)H} \|\mathbf{P}_{t+1} - \mathbf{P}_t\|_\infty$, $W_3 = (3)^{1/3}(4U_R^2)^{2/3}$ and $W_4 = (3L_P^2)^{1/3}(4U_R^2)^{2/3}$.*

*Proof.* Recall that $\phi_t := \eta_t - J_t^{\boldsymbol{\pi}_t}$. Using the average reward update equation (line 8 in Algorithm 1), we have

$$\phi_{t+1}^2 = \left(\eta_t + \gamma(r_t(s_t, a_t) - \eta_t) - J_{t+1}^{\boldsymbol{\pi}_{t+1}}\right)^2$$
$$= \left(\phi_t + J_t^{\boldsymbol{\pi}_t} - J_{t+1}^{\boldsymbol{\pi}_{t+1}} + \gamma(r_t(s_t, a_t) - \eta_t)\right)^2$$
$$\leq \phi_t^2 + 2\gamma\phi_t(r_t(s_t, a_t) - \eta_t) + 2\phi_t(J_t^{\boldsymbol{\pi}_t} - J_{t+1}^{\boldsymbol{\pi}_{t+1}}) + 2(J_t^{\boldsymbol{\pi}} - J_{t+1}^{\boldsymbol{\pi}_{t+1}})^2 + 2\gamma^2(r_t(s_t, a_t) - \eta_t)^2$$
$$= (1-2\gamma)\phi_t^2 + 2\gamma\phi_t(r_t(s_t, a_t) - J_t^{\boldsymbol{\pi}_t}) + 2\phi_t(J_t^{\boldsymbol{\pi}_t} - J_{t+1}^{\boldsymbol{\pi}_{t+1}})$$
$$+ 2(J_t^{\boldsymbol{\pi}} - J_{t+1}^{\boldsymbol{\pi}_{t+1}})^2 + 2\gamma^2(r_t(s_t, a_t) - \eta_t)^2$$
$$= (1-2\gamma)\phi_t^2 + 2\gamma\Lambda(\boldsymbol{\pi}_t, \mathbf{P}_t, \mathbf{r}_t, \eta_t, O_t) + 2\phi_t(J_t^{\boldsymbol{\pi}_t} - J_{t+1}^{\boldsymbol{\pi}_{t+1}})$$
$$+ 2(J_t^{\boldsymbol{\pi}} - J_{t+1}^{\boldsymbol{\pi}_{t+1}})^2 + 2\gamma^2(r_t(s_t, a_t) - \eta_t)^2$$
$$= (1-2\gamma)\phi_t^2 + 2\gamma\Lambda(\boldsymbol{\pi}_t, \mathbf{P}_t, \mathbf{r}_t, \eta_t, O_t) + 2\phi_t(J_t^{\boldsymbol{\pi}_t} - J_{t+1}^{\boldsymbol{\pi}_t}) + 2\phi_t(J_{t+1}^{\boldsymbol{\pi}_t} - J_{t+1}^{\boldsymbol{\pi}_{t+1}})$$
$$+ 2(J_t^{\boldsymbol{\pi}} - J_{t+1}^{\boldsymbol{\pi}_{t+1}})^2 + 2\gamma^2(r_t(s_t, a_t) - \eta_t)^2.$$

Rearranging and setting $\tau = \tau_H = \min\{i \geq 0 | m\rho^{i-1} \leq \min\{\beta, \alpha\}\}$, we have

$$\sum_{t=nH+\tau_H}^{nH+H-1} \mathbb{E}[\phi_t^2] \leq \underbrace{\sum_{t=nH+\tau_H}^{nH+H-1} \frac{\mathbb{E}[\phi_t^2 - \phi_{t+1}^2]}{2\gamma}}_{I_1} + \underbrace{\sum_{t=nH+\tau_H}^{nH+H-1} \mathbb{E}[\Lambda(\boldsymbol{\pi}_t, \mathbf{P}_t, \mathbf{r}_t, \eta_t, O_t)]}_{I_2}$$

$$+ \underbrace{\sum_{t=nH+\tau_H}^{nH+H-1} \frac{\mathbb{E}[\phi_t(J_t^{\boldsymbol{\pi}_t} - J_{t+1}^{\boldsymbol{\pi}_t})]}{\gamma}}_{I_3} + \underbrace{\sum_{t=nH+\tau_H}^{nH+H-1} \frac{\mathbb{E}[\phi_t(J_{t+1}^{\boldsymbol{\pi}_t} - J_{t+1}^{\boldsymbol{\pi}_{t+1}})]}{\gamma}}_{I_4}$$

$$+ \underbrace{\sum_{t=nH+\tau_H}^{nH+H-1} \frac{\mathbb{E}[(J_t^{\boldsymbol{\pi}_t} - J_{t+1}^{\boldsymbol{\pi}_{t+1}})^2]}{\gamma}}_{I_5} + \underbrace{\sum_{t=nH+\tau_H}^{nH+H-1} \gamma\mathbb{E}[(r_t(s_t, a_t) - \eta_t)^2]}_{I_6}.$$

We now analyze each of these terms starting with the first term as

$$I_1 = \frac{\mathbb{E}[\phi_{nH+\tau_H}^2 - \phi_{nH+H}^2]}{2\gamma} \leq \frac{2U_R^2}{\gamma}.$$

By Lemma 11 and the average reward update equation, we have

$$I_2 \leq \sum_{t=nH+\tau_H}^{nH+H-1} B_7\beta(\tau_H+1)^2 + F_6|\eta_t - \eta_{t-nH-\tau_H}| + F_7\tau\Delta_{P,t-nH-\tau_H+1,t}$$

$$\leq B_7\beta(\tau_H+1)^2(H - \tau_H) + 2U_R F_6\gamma\tau_H(H - \tau_H) + B_8(\tau_H+1)^2\Delta_{P,nH,(n+1)H}.$$

By Lemma 6, we have

$$I_3 \leq \frac{2U_R}{\gamma}\left(\sum_{t=nH+\tau_H}^{nH+H-1} \|\mathbf{r}_{t+1} - \mathbf{r}_t\|_\infty + L_P\|\mathbf{P}_{t+1} - \mathbf{P}_t\|_\infty\right) \leq \frac{2U_R\Delta_{R,nH,(n+1)H}}{\gamma} + \frac{2U_R\Delta_{P,(n+1)H}}{\gamma}.$$

By Lemma 6 and Cauchy-Schwartz inequality, we have

$$I_4 \leq \left(\sum_{t=nH+\tau_H}^{nH+H-1} \mathbb{E}[\phi_t^2]\right)^{1/2}\left(\sum_{t=nH+\tau_H}^{nH+H-1} \frac{\mathbb{E}[(J_{t+1}^{\boldsymbol{\pi}_{t+1}} - J_t^{\boldsymbol{\pi}_t})^2]}{\gamma^2}\right)^{1/2}$$

$$\leq \left(\sum_{t=nH+\tau_H}^{nH+H-1} \mathbb{E}[\phi_t^2]\right)^{1/2}\left(\frac{L_{\boldsymbol{\pi}}^2 B_2^2\beta^2 H}{\gamma^2}\right)^{1/2}.$$

We now bound $I_5$ as follows. For timesteps with small changes in the environment, we use Lemma 6, and for timesteps with large changes in the environment, we use a naive upper bound. Define the set of timesteps $\mathcal{T}_J := \{t : \|\mathbf{r}_{t+1} - \mathbf{r}_t\|_\infty \leq \delta_R, \|\mathbf{P}_{t+1} - \mathbf{P}_t\|_\infty \leq \delta_P\}$.

$$I_5 = \sum_{t=nH+\tau_H}^{nH+H-1} \frac{\mathbb{E}\left[(J_{t+1}^{\boldsymbol{\pi}_{t+1}} - J_t^{\boldsymbol{\pi}_t})^2\right]}{\gamma}$$

$$\leq \sum_{t\in\mathcal{T}_J} \frac{\mathbb{E}\left[(J_{t+1}^{\boldsymbol{\pi}_{t+1}} - J_t^{\boldsymbol{\pi}_t})^2\right]}{\gamma} + \sum_{t\notin\mathcal{T}_J} \frac{4U_R^2}{\gamma}$$

$$\overset{(a)}{\leq} \sum_{t\in\mathcal{T}_J} \frac{3\delta_R^2}{\gamma} + \frac{3L_P^2\delta_P^2}{\gamma} + \frac{3L_{\boldsymbol{\pi}}^2 B_2^2\beta^2}{\gamma} + \sum_{t\notin\mathcal{T}_J} \frac{4U_R^2}{\gamma}$$

$$\overset{(b)}{\leq} \frac{3\delta_R^2 H}{\gamma} + \frac{3L_P^2\delta_P^2 H}{\gamma} + \frac{3L_{\boldsymbol{\pi}}^2 B_2^2\beta^2 H}{\gamma} + \frac{4U_R^2\Delta_{R,nH,(n+1)H}}{\gamma\delta_R} + \frac{4U_R^2\Delta_{P,nH,(n+1)H}}{\gamma\delta_P}$$

$$\leq \frac{W_3\Delta_{R,nH,(n+1)H}^{2/3} T^{1/3}}{\gamma} + \frac{W_4\Delta_{P,nH,(n+1)H}^{2/3} T^{1/3}}{\gamma} + \frac{3L_{\boldsymbol{\pi}}^2 B_2^2\beta^2 T}{\gamma} \tag{23}$$

where $(a)$ follows from Lemma 6 and $(b)$ is obtained by choosing $\delta_R = \left(\frac{4U_R^2\Delta_{R,nH,(n+1)H}}{3T}\right)^{1/3}$ and $\delta_P = \left(\frac{4U_R^2\Delta_{P,nH,(n+1)H}}{3L_P^2 T}\right)^{1/3}$ and defining $W_3 = (3)^{1/3}(4U_R^2)^{2/3}$, $W_2 = (3L_P^2)^{1/3}(4U_R^2)^{2/3}$.

For the final term, we have

$$I_6 \le 4U_R^2 \gamma (H - \tau_H).$$

Putting everything together, we have

$$\sum_{t=nH+\tau_H}^{nH+H-1} \mathbb{E}[\phi_t^2]$$

$$\le \frac{2U_R^2}{\gamma} + B_7 \beta (\tau_H + 1)^2 (H - \tau_H) + 2U_R F_6 \gamma \tau_H (H - \tau_H) + B_8 (\tau_H + 1)^2 \Delta_{P,nH,(n+1)H}$$

$$+ \frac{2U_R \Delta_{R,nH,(n+1)H}}{\gamma} + \frac{2U_R \Delta_{P,nH,(n+1)H}}{\gamma} + \left( \sum_{t=nH+\tau_H}^{nH+H-1} \mathbb{E}[\phi_t^2] \right)^{1/2} \left( \frac{L_{\boldsymbol{\pi}}^2 B_2^2 \beta^2 H}{\gamma^2} \right)^{1/2}$$

$$+ \frac{W_3 \Delta_{R,nH,(n+1)H}^{2/3} H^{1/3}}{\gamma} + \frac{W_4 \Delta_{P,nH,(n+1)H}^{2/3} H^{1/3}}{\gamma} + \frac{3L_{\boldsymbol{\pi}}^2 B_2^2 \beta^2 H}{\gamma} + 4U_R^2 \gamma (H - \tau_H).$$

Now, we use the squaring trick from Section C.3, [33]. The above equation is of the form, $X \le Y + Z\sqrt{X}$. Completing the squares and rearranging, we get $X \le 2Y + Z^2$. Hence,

$$\sum_{t=nH+\tau_H}^{nH+H-1} \mathbb{E}[\phi_t^2]$$

$$\le \frac{4U_R^2}{\gamma} + 2B_7 \beta (\tau_H + 1)^2 (H - \tau_H) + 4U_R F_6 \gamma \tau_H (H - \tau_H) + 2B_8 (\tau_H + 1)^2 \Delta_{P,nH,(n+1)H}$$

$$+ \frac{4U_R \Delta_{R,nH,(n+1)H}}{\gamma} + \frac{4U_R \Delta_{P,nH,(n+1)H}}{\gamma} + \frac{9L_{\boldsymbol{\pi}}^2 B_2^2 \beta^2 H}{\gamma^2}$$

$$+ \frac{2W_3 \Delta_{R,nH,(n+1)H}^{2/3} H^{1/3}}{\gamma} + \frac{2W_4 \Delta_{P,nH,(n+1)H}^{2/3} H^{1/3}}{\gamma} + 8U_R^2 \gamma (H - \tau_H)$$

$$\le \frac{4U_R^2}{\gamma} + 2B_7 \beta (\tau_H + 1)^2 (H - \tau_H) + 4U_R F_6 \gamma \tau_H (H - \tau_H) + 2B_8 (\tau_H + 1)^2 \Delta_{P,nH,(n+1)H}$$

$$+ \frac{9L_{\boldsymbol{\pi}}^2 B_2^2 \beta^2 H}{\gamma^2} + \frac{4W_3 \Delta_{R,nH,(n+1)H}^{2/3} H^{1/3}}{\gamma} + \frac{4W_4 \Delta_{P,nH,(n+1)H}^{2/3} H^{1/3}}{\gamma} + 8U_R^2 \gamma (H - \tau_H).$$

$\square$

### D.4 Technical Lemmas

### D.4.1 Actor

**Lemma 2.** *If Assumption 1 holds, for any $t, t' \ge 0$, we have*

$$\sum_s d^{\boldsymbol{\pi}_{t'}^{\star}, \mathbf{P}_{t'}}(s) \left[ \frac{\log Z_t(s)}{\beta} - V_t^{\boldsymbol{\pi}_t}(s) \right] \le \frac{J_{t+1}^{\boldsymbol{\pi}_{t+1}} - J_t^{\boldsymbol{\pi}_t}}{C} + \|\mathbf{Q}_t^{\boldsymbol{\pi}_t} - \mathbf{Q}_t\|_\infty + \frac{B_1 \beta}{C}$$

$$+ \frac{\|\mathbf{r}_{t+1} - \mathbf{r}_t\|_\infty}{C} + \frac{L_P \|\mathbf{P}_{t+1} - \mathbf{P}_t\|_\infty}{C}$$

*where $Z_t(s) = \sum_{a' \in \mathcal{A}} \pi_t(a'|s) \exp(\beta Q_t(s, a'))$, $C$ is defined in Assumption 1 and other constants in Section C.*

*Proof.* We have

$$J_{t+1}^{\boldsymbol{\pi}_{t+1}} - J_t^{\boldsymbol{\pi}_t}$$

$$= J_{t+1}^{\boldsymbol{\pi}_{t+1}} - J_t^{\boldsymbol{\pi}_{t+1}} + J_t^{\boldsymbol{\pi}_{t+1}} - J_t^{\boldsymbol{\pi}_t}$$

$$\overset{(a)}{=} J_{t+1}^{\boldsymbol{\pi}_{t+1}} - J_t^{\boldsymbol{\pi}_{t+1}} + \sum_{s,a} d^{\boldsymbol{\pi}_{t+1},\mathbf{P}_t}(s)\pi_{t+1}(a|s)\left[Q_t^{\pi_t}(s,a) - V_t^{\pi_t}(s) + Q_t(s,a) - Q_t(s,a)\right]$$

$$\overset{(b)}{=} J_{t+1}^{\boldsymbol{\pi}_{t+1}} - J_t^{\boldsymbol{\pi}_{t+1}} + \sum_{s,a} d^{\boldsymbol{\pi}_{t+1},\mathbf{P}_t}(s)\pi_{t+1}(a|s)\left[Q_t^{\pi_t}(s,a) - V_t^{\pi_t}(s)\right.$$

$$\left. + \frac{\log Z_t(s)}{\beta} + \frac{1}{\beta}\log\frac{\pi_{t+1}(a|s)}{\pi}_t(a|s) - Q_t(s,a)\right]$$

$$\overset{(c)}{\geq} J_{t+1}^{\boldsymbol{\pi}_{t+1}} - J_t^{\boldsymbol{\pi}_{t+1}} + \sum_{s,a} d^{\boldsymbol{\pi}_{t+1},\mathbf{P}_t}(s)\pi_{t+1}(a|s)\left[Q_t^{\pi_t}(s,a) - V_t^{\pi_t}(s) + \frac{\log Z_t(s)}{\beta} - Q_t(s,a)\right]$$

$$\geq J_{t+1}^{\boldsymbol{\pi}_{t+1}} - J_t^{\boldsymbol{\pi}_{t+1}} + \underbrace{\sum_{s,a} d^{\boldsymbol{\pi}_{t+1},\mathbf{P}_t}(s)\pi_t(a|s)\left[\frac{\log Z_t(s)}{\beta} - Q_t(s,a)\right]}_{I_1}$$

$$+ \underbrace{\sum_{s,a} d^{\boldsymbol{\pi}_{t+1},\mathbf{P}_t}(s)(\pi_{t+1}(a|s) - \pi_t(a|s))\left[Q_t^{\pi_t}(s,a) - Q_t(s,a)\right]}_{I_2} \tag{24}$$

where $(a)$ follows from Lemma 4, $(b)$ follows from the actor update (line 10 in Algorithm 1), and $(c)$ is due to the non-negativity of KL-Divergence.

Next, we bound the last two terms in (24). Under Assumption 1, we have

$$I_1 = \sum_{s,a} d^{\boldsymbol{\pi}_{t'}^\star,\mathbf{P}_{t'}}(s)\left(\frac{d^{\boldsymbol{\pi}_{t+1},\mathbf{P}_t}(s)}{d^{\boldsymbol{\pi}_{t'}^\star,\mathbf{P}_{t'}}(s)}\right)\pi_t(a|s)\left[\frac{\log Z_t(s)}{\beta} - Q_t(s,a)\right]$$

$$\geq C\sum_{s,a} d^{\boldsymbol{\pi}_{t'}^\star,\mathbf{P}_{t'}}(s)\pi_t(a|s)\left[\frac{\log Z_t(s)}{\beta} - Q_t(s,a)\right]$$

$$\geq C\sum_{s} d^{\boldsymbol{\pi}_{t'}^\star,\mathbf{P}_{t'}}(s)\left[\frac{\log Z_t(s)}{\beta} - V_t^{\pi_t}(s)\right] + C\sum_{s,a} d^{\boldsymbol{\pi}_{t'}^\star,\mathbf{P}_{t'}}(s)\pi_t(a|s)\left[Q_t^{\pi_t}(s,a) - Q_t(s,a)\right] \tag{25}$$

Further, we have by 1-Lipschitzness of the tabular softmax policy

$$I_2 \geq -2U_Q\sum_{s,a} d^{\boldsymbol{\pi}_{t+1},\mathbf{P}_t}(s)(\pi_{t+1}(a|s) - \pi_t(a|s)) \geq -2U_Q \cdot \beta U_Q\sqrt{|\mathcal{A}|} \geq -B_1\beta. \tag{26}$$

Plugging the bounds from (25) and (26) into (24) and rearranging, we have

$$\sum_{s} d^{\boldsymbol{\pi}_{t'}^\star,\mathbf{P}_{t'}}(s)\left[\frac{\log Z_t(s)}{\beta} - V_t^{\pi_t}(s)\right]$$

$$\leq \frac{J_{t+1}^{\boldsymbol{\pi}_{t+1}} - J_t^{\boldsymbol{\pi}_t}}{C} + \sum_{s,a} d^{\boldsymbol{\pi}_{t'}^\star,\mathbf{P}_{t'}}(s)\pi_t(a|s)\left[Q_t(s,a) - Q_t^{\pi_t}(s,a)\right] + \frac{B_1\beta}{C} + \frac{J_t^{\boldsymbol{\pi}_{t+1}} - J_{t+1}^{\boldsymbol{\pi}_{t+1}}}{C}$$

$$\leq \frac{J_{t+1}^{\boldsymbol{\pi}_{t+1}} - J_t^{\boldsymbol{\pi}_t}}{C} + \|\mathbf{Q}_t^{\boldsymbol{\pi}_t} - \mathbf{Q}_t\|_\infty + \frac{B_1\beta}{C} + \frac{\|\mathbf{r}_{t+1} - \mathbf{r}_t\|_\infty}{C} + \frac{L_P\|\mathbf{P}_{t+1} - \mathbf{P}_t\|_\infty}{C}$$

where the last inequality follows from Lemma 6. $\qquad\square$

**Lemma 3.** *For $t \geq 0$, policy $\boldsymbol{\pi}_t$ satisfies*

$$\|\boldsymbol{\pi}_{t+1} - \boldsymbol{\pi}_t\|_2 \leq B_2\beta$$

*where $B_2 = U_Q$.*

*Proof.* By 1-Lipschitzness of the softmax parameterization of the actor [54] and Lemma 24, we have

$$\|\boldsymbol{\pi}_{t+1} - \boldsymbol{\pi}_t\|_2 \leq \|\beta \mathbf{Q}_t\|_2 \leq \beta U_Q.$$

$\square$

**Lemma 4** (Average-Reward Performance Difference Lemma [41])**.** *The average rewards for any two policies* $\boldsymbol{\pi}, \boldsymbol{\pi}'$ *at time* $t$ *satisfy*

$$J_t^{\boldsymbol{\pi}} - J_t^{\boldsymbol{\pi}'} = \sum_{s \in \mathcal{S}} d^{\boldsymbol{\pi}, \mathbf{P}_t}(s) \sum_{a \in \mathcal{A}} \boldsymbol{\pi}(a|s) \left[ Q_t^{\boldsymbol{\pi}'}(s, a) - V_t^{\boldsymbol{\pi}'}(s) \right].$$

**Lemma 5.** *For any* $t, t' \geq 0$, *it holds that*

$$J_t^{\boldsymbol{\pi}_t^\star} - J_{t'}^{\boldsymbol{\pi}_{t'}^\star} \leq \|\mathbf{r}_t - \mathbf{r}_{t'}\|_\infty + U_Q \|\mathbf{P}_t - \mathbf{P}_{t'}\|_\infty.$$

*where* $\boldsymbol{\pi}_t^\star$ *represents the optimal policy for MDP* $\mathcal{M}_t(\mathcal{S}, \mathcal{A}, \mathbf{P}_t, \mathbf{r}_t)$.

*Proof.* Consider the linear programming formulation of an MDP $\mathcal{M}(\mathcal{S}, \mathcal{A}, \mathbf{P}, \mathbf{r})$ [55]

$$\min_{J, V(s)} J$$

$$\text{such that } J + V(s) \geq r(s, a) + \sum_{s'} P(s'|s, a) V(s') \ \forall s \in \mathcal{S}, a \in \mathcal{A}. \tag{27}$$

If the optimal solution for $\mathcal{M}_{t'}(\mathcal{S}, \mathcal{A}, \mathbf{P}_{t'}, \mathbf{r}_{t'})$ is $J_{t'}^\star, \mathbf{V}_{t'}^\star$, we have

$$J_{t'}^\star \mathbf{1} \geq \mathbf{r}_{t'} + (\mathbf{P}_{t'} - \mathbf{I}) \mathbf{V}_{t'}^\star.$$

Now for $\mathcal{M}_t(\mathcal{S}, \mathcal{A}, \mathbf{P}_t, \mathbf{r}_t)$, we know

$$\begin{aligned}
J_t^\star &\leq \|\mathbf{r}_t + (\mathbf{P}_t - \mathbf{I}) \mathbf{V}_{t'}^\star\|_\infty \\
&\leq \|\mathbf{r}_{t'} + (\mathbf{P}_{t'} - \mathbf{I}) \mathbf{V}_{t'}^\star + (\mathbf{r}_t - \mathbf{r}_{t'}) + (\mathbf{P}_t - \mathbf{P}_{t'}) \mathbf{V}_{t'}^\star\|_\infty \\
&\leq \|J_{t'}^\star \mathbf{1}\|_\infty + \|\mathbf{r}_t - \mathbf{r}_{t'}\|_\infty + \|(\mathbf{P}_t - \mathbf{P}_{t'}) \mathbf{V}_{t'}^\star\|_\infty.
\end{aligned}$$

Hence, we have

$$\begin{aligned}
J_t^\star - J_{t'}^\star &\leq \|\mathbf{r}_t - \mathbf{r}_{t'}\|_\infty + \|(\mathbf{P}_t - \mathbf{P}_{t'}) \mathbf{V}_{t'}^\star\|_\infty \\
J_t^{\boldsymbol{\pi}_t^\star} - J_{t'}^{\boldsymbol{\pi}_{t'}^\star} &\leq \|\mathbf{r}_t - \mathbf{r}_{t'}\|_\infty + U_Q \|\mathbf{P}_t - \mathbf{P}_{t'}\|_\infty.
\end{aligned}$$

$\square$

**Lemma 6.** *There exist constants* $L_{\boldsymbol{\pi}} = 4 U_R (M + 1) \sqrt{|\mathcal{S}||\mathcal{A}|}$ *and* $L_P = 4 U_R M$ *such that for all policies* $\boldsymbol{\pi}, \boldsymbol{\pi}'$ *and timesteps* $t, t'$, *it holds that*

$$J_t^{\boldsymbol{\pi}} - J_{t'}^{\boldsymbol{\pi}'} \leq L_{\boldsymbol{\pi}} \|\boldsymbol{\pi} - \boldsymbol{\pi}'\|_2 + \|\mathbf{r}_t - \mathbf{r}_{t'}\|_\infty + L_P \|\mathbf{P}_t - \mathbf{P}_{t'}\|_\infty.$$

*Proof.*

$$J_t^{\boldsymbol{\pi}} - J_{t'}^{\boldsymbol{\pi}'} = \underbrace{J_t^{\boldsymbol{\pi}} - J_t^{\boldsymbol{\pi}'}}_{T_1} + \underbrace{J_t^{\boldsymbol{\pi}'} - J_{t'}^{\boldsymbol{\pi}'}}_{T_2}, \tag{28}$$

where $T_1$ is the difference in the average rewards between two policies $\boldsymbol{\pi}, \boldsymbol{\pi}'$ under the same environments $(\mathbf{r}_t, \mathbf{P}_t)$, while $T_2$ is the difference in the average rewards with the same policy $\boldsymbol{\pi}'$, but under two different environments $(\mathbf{r}_t, \mathbf{P}_t)$ and $(\mathbf{r}_{t'}, \mathbf{P}_{t'})$.

$$T_1 = J_t^{\boldsymbol{\pi}} - J_t^{\boldsymbol{\pi}'} = \mathbb{E}_{s \sim d^{\boldsymbol{\pi}, \mathbf{P}_t}, a \sim \boldsymbol{\pi}, s' \sim d^{\boldsymbol{\pi}', \mathbf{P}_t}, a' \sim \boldsymbol{\pi}'} [r_t(s, a) - r_t(s', a')]$$

$$= 4U_R d_{TV} \left( d^{\boldsymbol{\pi}, \mathbf{P}_t} \otimes \boldsymbol{\pi}, d^{\boldsymbol{\pi}', \mathbf{P}_t} \otimes \boldsymbol{\pi}' \right)$$

$$\overset{(a)}{\leq} L_{\boldsymbol{\pi}} \|\boldsymbol{\pi} - \boldsymbol{\pi}'\|_2, \tag{29}$$

where $(a)$ follows from Lemma 22, where $\otimes$ denotes the Kronecker product. Next, we bound $T_2$.

$$T_2 = J_t^{\boldsymbol{\pi}'} - J_{t'}^{\boldsymbol{\pi}'} = \sum_{s,a} d^{\boldsymbol{\pi}', \mathbf{P}_t}(s) \boldsymbol{\pi}'(a|s) r_t(s,a) - d^{\boldsymbol{\pi}', \mathbf{P}_{t'}}(s) \boldsymbol{\pi}'(a|s) r_{t'}(s,a)$$

$$\leq \sum_{s,a} \left| d^{\boldsymbol{\pi}', \mathbf{P}_t}(s) \boldsymbol{\pi}'(a|s) r_t(s,a) - d^{\boldsymbol{\pi}', \mathbf{P}_t}(s) \boldsymbol{\pi}'(a|s) r_{t'}(s,a) \right|$$

$$+ \sum_{s,a} \left| d^{\boldsymbol{\pi}', \mathbf{P}_t}(s) \boldsymbol{\pi}'(a|s) r_{t'}(s,a) - d^{\boldsymbol{\pi}', \mathbf{P}_{t'}}(s) \boldsymbol{\pi}'(a|s) r_{t'}(s,a) \right|$$

$$\leq \|\mathbf{r}_t - \mathbf{r}_{t'}\|_\infty + 4U_R d_{TV}(d^{\boldsymbol{\pi}', \mathbf{P}_t} \otimes \boldsymbol{\pi}', d^{\boldsymbol{\pi}', \mathbf{P}_{t'}} \otimes \boldsymbol{\pi}')$$

$$\overset{(b)}{\leq} \|\mathbf{r}_t - \mathbf{r}_{t'}\|_\infty + L_P \|\mathbf{P}_t - \mathbf{P}_{t'}\|_\infty \tag{30}$$

where $(b)$ also follows from Lemma 22. Substituting the bounds from (29) and (30) into (28), we get the result. $\qquad\square$

### D.4.2 Critic

**Lemma 7.** *For any policies $\boldsymbol{\pi}, \boldsymbol{\pi}'$, we have*

$$\|\mathbf{Q}_t^{\boldsymbol{\pi}} - \mathbf{Q}_t^{\boldsymbol{\pi}'}\|_2 \leq G_{\boldsymbol{\pi}} \|\boldsymbol{\pi} - \boldsymbol{\pi}'\|_2$$

*where $G_{\boldsymbol{\pi}} = 2U_Q \sqrt{|\mathcal{S}||\mathcal{A}|}$.*

*Proof.*

$$Q_t^{\boldsymbol{\pi}}(s,a) \overset{(a)}{=} r_t(s,a) - J_t^{\boldsymbol{\pi}} + \mathbb{E}_{s' \sim P_t(\cdot|s,a)} \left[ V_t^{\boldsymbol{\pi}}(s') \right]$$

$$\Rightarrow \frac{\partial Q_t^{\boldsymbol{\pi}}(s,a)}{\partial \boldsymbol{\pi}} = \frac{-\partial J_t^{\boldsymbol{\pi}}}{\partial \boldsymbol{\pi}} + \sum_{s' \in \mathcal{S}} P_t(s'|s,a) \frac{\partial V_t^{\boldsymbol{\pi}}(s')}{\partial \boldsymbol{\pi}}$$

$$\left\| \frac{\partial Q_t^{\boldsymbol{\pi}}(s,a)}{\partial \boldsymbol{\pi}} \right\|_2 \leq 2 \left\| \frac{\partial J_t^{\boldsymbol{\pi}}}{\partial \boldsymbol{\pi}} \right\|_2 \tag{31}$$

$$\left\| \frac{\partial Q_t^{\boldsymbol{\pi}}(s,a)}{\partial \boldsymbol{\pi}} \right\|_2 \overset{(b)}{\leq} 2 \left\| d^{\boldsymbol{\pi}, \mathbf{P}_t}(s) Q_t^{\boldsymbol{\pi}}(s,a) \right\|_2 \leq 2U_Q \tag{32}$$

It follows from mean-value theorem that

$$|Q_t^{\boldsymbol{\pi}}(s,a) - Q_t^{\boldsymbol{\pi}'}(s,a)| \leq 2U_Q \|\boldsymbol{\pi} - \boldsymbol{\pi}'\|_2, \text{ for all } s, a$$

$$\Rightarrow \|\mathbf{Q}_t^{\boldsymbol{\pi}} - \mathbf{Q}_t^{\boldsymbol{\pi}'}\|_2 \leq G_{\boldsymbol{\pi}} \|\boldsymbol{\pi} - \boldsymbol{\pi}'\|_2,$$

where $(a)$ is by using the Bellman equation, and $(b)$ follows from Policy Gradient Theorem [20] and Lemma 24. $\qquad\square$

**Lemma 8.** *For any timesteps $t, t' \geq 0$, we have*

$$\|\Pi_E \left[ \mathbf{Q}_t^{\boldsymbol{\pi}} - \mathbf{Q}_{t'}^{\boldsymbol{\pi}} \right] \|_2 \leq G_R \|\mathbf{r}_t - \mathbf{r}_{t'}\|_\infty + G_P \|\mathbf{P}_t - \mathbf{P}_{t'}\|_\infty$$

*where $G_R = 2\lambda^{-1} \sqrt{|\mathcal{S}||\mathcal{A}|}$ and $G_P = (\lambda^{-1} L_P + 4U_R \lambda^{-1} M + 4U_R \lambda^{-2}(M+1)) \sqrt{|\mathcal{S}||\mathcal{A}|}$.*

*Proof.* Recall the diagonal matrix $D^{\boldsymbol{\pi}, \mathbf{P}_t} = diag\left( d^{\boldsymbol{\pi}, \mathbf{P}_t}(s) \pi(a|s) \right)$, where $d^{\boldsymbol{\pi}, \mathbf{P}_t}(\cdot)$ denotes the stationary distribution induced over the states, while $\mathbf{1}$ denotes the all ones vector. $E$ denotes the subspace orthogonal to the all ones vector. Pseudo-inverse of a matrix is represented by $\mathbf{X}^\dagger$. Now, we have

$$\|\Pi_E \left[ \mathbf{Q}_t^{\boldsymbol{\pi}} - \mathbf{Q}_{t'}^{\boldsymbol{\pi}} \right] \|_2 \overset{(a)}{\leq} \|(\bar{\mathbf{A}}^{\boldsymbol{\pi}, \mathbf{P}_t})^\dagger D^{\boldsymbol{\pi}, \mathbf{P}_t}(J_t^{\boldsymbol{\pi}} \mathbf{1} - \mathbf{r}_t) - (\bar{\mathbf{A}}^{\boldsymbol{\pi}, \mathbf{P}_{t'}})^\dagger D^{\boldsymbol{\pi}, \mathbf{P}_{t'}}(J_{t'}^{\boldsymbol{\pi}} \mathbf{1} - \mathbf{r}_{t'})\|_2$$

$$\leq \|(\bar{\mathbf{A}}^{\boldsymbol{\pi},\mathbf{P}_t})^{\dagger} D^{\boldsymbol{\pi},\mathbf{P}_t}(J_t^{\boldsymbol{\pi}}\mathbf{1} - \mathbf{r}_t) - (\bar{\mathbf{A}}^{\boldsymbol{\pi},\mathbf{P}_t})^{\dagger} D^{\boldsymbol{\pi},\mathbf{P}_{t'}}(J_{t'}^{\boldsymbol{\pi}}\mathbf{1} - \mathbf{r}_{t'})\|_2$$

$$+ \|(\bar{\mathbf{A}}^{\boldsymbol{\pi},\mathbf{P}_t})^{\dagger} D^{\boldsymbol{\pi},\mathbf{P}_{t'}}(J_{t'}^{\boldsymbol{\pi}}\mathbf{1} - \mathbf{r}_{t'}) - (\bar{\mathbf{A}}^{\boldsymbol{\pi},\mathbf{P}_{t'}})^{\dagger} D^{\boldsymbol{\pi},\mathbf{P}_{t'}}(J_{t'}^{\boldsymbol{\pi}}\mathbf{1} - \mathbf{r}_{t'})\|_2$$

$$\leq \|(\bar{\mathbf{A}}^{\boldsymbol{\pi},\mathbf{P}_t})^{\dagger}\|_2 \left( \|D^{\boldsymbol{\pi},\mathbf{P}_t}J_t^{\boldsymbol{\pi}}\mathbf{1} - D^{\boldsymbol{\pi},\mathbf{P}_{t'}}J_{t'}^{\boldsymbol{\pi}}\mathbf{1}\|_2 + \|D^{\boldsymbol{\pi},\mathbf{P}_t}\mathbf{r}_t - D^{\boldsymbol{\pi},\mathbf{P}_{t'}}\mathbf{r}_{t'}\|_2 \right)$$

$$+ \|(\bar{\mathbf{A}}^{\boldsymbol{\pi},\mathbf{P}_t})^{\dagger} D^{\boldsymbol{\pi},\mathbf{P}_{t'}}(J_{t'}^{\boldsymbol{\pi}}\mathbf{1} - \mathbf{r}_{t'}) - (\bar{\mathbf{A}}^{\boldsymbol{\pi},\mathbf{P}_{t'}})^{\dagger} D^{\boldsymbol{\pi},\mathbf{P}_{t'}}(J_{t'}^{\boldsymbol{\pi}}\mathbf{1} - \mathbf{r}_{t'})\|_2$$

$$\overset{(b)}{\leq} \lambda^{-1} \left( \|D^{\boldsymbol{\pi},\mathbf{P}_t}(J_t^{\boldsymbol{\pi}} - J_{t'}^{\boldsymbol{\pi}})\mathbf{1}\|_2 + \|(D^{\boldsymbol{\pi},\mathbf{P}_t} - D^{\boldsymbol{\pi},\mathbf{P}_{t'}})J_{t'}^{\boldsymbol{\pi}}\mathbf{1}\|_2 + \|D^{\boldsymbol{\pi},\mathbf{P}_t}\mathbf{r}_t - D^{\boldsymbol{\pi},\mathbf{P}_{t'}}\mathbf{r}_{t'}\|_2 \right)$$

$$+ \|(\bar{\mathbf{A}}^{\boldsymbol{\pi},\mathbf{P}_t})^{\dagger} D^{\boldsymbol{\pi},\mathbf{P}_{t'}}(J_{t'}^{\boldsymbol{\pi}}\mathbf{1} - \mathbf{r}_{t'}) - (\bar{\mathbf{A}}^{\boldsymbol{\pi},\mathbf{P}_{t'}})^{\dagger} D^{\boldsymbol{\pi},\mathbf{P}_{t'}}(J_{t'}^{\boldsymbol{\pi}}\mathbf{1} - \mathbf{r}_{t'})\|_2$$

$$\leq \lambda^{-1} \left( \sqrt{|\mathcal{S}||\mathcal{A}|} \|D^{\boldsymbol{\pi},\mathbf{P}_t}\|_2 |J_t^{\boldsymbol{\pi}} - J_{t'}^{\boldsymbol{\pi}}| + \|D^{\boldsymbol{\pi},\mathbf{P}_t} - D^{\boldsymbol{\pi},\mathbf{P}_{t'}}\|_2 \cdot U_R\sqrt{|\mathcal{S}||\mathcal{A}|} + \|D^{\boldsymbol{\pi},\mathbf{P}_t}\mathbf{r}_t - D^{\boldsymbol{\pi},\mathbf{P}_{t'}}\mathbf{r}_{t'}\|_2 \right)$$

$$+ \|(\bar{\mathbf{A}}^{\boldsymbol{\pi},\mathbf{P}_t})^{\dagger} D^{\boldsymbol{\pi},\mathbf{P}_{t'}}(J_{t'}^{\boldsymbol{\pi}}\mathbf{1} - \mathbf{r}_{t'}) - (\bar{\mathbf{A}}^{\boldsymbol{\pi},\mathbf{P}_{t'}})^{\dagger} D^{\boldsymbol{\pi},\mathbf{P}_{t'}}(J_{t'}^{\boldsymbol{\pi}}\mathbf{1} - \mathbf{r}_{t'})\|_2$$

$$\overset{(c)}{\leq} \lambda^{-1}\sqrt{|\mathcal{S}||\mathcal{A}|} \left( \|\mathbf{r}_t - \mathbf{r}_{t'}\|_\infty + L_P\|\mathbf{P}_t - \mathbf{P}_{t'}\|_\infty + 2U_R d_{TV}(d^{\boldsymbol{\pi},\mathbf{P}_t} \otimes \boldsymbol{\pi}, d^{\boldsymbol{\pi},\mathbf{P}_{t'}} \otimes \boldsymbol{\pi}) \right)$$

$$+ \lambda^{-1}\|D^{\boldsymbol{\pi},\mathbf{P}_t}\mathbf{r}_t - D^{\boldsymbol{\pi},\mathbf{P}_{t'}}\mathbf{r}_{t'}\|_2 \tag{33}$$

$$+ \|(\bar{\mathbf{A}}^{\boldsymbol{\pi},\mathbf{P}_t})^{\dagger} D^{\boldsymbol{\pi},\mathbf{P}_{t'}}(J_{t'}^{\boldsymbol{\pi}}\mathbf{1} - \mathbf{r}_{t'}) - (\bar{\mathbf{A}}^{\boldsymbol{\pi},\mathbf{P}_{t'}})^{\dagger} D^{\boldsymbol{\pi},\mathbf{P}_{t'}}(J_{t'}^{\boldsymbol{\pi}}\mathbf{1} - \mathbf{r}_{t'})\|_2$$

$$\overset{(d)}{\leq} \lambda^{-1}\sqrt{|\mathcal{S}||\mathcal{A}|} \left( \|\mathbf{r}_t - \mathbf{r}_{t'}\|_\infty + L_P\|\mathbf{P}_t - \mathbf{P}_{t'}\|_\infty + 2U_R M\|\mathbf{P}_t - \mathbf{P}_{t'}\|_\infty \right)$$

$$+ \lambda^{-1}\|D^{\boldsymbol{\pi},\mathbf{P}_t}\mathbf{r}_t - D^{\boldsymbol{\pi},\mathbf{P}_{t'}}\mathbf{r}_{t'}\|_2 \tag{34}$$

$$+ \|(\bar{\mathbf{A}}^{\boldsymbol{\pi},\mathbf{P}_t})^{\dagger} D^{\boldsymbol{\pi},\mathbf{P}_{t'}}(J_{t'}^{\boldsymbol{\pi}}\mathbf{1} - \mathbf{r}_{t'}) - (\bar{\mathbf{A}}^{\boldsymbol{\pi},\mathbf{P}_{t'}})^{\dagger} D^{\boldsymbol{\pi},\mathbf{P}_{t'}}(J_{t'}^{\boldsymbol{\pi}}\mathbf{1} - \mathbf{r}_{t'})\|_2$$

$$\overset{(e)}{\leq} \lambda^{-1}\sqrt{|\mathcal{S}||\mathcal{A}|} \left( 2\|\mathbf{r}_t - \mathbf{r}_{t'}\|_\infty + L_P\|\mathbf{P}_t - \mathbf{P}_{t'}\|_\infty + 4U_R M\|\mathbf{P}_t - \mathbf{P}_{t'}\|_\infty \right)$$

$$+ \|(\bar{\mathbf{A}}^{\boldsymbol{\pi},\mathbf{P}_t})^{\dagger} D^{\boldsymbol{\pi},\mathbf{P}_{t'}}(J_{t'}^{\boldsymbol{\pi}}\mathbf{1} - \mathbf{r}_{t'}) - (\bar{\mathbf{A}}^{\boldsymbol{\pi},\mathbf{P}_{t'}})^{\dagger} D^{\boldsymbol{\pi},\mathbf{P}_{t'}}(J_{t'}^{\boldsymbol{\pi}}\mathbf{1} - \mathbf{r}_{t'})\|_2$$

$$\leq \lambda^{-1}\sqrt{|\mathcal{S}||\mathcal{A}|} \left( 2\|\mathbf{r}_t - \mathbf{r}_{t'}\|_\infty + L_P\|\mathbf{P}_t - \mathbf{P}_{t'}\|_\infty + 4U_R M\|\mathbf{P}_t - \mathbf{P}_{t'}\|_\infty \right)$$

$$+ \|(\bar{\mathbf{A}}^{\boldsymbol{\pi},\mathbf{P}_t})^{\dagger} - (\bar{\mathbf{A}}^{\boldsymbol{\pi},\mathbf{P}_{t'}})^{\dagger}\|_2 \cdot 2U_R$$

$$\overset{(f)}{\leq} \lambda^{-1}\sqrt{|\mathcal{S}||\mathcal{A}|} \left( 2\|\mathbf{r}_t - \mathbf{r}_{t'}\|_\infty + L_P\|\mathbf{P}_t - \mathbf{P}_{t'}\|_\infty + 4U_R M\|\mathbf{P}_t - \mathbf{P}_{t'}\|_\infty \right)$$

$$+ 2U_R\lambda^{-2}\|\bar{\mathbf{A}}^{\boldsymbol{\pi},\mathbf{P}_t} - \bar{\mathbf{A}}^{\boldsymbol{\pi},\mathbf{P}_{t'}}\|_2$$

$$\overset{(g)}{\leq} \lambda^{-1}\sqrt{|\mathcal{S}||\mathcal{A}|} \left( 2\|\mathbf{r}_t - \mathbf{r}_{t'}\|_\infty + L_P\|\mathbf{P}_t - \mathbf{P}_{t'}\|_\infty + 4U_R M\|\mathbf{P}_t - \mathbf{P}_{t'}\|_\infty \right)$$

$$+ 2U_R\lambda^{-2} \cdot 2(M+1)\sqrt{|\mathcal{S}||\mathcal{A}|}\|\mathbf{P}_t - \mathbf{P}_{t'}\|_\infty$$

$$\leq G_R\|\mathbf{r}_t - \mathbf{r}_{t'}\|_\infty + G_P\|\mathbf{P}_t - \mathbf{P}_{t'}\|_\infty$$

where $(a)$ is because $\mathbb{E}\left[\mathbf{r}(O) - \mathbf{J}(O) + \mathbf{A}(O)\mathbf{Q}^{\boldsymbol{\pi}}\right] = 0$ (see TD limiting point (3) in Section 5.1) $(b)$ is from Lemma 1, $(c)$ is by Lemma 6, $(d)$ is due to Lemma 22, $(e)$ is using the same process as the last step for the second term, $(f)$ is because $\|\mathbf{X}^{\dagger} - \mathbf{Y}^{\dagger}\|_2 \leq \|\mathbf{X}^{\dagger}(\mathbf{X} - \mathbf{Y})\mathbf{Y}^{\dagger}\|_2 \leq \|\mathbf{X}^{\dagger}\|_2\|\mathbf{X} - \mathbf{Y}\|_2\|\mathbf{Y}^{\dagger}\|_2$ and $(g)$ is by Lemma 24 and Lemma 22. $\qquad\square$

**Lemma 9.** *For any* $t \geq 0$, *we have*

$$\|\Pi_E\left[\mathbf{Q}_{t+1}^{\boldsymbol{\pi}_{t+1}} - \mathbf{Q}_t^{\boldsymbol{\pi}_t}\right]\|_2 \leq G_R\|\mathbf{r}_{t+1} - \mathbf{r}_t\|_\infty + G_P\|\mathbf{P}_{t+1} - \mathbf{P}_t\|_\infty + G_{\boldsymbol{\pi}}B_2\beta.$$

*See Section C for constants.*

*Proof.*

$$\|\Pi_E\left[\mathbf{Q}_{t+1}^{\boldsymbol{\pi}_{t+1}} - \mathbf{Q}_t^{\boldsymbol{\pi}_t}\right]\|_2 \leq \|\Pi_E\left[\mathbf{Q}_{t+1}^{\boldsymbol{\pi}_{t+1}} - \mathbf{Q}_t^{\boldsymbol{\pi}_{t+1}}\right]\|_2 + \|\Pi_E\left[\mathbf{Q}_t^{\boldsymbol{\pi}_{t+1}} - \mathbf{Q}_t^{\boldsymbol{\pi}_t}\right]\|_2$$

$$\overset{(a)}{\leq} G_R\|\mathbf{r}_{t+1} - \mathbf{r}_t\|_\infty + G_P\|\mathbf{P}_{t+1} - \mathbf{P}_t\|_\infty + G_{\boldsymbol{\pi}}\|\boldsymbol{\pi}_{t+1} - \boldsymbol{\pi}_t\|_2$$

$$\overset{(b)}{\leq} G_R\|\mathbf{r}_{t+1} - \mathbf{r}_t\|_\infty + G_P\|\mathbf{P}_{t+1} - \mathbf{P}_t\|_\infty + G_{\boldsymbol{\pi}}B_2\beta$$

where $(a)$ is by Lemma 8 and Lemma 7 and $(b)$ is from Lemma 3. $\qquad\square$

**Lemma 10.** *If Assumption 1 holds, for any $t > \tau$, we have*

$$\mathbb{E}\left[\Gamma(\boldsymbol{\pi}_t, \mathbf{P}_t, \mathbf{r}_t, \boldsymbol{\psi}_t, O_t)\right] \leq B_3 \beta (\tau+1)^2 + B_4 \alpha \tau + B_5 \Delta_{R,t-\tau+1,t} + B_6 \tau \Delta_{P,t-\tau+1,t}$$

*where $B_3 = (F_{1\boldsymbol{\pi}} + F_2 G_{\boldsymbol{\pi}} + F_3 \sqrt{|\mathcal{S}||\mathcal{A}|} + F_4) B_2$, $B_4 = F_2(2U_R + 2U_Q)$, $B_5 = F_2 G_R$ and $B_6 = F_{1\mathbf{P}} + F_2 G_P + F_3$, $\Delta_{R,t-\tau+1,t} = \sum_{i=t-\tau+1}^{t} \|\mathbf{r}_i - \mathbf{r}_{i-1}\|_\infty$ and $\Delta_{P,t-\tau+1,t} = \sum_{i=t-\tau+1}^{t} \|\mathbf{P}_i - \mathbf{P}_{i-1}\|_\infty$.*

*Proof.* Recall from Section B, the definition

$$\Gamma(\boldsymbol{\pi}, \mathbf{P}, \mathbf{r}, \boldsymbol{\psi}, O) = \boldsymbol{\psi}^\top \left(\mathbf{r}(O) - \mathbf{J}^{\boldsymbol{\pi},\mathbf{P},\mathbf{r}}(O) + \mathbf{A}(O)\mathbf{Q}^{\boldsymbol{\pi},\mathbf{P},\mathbf{r}}\right) + \boldsymbol{\psi}^\top \left(\mathbf{A}(O) - \bar{\mathbf{A}}^{\boldsymbol{\pi},\mathbf{P}}\right)\boldsymbol{\psi}.$$

We first decompose $\Gamma(\cdot)$ into the following four terms

$$\mathbb{E}\left[\Gamma(\boldsymbol{\pi}_t, \mathbf{P}_t, \mathbf{r}_t, \boldsymbol{\psi}_t, O_t)\right] \leq \underbrace{\mathbb{E}\left[\Gamma(\boldsymbol{\pi}_t, \mathbf{P}_t, \mathbf{r}_t, \boldsymbol{\psi}_t, O_t) - \Gamma(\boldsymbol{\pi}_{t-\tau-1}, \mathbf{P}_{t-\tau}, \mathbf{r}_t, \boldsymbol{\psi}_t, O_t)\right]}_{I_1}$$

$$+ \underbrace{\mathbb{E}\left[\Gamma(\boldsymbol{\pi}_{t-\tau-1}, \mathbf{P}_{t-\tau}, \mathbf{r}_t, \boldsymbol{\psi}_t, O_t) - \Gamma(\boldsymbol{\pi}_{t-\tau-1}, \mathbf{P}_{t-\tau}, \mathbf{r}_t, \boldsymbol{\psi}_{t-\tau}, O_t)\right]}_{I_2}$$

$$+ \underbrace{\mathbb{E}\left[\Gamma(\boldsymbol{\pi}_{t-\tau-1}, \mathbf{P}_{t-\tau}, \mathbf{r}_t, \boldsymbol{\psi}_{t-\tau}, O_t) - \Gamma(\boldsymbol{\pi}_{t-\tau-1}, \mathbf{P}_{t-\tau}, \mathbf{r}_t, \boldsymbol{\psi}_{t-\tau}, \tilde{O}_t)\right]}_{I_3}$$

$$+ \underbrace{\mathbb{E}\left[\Gamma(\boldsymbol{\pi}_{t-\tau-1}, \mathbf{P}_{t-\tau}, \mathbf{r}_t, \boldsymbol{\psi}_{t-\tau}, \tilde{O}_t)\right]}_{I_4}.$$

We now bound each term as follows.

$$I_1 \overset{(a)}{\leq} F_{1\boldsymbol{\pi}} \mathbb{E}\left[\|\boldsymbol{\pi}_t - \boldsymbol{\pi}_{t-\tau-1}\|_2\right] + F_{1\mathbf{P}} \|\mathbf{P}_t - \mathbf{P}_{t-\tau}\|_\infty$$

$$\leq F_{1\boldsymbol{\pi}} \mathbb{E}\left[\sum_{i=t-\tau}^{t} \|\boldsymbol{\pi}_i - \boldsymbol{\pi}_{i-1}\|_2\right] + F_{1\mathbf{P}} \sum_{i=t-\tau+1}^{t} \|\mathbf{P}_i - \mathbf{P}_{i-1}\|_\infty$$

$$\overset{(b)}{\leq} F_{1\boldsymbol{\pi}} B_2 \beta(\tau+1) + F_{1\mathbf{P}} \Delta_{P,t-\tau+1,t}$$

where $(a)$ is by Lemma 13 and $(b)$ is due to Lemma 12. For the second term, we have

$$I_2 \overset{(c)}{\leq} F_2 \mathbb{E}\left[\|\boldsymbol{\psi}_t - \boldsymbol{\psi}_{t-\tau}\|_2\right] \leq F_2 \mathbb{E}\left[\sum_{i=t-\tau+1}^{t} \|\boldsymbol{\psi}_i - \boldsymbol{\psi}_{i-1}\|_2\right]$$

$$\overset{(d)}{\leq} F_2 \left[\sum_{i=t-\tau+1}^{t} (2U_R + 2U_Q)\alpha + G_R \|\mathbf{r}_i - \mathbf{r}_{i-1}\|_\infty + G_P \|\mathbf{P}_i - \mathbf{P}_{i-1}\|_\infty + G_{\boldsymbol{\pi}} B_2 \beta\right]$$

$$\leq F_2(2U_R + 2U_Q)\alpha\tau + F_2 G_R \Delta_{R,t-\tau+1,t} + F_2 G_P \Delta_{P,t-\tau+1,t} + F_2 G_{\boldsymbol{\pi}} B_2 \beta\tau$$

where $(c)$ is by Lemma 14, $(d)$ follows from Lemma 24, $\|\mathbf{Q}_{t+1} - \mathbf{Q}_t\|_2 \leq \beta(U_R + U_Q)$ by the critic update equation (9), Lemma 9 and Lemma 12. We also define $\Delta_{R,t-\tau+1,t} = \sum_{i=t-\tau+1}^{t} \|\mathbf{r}_i - \mathbf{r}_{i-1}\|_\infty$ and $\Delta_{P,t-\tau+1,t} = \sum_{i=t-\tau+1}^{t} \|\mathbf{P}_i - \mathbf{P}_{i-1}\|_\infty$.

For the third term, we have

$$I_3 \overset{(e)}{\leq} F_3 \sqrt{|\mathcal{S}||\mathcal{A}|} \mathbb{E}\left[\sum_{i=t-\tau}^{t} \|\boldsymbol{\pi}_i - \boldsymbol{\pi}_{t-\tau-1}\|_2 \Big| \mathcal{F}_{t-\tau}\right] + F_3 \sum_{i=t-\tau}^{t} \|\mathbf{P}_i - \mathbf{P}_{t-\tau}\|_\infty$$

$$\overset{(f)}{\leq} F_3 \sqrt{|\mathcal{S}||\mathcal{A}|} B_2 \beta(\tau+1)^2 + F_3 \tau \Delta_{P,t-\tau+1,t}.$$

where $(e)$ is due to Lemma 15 and $(f)$ follows from Lemma 12. For the last term, by Lemma 16, we have

$$I_4 \leq F_4 m \rho^\tau.$$

We get the final result by putting all the four terms together. $\qquad\square$

### D.4.3 Average Reward Estimation

**Lemma 11.** *If Assumption 1 holds, for any $t > \tau$, we have*

$$\mathbb{E}[\Lambda(\boldsymbol{\pi}_t, \mathbf{P}_t, \mathbf{r}_t, \eta_t, O_t)] \leq B_7 \beta (\tau + 1)^2 + F_6 |\eta_t - \eta_{t-\tau}| + B_8 \tau \Delta_{P, t-\tau+1, t}$$

*where $B_7 = (F_5 L_{\boldsymbol{\pi}} + F_7 \sqrt{|\mathcal{S}||\mathcal{A}|} + F_8) B_2$, $B_8 = F_7 + F_5 L_P$ and $\Delta_{P, t-\tau+1, t} = \sum_{i=t-\tau+1}^{t} \|\mathbf{P}_i - \mathbf{P}_{i-1}\|_\infty$.*

*Proof.* Recall from Section B, the definition

$$\Lambda(\boldsymbol{\pi}, \mathbf{P}, \mathbf{r}, \eta, O) = (\eta - J^{\boldsymbol{\pi}, \mathbf{P}, \mathbf{r}})(r(s,a) - J^{\boldsymbol{\pi}, \mathbf{P}, \mathbf{r}})$$

We first decompose $\Lambda(\boldsymbol{\pi}_t, \mathbf{P}_t, \mathbf{r}_t, \eta_t, O_t)$ into the following four terms

$$\mathbb{E}[\Lambda(\boldsymbol{\pi}_t, \mathbf{P}_t, \mathbf{r}_t, \eta_t, O_t)] = \underbrace{\mathbb{E}[\Lambda(\boldsymbol{\pi}_t, \mathbf{P}_t, \mathbf{r}_t, \eta_t, O_t) - \Lambda(\boldsymbol{\pi}_{t-\tau-1}, \mathbf{P}_{t-\tau}, \mathbf{r}_t, \eta_t, O_t)]}_{I_1}$$

$$+ \underbrace{\mathbb{E}[\Lambda(\boldsymbol{\pi}_{t-\tau-1}, \mathbf{P}_{t-\tau}, \mathbf{r}_t, \eta_t, O_t) - \Lambda(\boldsymbol{\pi}_{t-\tau-1}, \mathbf{P}_{t-\tau}, \mathbf{r}_t, \eta_{t-\tau}, O_t)]}_{I_2}$$

$$+ \underbrace{\mathbb{E}[\Lambda(\boldsymbol{\pi}_{t-\tau-1}, \mathbf{P}_{t-\tau}, \mathbf{r}_t, \eta_{t-\tau}, O_t) - \Lambda(\boldsymbol{\pi}_{t-\tau-1}, \mathbf{P}_{t-\tau}, \mathbf{r}_t, \eta_{t-\tau}, \tilde{O}_t)]}_{I_3}$$

$$+ \underbrace{\mathbb{E}[\Lambda(\boldsymbol{\pi}_{t-\tau-1}, \mathbf{P}_{t-\tau}, \mathbf{r}_t, \eta_{t-\tau}, \tilde{O}_t)]}_{I_4}.$$

We now bound each term as follows.

$$I_1 \overset{(a)}{\leq} F_5 L_{\boldsymbol{\pi}} \mathbb{E}[\|\boldsymbol{\pi}_t - \boldsymbol{\pi}_{t-\tau-1}\|_2] + F_5 L_P \|\mathbf{P}_t - \mathbf{P}_{t-\tau}\|_\infty$$

$$\leq F_5 L_{\boldsymbol{\pi}} \mathbb{E}\left[\sum_{i=t-\tau}^{t} \|\boldsymbol{\pi}_i - \boldsymbol{\pi}_{i-1}\|_2\right] + F_5 L_P \sum_{i=t-\tau+1}^{t} \|\mathbf{P}_i - \mathbf{P}_{i-1}\|_\infty$$

$$\overset{(b)}{\leq} F_5 L_{\boldsymbol{\pi}} B_2 \beta (\tau + 1) + F_5 L_P \Delta_{P, t-\tau+1, t}$$

where $(a)$ follows from Lemma 18, and $(b)$ is due to Lemma 12. For the second term $I_2$, we have

$$I_2 \overset{(c)}{\leq} F_6 |\eta_t - \eta_{t-\tau}|$$

where $(c)$ is by Lemma 19. For the third term $I_3$, we have

$$I_3 \overset{(d)}{\leq} F_7 \sqrt{|\mathcal{S}||\mathcal{A}|} \mathbb{E}\left[\sum_{i=t-\tau}^{t} \|\boldsymbol{\pi}_i - \boldsymbol{\pi}_{t-\tau-1}\|_2 \Big| \mathcal{F}_{t-\tau}\right] + F_7 \sum_{i=t-\tau}^{t} \|\mathbf{P}_i - \mathbf{P}_{t-\tau}\|_\infty$$

$$\overset{(e)}{\leq} F_7 \sqrt{|\mathcal{S}||\mathcal{A}|} B_2 \beta (\tau + 1)^2 + F_7 \Delta_{P, t-\tau+1, t}.$$

where $(d)$ is due to Lemma 20 and $(e)$ follows from Lemma 12. For the last term, by Lemma 21, we have

$$I_4 \leq F_8 m \rho^\tau.$$

We get the final result by putting all the four terms together. $\qquad \square$

### D.5 Auxiliary Lemmas

### D.5.1 Actor

**Lemma 12.** *For any timesteps $t > \tau > 0$, the policies generated by Algorithm 1 satisfy*

$$\sum_{i=t-\tau}^{t} \|\boldsymbol{\pi}_i - \boldsymbol{\pi}_{t-\tau-1}\|_2 \leq B_2 \beta (\tau + 1)^2$$

*and reward and transition probability matrices satisfy*

$$\sum_{i=t-\tau}^{t} \|\mathbf{r}_i - \mathbf{r}_{t-\tau}\|_\infty \leq \tau \sum_{i=t-\tau+1}^{t} \|\mathbf{r}_i - \mathbf{r}_{i-1}\|_\infty$$

$$\sum_{i=t-\tau}^{t} \|\mathbf{P}_i - \mathbf{P}_{t-\tau}\|_\infty \leq \tau \sum_{i=t-\tau+1}^{t} \|\mathbf{P}_i - \mathbf{P}_{i-1}\|_\infty.$$

*Proof.* By triangle inequality, we have

$$\sum_{i=t-\tau}^{t} \|\boldsymbol{\pi}_i - \boldsymbol{\pi}_{t-\tau-1}\|_2 \leq \sum_{i=t-\tau}^{t} \|\sum_{j=t-\tau}^{i} \boldsymbol{\pi}_j - \boldsymbol{\pi}_{j-1}\|_2$$

$$\leq \sum_{i=t-\tau}^{t} \sum_{j=t-\tau}^{i} \|\boldsymbol{\pi}_j - \boldsymbol{\pi}_{j-1}\|_2$$

$$\overset{(a)}{\leq} B_2 \beta (\tau+1)^2$$

where $(a)$ is by Lemma 3. The rest follow similarly using triangle inequality. $\qquad\square$

### D.5.2 Critic

**Lemma 13.** *For any $\boldsymbol{\pi}, \boldsymbol{\pi}', \mathbf{P}, \mathbf{P}', \mathbf{r}, \boldsymbol{\psi}$ and $O = (s, a, s', a')$, we have*

$$|\Gamma(\boldsymbol{\pi}, \mathbf{P}, \mathbf{r}, \boldsymbol{\psi}, O) - \Gamma(\boldsymbol{\pi}', \mathbf{P}', \mathbf{r}, \boldsymbol{\psi}, O)| \leq F_{1\boldsymbol{\pi}} \|\boldsymbol{\pi} - \boldsymbol{\pi}'\|_2 + F_{1\mathbf{P}} \|\mathbf{P} - \mathbf{P}'\|_\infty$$

*where $F_{1\boldsymbol{\pi}} = 2U_Q L_{\boldsymbol{\pi}} + 4U_Q G_{\boldsymbol{\pi}} + 8U_Q^2 (M+2)|\mathcal{S}||\mathcal{A}|$, $F_{1\mathbf{P}} = 2U_Q L_P + 4U_Q G_P + 8U_Q^2(M+1)\sqrt{|\mathcal{S}||\mathcal{A}|}$.*

*Proof.*
$$|\Gamma(\boldsymbol{\pi}, \mathbf{P}, \mathbf{r}, \boldsymbol{\psi}, O) - \Gamma(\boldsymbol{\pi}', \mathbf{P}', \mathbf{r}, \boldsymbol{\psi}, O)|$$

$$= |\boldsymbol{\psi}^\top (\mathbf{J}^{\boldsymbol{\pi}', \mathbf{P}', \mathbf{r}}(O) - \mathbf{J}^{\boldsymbol{\pi}, \mathbf{P}, \mathbf{r}}(O)) + \boldsymbol{\psi}^\top \mathbf{A}(O) \left( \mathbf{Q}^{\boldsymbol{\pi}, \mathbf{P}, \mathbf{r}} - \mathbf{Q}^{\boldsymbol{\pi}', \mathbf{P}', \mathbf{r}} \right) + \boldsymbol{\psi}^\top \left( \bar{\mathbf{A}}^{\boldsymbol{\pi}', \mathbf{P}'} - \bar{\mathbf{A}}^{\boldsymbol{\pi}, \mathbf{P}} \right) \boldsymbol{\psi}|$$

$$\overset{(a)}{\leq} \|\boldsymbol{\psi}\|_\infty |J^{\boldsymbol{\pi}', \mathbf{P}', \mathbf{r}} - J^{\boldsymbol{\pi}, \mathbf{P}, \mathbf{r}}| + \|\boldsymbol{\psi}\|_2 \|\mathbf{A}(O)\|_2 \left\| \mathbf{Q}^{\boldsymbol{\pi}, \mathbf{P}, \mathbf{r}} - \mathbf{Q}^{\boldsymbol{\pi}', \mathbf{P}', \mathbf{r}} \right\|_2$$
$$+ \|\boldsymbol{\psi}\|_\infty \left\| \bar{\mathbf{A}}^{\boldsymbol{\pi}', \mathbf{P}'} - \bar{\mathbf{A}}^{\boldsymbol{\pi}, \mathbf{P}} \right\|_\infty \|\boldsymbol{\psi}\|_1$$

$$\overset{(b)}{\leq} 2U_Q L_{\boldsymbol{\pi}} \|\boldsymbol{\pi} - \boldsymbol{\pi}'\|_2 + 2U_Q L_P \|\mathbf{P} - \mathbf{P}'\|_\infty + \|\boldsymbol{\psi}\|_2 \|\mathbf{A}(O)\|_2 \left\| \mathbf{Q}^{\boldsymbol{\pi}, \mathbf{P}, \mathbf{r}} - \mathbf{Q}^{\boldsymbol{\pi}', \mathbf{P}', \mathbf{r}} \right\|_2$$
$$+ \|\boldsymbol{\psi}\|_\infty \left\| \bar{\mathbf{A}}^{\boldsymbol{\pi}', \mathbf{P}'} - \bar{\mathbf{A}}^{\boldsymbol{\pi}, \mathbf{P}} \right\|_\infty \|\boldsymbol{\psi}\|_1$$

$$\overset{(c)}{\leq} 2U_Q L_{\boldsymbol{\pi}} \|\boldsymbol{\pi} - \boldsymbol{\pi}'\|_2 + 2U_Q L_P \|\mathbf{P} - \mathbf{P}'\|_\infty + 4U_Q \cdot G_{\boldsymbol{\pi}} \|\boldsymbol{\pi} - \boldsymbol{\pi}'\|_2 + 4U_Q G_P \|\mathbf{P} - \mathbf{P}'\|_\infty$$
$$+ \|\boldsymbol{\psi}\|_\infty \left\| \bar{\mathbf{A}}^{\boldsymbol{\pi}', \mathbf{P}'} - \bar{\mathbf{A}}^{\boldsymbol{\pi}, \mathbf{P}} \right\|_\infty \|\boldsymbol{\psi}\|_1$$

$$\overset{(d)}{\leq} 2U_Q L_{\boldsymbol{\pi}} \|\boldsymbol{\pi} - \boldsymbol{\pi}'\|_2 + 2U_Q L_P \|\mathbf{P} - \mathbf{P}'\|_\infty + 4U_Q G_{\boldsymbol{\pi}} \|\boldsymbol{\pi} - \boldsymbol{\pi}'\|_2 + 4U_Q G_P \|\mathbf{P} - \mathbf{P}'\|_\infty$$
$$+ 2U_Q \cdot 2d_{TV} \left( d^{\boldsymbol{\pi}', \mathbf{P}'} \otimes \boldsymbol{\pi}' \otimes \mathbf{P}' \otimes \boldsymbol{\pi}', d^{\boldsymbol{\pi}, \mathbf{P}} \otimes \boldsymbol{\pi} \otimes \mathbf{P} \otimes \boldsymbol{\pi} \right) \cdot 2U_Q \sqrt{|\mathcal{S}||\mathcal{A}|}$$

$$\overset{(e)}{\leq} 2U_Q L_{\boldsymbol{\pi}} \|\boldsymbol{\pi} - \boldsymbol{\pi}'\|_2 + 2U_Q L_P \|\mathbf{P} - \mathbf{P}'\|_\infty + 4U_Q G_{\boldsymbol{\pi}} \|\boldsymbol{\pi} - \boldsymbol{\pi}'\|_2 + 4U_Q G_P \|\mathbf{P} - \mathbf{P}'\|_\infty$$
$$+ 8U_Q^2 (M+2)|\mathcal{S}||\mathcal{A}| \|\boldsymbol{\pi} - \boldsymbol{\pi}'\|_2 + 8U_Q^2 (M+1)\sqrt{|\mathcal{S}||\mathcal{A}|} \|\boldsymbol{\pi} - \boldsymbol{\pi}'\|_\infty$$

where $(a)$ follows from Holder's inequality; $(b)$ is due to Lemma 6; $(c)$ is by Lemma 9 and Lemma 24 ($\|\mathbf{A}(O)\|_1 \leq 1$); $(d)$ is by Lemma 24 and $(e)$ uses Lemma 22. $\qquad\square$

**Lemma 14.** *For any $\boldsymbol{\pi}, \mathbf{P}, \mathbf{r}, \boldsymbol{\psi}, \boldsymbol{\psi}'$ and $O = (s, a, s', a')$, we have*

$$|\Gamma(\boldsymbol{\pi}, \mathbf{P}, \mathbf{r}, \boldsymbol{\psi}, O) - \Gamma(\boldsymbol{\pi}, \mathbf{P}, \mathbf{r}, \boldsymbol{\psi}', O)| \leq F_2 \|\boldsymbol{\psi} - \boldsymbol{\psi}'\|_2$$

*where $F_2 = 2U_R + 18U_Q$.*

*Proof.*

$$\begin{aligned}
&|\Gamma(\boldsymbol{\pi}, \mathbf{P}, \mathbf{r}, \boldsymbol{\psi}, O) - \Gamma(\boldsymbol{\pi}, \mathbf{P}, \mathbf{r}, \boldsymbol{\psi}', O)| \\
&\leq \left( \|\mathbf{r}(O)\|_2 + \|\mathbf{J}^{\boldsymbol{\pi}, \mathbf{P}, \mathbf{r}}(O)\|_2 + \|\mathbf{A}(O)\|_2 \|\mathbf{Q}^{\boldsymbol{\pi}, \mathbf{P}, \mathbf{r}}\|_2 \right) \|\boldsymbol{\psi} - \boldsymbol{\psi}'\|_2 \\
&\quad + \|\mathbf{A}(O) - \bar{\mathbf{A}}^{\boldsymbol{\pi}, \mathbf{P}}\|_2 \|\boldsymbol{\psi} - \boldsymbol{\psi}'\|_2 (\|\boldsymbol{\psi}\|_2 + \|\boldsymbol{\psi}'\|_2) \\
&\leq (2U_R + 18U_Q) \|\boldsymbol{\psi} - \boldsymbol{\psi}'\|_2.
\end{aligned}$$

$\square$

**Lemma 15.** *Consider an observation from the original Markov chain by $O_t = (s_t, a_t, s_{t+1}, a_{t+1})$ and auxiliary Markov chain by $\tilde{O}_t = (\tilde{s}_t, \tilde{a}_t, \tilde{s}_{t+1}, \tilde{a}_{t+1})$. Conditioned on $\mathcal{F}_{t-\tau} = \{s_{t-\tau}, \boldsymbol{\pi}_{t-\tau-1}, \mathbf{P}_{t-\tau}\}$, we have*

$$\mathbb{E}\left[\Gamma(\boldsymbol{\pi}_{t-\tau-1}, \mathbf{P}_{t-\tau}, \mathbf{r}_t, \boldsymbol{\psi}_{t-\tau}, O_t) - \Gamma(\boldsymbol{\pi}_{t-\tau-1}, \mathbf{P}_{t-\tau}, \mathbf{r}_t, \boldsymbol{\psi}_{t-\tau}, \tilde{O}_t) \big| \mathcal{F}_{t-\tau}\right]$$

$$\leq F_3 \sqrt{|\mathcal{S}||\mathcal{A}|} \mathbb{E}\left[\sum_{i=t-\tau}^{t} \|\boldsymbol{\pi}_i - \boldsymbol{\pi}_{t-\tau-1}\|_2 \Big| \mathcal{F}_{t-\tau}\right] + F_3 \sum_{i=t-\tau}^{t} \|\mathbf{P}_i - \mathbf{P}_{t-\tau}\|_\infty$$

*where $F_3 = 16U_R U_Q + 24U_Q^2 \sqrt{|\mathcal{S}||\mathcal{A}|}$.*

*Proof.* Consider the original and auxiliary Markov chains whose construction is described in Section B.

$$\begin{aligned}
&\mathbb{E}\left[\Gamma(\boldsymbol{\pi}_{t-\tau-1}, \mathbf{P}_{t-\tau}, \mathbf{r}_t, \boldsymbol{\psi}_{t-\tau}, O_t) - \Gamma(\boldsymbol{\pi}_{t-\tau-1}, \mathbf{P}_{t-\tau}, \mathbf{r}_t, \boldsymbol{\psi}_{t-\tau}, \tilde{O}_t) \big| \mathcal{F}_{t-\tau}\right] \\
&= \boldsymbol{\psi}_{t-\tau}^\top \mathbb{E}\left[\mathbf{r}_t(O_t) - \mathbf{r}_t(\tilde{O}_t) + \mathbf{J}^{\boldsymbol{\pi}_{t-\tau-1}, \mathbf{P}_{t-\tau}, \mathbf{r}_t}(\tilde{O}_t) - \mathbf{J}^{\boldsymbol{\pi}_{t-\tau}, \mathbf{P}_{t-\tau}, \mathbf{r}_t}(O_t) \big| \mathcal{F}_{t-\tau}\right] \\
&\quad + \boldsymbol{\psi}_{t-\tau}^\top \mathbb{E}\left[\left(\mathbf{A}(O_t) - \mathbf{A}(\tilde{O}_t)\right) \mathbf{Q}^{\boldsymbol{\pi}_{t-\tau-1}, \mathbf{P}_{t-\tau}, \mathbf{r}_t} \big| \mathcal{F}_{t-\tau}\right] \\
&\quad + \boldsymbol{\psi}_{t-\tau}^\top \mathbb{E}\left[\left(\mathbf{A}(O_t) - \mathbf{A}(\tilde{O}_t)\right) \big| \mathcal{F}_{t-\tau}\right] \boldsymbol{\psi}_{t-\tau} \\
&\leq \|\boldsymbol{\psi}_{t-\tau}\|_\infty \left\|\mathbb{E}\left[\mathbf{r}_t(O_t) - \mathbf{r}_t(\tilde{O}_t) + \mathbf{J}_t^{\boldsymbol{\pi}_{t-\tau-1}}(\tilde{O}_t) - \mathbf{J}_t^{\boldsymbol{\pi}_{t-\tau-1}}(O_t) \big| \mathcal{F}_{t-\tau}\right]\right\|_1 \\
&\quad + \|\boldsymbol{\psi}_{t-\tau}\|_\infty \left\|\mathbb{E}\left[\mathbf{A}(O_t) - \mathbf{A}(\tilde{O}_t) \big| \mathcal{F}_{t-\tau}\right]\right\|_1 \|\mathbf{Q}^{\boldsymbol{\pi}_{t-\tau-1}, \mathbf{P}_{t-\tau}, \mathbf{r}_t}\|_1 \\
&\quad + \|\boldsymbol{\psi}_{t-\tau}\|_\infty \left\|\mathbb{E}\left[\mathbf{A}(O_t) - \mathbf{A}(\tilde{O}_t) \big| \mathcal{F}_{t-\tau}\right]\right\|_1 \|\boldsymbol{\psi}_{t-\tau}\|_1 \\
&\leq 2U_Q \cdot 4U_R \cdot 2d_{TV}\left(P(O_t \in \cdot | \mathcal{F}_{t-\tau}), P(\tilde{O}_t \in \cdot | \mathcal{F}_{t-\tau})\right) \\
&\quad + 2U_Q \cdot 4d_{TV}\left(P(O_t \in \cdot | \mathcal{F}_{t-\tau}), P(\tilde{O}_t \in \cdot | \mathcal{F}_{t-\tau})\right) \cdot U_Q \sqrt{|\mathcal{S}||\mathcal{A}|} \\
&\quad + 2U_Q \cdot 4d_{TV}\left(P(O_t \in \cdot | \mathcal{F}_{t-\tau}), P(\tilde{O}_t \in \cdot | \mathcal{F}_{t-\tau})\right) \cdot 2U_Q \sqrt{|\mathcal{S}||\mathcal{A}|} \\
&\leq (16U_R U_Q + 24U_Q^2 \sqrt{|\mathcal{S}||\mathcal{A}|}) \left(\sqrt{|\mathcal{S}||\mathcal{A}|} \mathbb{E}\left[\sum_{i=t-\tau}^{t} \|\boldsymbol{\pi}_i - \boldsymbol{\pi}_{t-\tau-1}\|_2 \Big| \mathcal{F}_{t-\tau}\right] + \sum_{i=t-\tau}^{t} \|\mathbf{P}_i - \mathbf{P}_{t-\tau}\|_\infty\right)
\end{aligned}$$

where the last inequality is from Lemma 23. $\square$

**Lemma 16.** *Consider an observation from the original Markov chain by $O_t = (s_t, a_t, s_{t+1}, a_{t+1})$ and auxiliary Markov chain by $\tilde{O}_t = (\tilde{s}_t, \tilde{a}_t, \tilde{s}_{t+1}, \tilde{a}_{t+1})$. Conditioned on $\mathcal{F}_{t-\tau} = \{s_{t-\tau}, \boldsymbol{\pi}_{t-\tau-1}, \mathbf{P}_{t-\tau}\}$, we have*

$$\mathbb{E}\left[\Gamma(\boldsymbol{\pi}_{t-\tau-1}, \mathbf{P}_{t-\tau}, \mathbf{r}_t, \boldsymbol{\psi}_{t-\tau}, \tilde{O}_t) \big| \mathcal{F}_{t-\tau}\right] \leq F_4 m \rho^\tau$$

*where $F_4 = 8U_R U_Q + 24U_Q^2 \sqrt{|\mathcal{S}||\mathcal{A}|}$.*

*Proof.* Consider the original and auxiliary Markov chains whose construction is described in Section B. Also, consider the observation tuple $O_t' = (s_t', a_t', s_{t+1}', a_{t+1}')$ where $s_t' \sim d^{\boldsymbol{\pi}_{t-\tau-1}, \mathbf{P}_{t-\tau}}(\cdot)$, $a_t' \sim \boldsymbol{\pi}_{t-\tau-1}(\cdot | s_t')$,

$s'_{t+1} \sim \mathbf{P}_{t-\tau}(\cdot|s'_t, a'_t)$ and $a'_{t+1} \sim \boldsymbol{\pi}_{t-\tau-1}(\cdot|s'_{t+1})$. From the definition of $\Gamma(\cdot)$ and the TD limit point equation (3), it follows that

$$\mathbb{E}\left[\Gamma(\boldsymbol{\pi}_{t-\tau-1}, \mathbf{P}_{t-\tau}, \mathbf{r}_t, \boldsymbol{\psi}_{t-\tau}, O'_t)\big|\mathcal{F}_{t-\tau}\right] = 0$$

Hence, we have

$$
\begin{aligned}
\mathbb{E}&\left[\Gamma(\boldsymbol{\pi}_{t-\tau-1}, \mathbf{P}_{t-\tau}, \mathbf{r}_t, \boldsymbol{\psi}_{t-\tau}, \tilde{O}_t)\big|\mathcal{F}_{t-\tau}\right] \\
&\leq \mathbb{E}\left[\Gamma(\boldsymbol{\pi}_{t-\tau-1}, \mathbf{P}_{t-\tau}, \mathbf{r}_t, \boldsymbol{\psi}_{t-\tau}, \tilde{O}_t) - \Gamma(\boldsymbol{\pi}_{t-\tau-1}, \mathbf{P}_{t-\tau}, \mathbf{r}_t, \boldsymbol{\psi}_{t-\tau}, O'_t)\big|\mathcal{F}_{t-\tau}\right] \\
&\leq \|\boldsymbol{\psi}_{t-\tau}\|_\infty \left\|\mathbb{E}\left[\mathbf{r}_t(\tilde{O}_t) - \mathbf{J}^{\boldsymbol{\pi}_{t-\tau-1}, \mathbf{P}_{t-\tau}, \mathbf{r}_t}(\tilde{O}_t) - \mathbf{r}_t(O'_t) + \mathbf{J}^{\boldsymbol{\pi}_{t-\tau-1}, \mathbf{P}_{t-\tau}, \mathbf{r}_t}(O'_t)\big|\mathcal{F}_{t-\tau}\right]\right\|_1 \\
&\quad + \|\boldsymbol{\psi}_{t-\tau}\|_\infty \left\|\mathbb{E}\left[\left(\mathbf{A}(\tilde{O}_t) - \mathbf{A}(O'_t)\right)\mathbf{Q}^{\boldsymbol{\pi}_{t-\tau-1}, \mathbf{P}_{t-\tau}, \mathbf{r}_t}\big|\mathcal{F}_{t-\tau}\right]\right\|_1 \\
&\quad + \|\boldsymbol{\psi}_{t-\tau}\|_\infty \left\|\mathbb{E}\left[\left(\mathbf{A}(\tilde{O}_t) - \mathbf{A}(O'_t)\right)\boldsymbol{\psi}_{t-\tau}\big|\mathcal{F}_{t-\tau}\right]\right\|_1 \\
&\leq 2U_Q \cdot 4U_R \cdot 2d_{TV}\left(P(\tilde{O}_t \in \cdot|\mathcal{F}_{t-\tau}), P(O'_t \in \cdot|\mathcal{F}_{t-\tau})\right) \\
&\quad + 2U_Q \cdot 4d_{TV}\left(P(\tilde{O}_t \in \cdot|\mathcal{F}_{t-\tau}), P(O'_t \in \cdot|\mathcal{F}_{t-\tau})\right) \cdot U_Q\sqrt{|\mathcal{S}||\mathcal{A}|} \\
&\quad + 2U_Q \cdot 4d_{TV}\left(P(\tilde{O}_t \in \cdot|\mathcal{F}_{t-\tau}), P(O'_t \in \cdot|\mathcal{F}_{t-\tau})\right) \cdot 2U_Q\sqrt{|\mathcal{S}||\mathcal{A}|} \\
&= F_4 \sum_{s,a,s',a'} |P(\tilde{s}_t = s|\mathcal{F}_{t-\tau})\pi_{t-\tau-1}(a|s)P_{t-\tau}(s'|s,a)\pi_{t-\tau-1}(a'|s') \\
&\quad - P(s'_t = s|\mathcal{F}_{t-\tau})\pi_{t-\tau-1}(a|s)P_{t-\tau}(s'|s,a)\pi_{t-\tau-1}(a'|s')| \\
&= F_4 \sum_{s,a,s',a'} \pi_{t-\tau-1}(a|s)P(s'|s,a)\pi_{t-\tau-1}(a'|s')|P(\tilde{s}_t = s|\mathcal{F}_{t-\tau}) - P(s'_t = s|\mathcal{F}_{t-\tau})| \\
&= F_4 \sum_s |P(\tilde{s}_t = s|\mathcal{F}_{t-\tau}) - P(s'_t = s|\mathcal{F}_{t-\tau})| \\
&\leq F_4 m\rho^\tau
\end{aligned}
$$

where the last inequality follows from Assumption 1. $\qquad\square$

### D.5.3 Average Reward Estimation

**Lemma 17.** *Consider an observation from the original Markov chain by $O_t = (s_t, a_t, s'_t, a'_t)$ and auxiliary Markov chain by $\tilde{O}_t = (\tilde{s}_t, \tilde{a}_t, \tilde{s}_{t+1}, \tilde{a}_{t+1})$. Conditioned on $\mathcal{F}_{t-\tau} = \{s_{t-\tau}, \boldsymbol{\pi}_{t-\tau-1}, \mathbf{P}_{t-\tau}\}$, we have*

$$\mathbb{E}\left[J^{\boldsymbol{\pi}_{t-\tau-1}, \mathbf{P}_{t-\tau}, \mathbf{r}_t} - r_t(\tilde{s}_t, \tilde{a}_t)|\mathcal{F}_{t-\tau}\right] \leq 4U_R m\rho^\tau$$

*where $J^{\boldsymbol{\pi}_{t-\tau-1}, \mathbf{P}_{t-\tau}, \mathbf{r}_t} = \sum_{s,a} d^{\boldsymbol{\pi}_{t-\tau-1}, \mathbf{P}_{t-\tau}}(s)\boldsymbol{\pi}_{t-\tau-1}(a|s)r_t(s,a)$.*

*Proof.* Consider the observation tuple $O'_t = (s'_t, a'_t, s'_{t+1}, a'_{t+1})$ where $s'_t \sim d^{\boldsymbol{\pi}_{t-\tau-1}, \mathbf{P}_{t-\tau}}(\cdot)$, $a'_t \sim \boldsymbol{\pi}_{t-\tau-1}(\cdot|s'_t)$, $s'_{t+1} \sim \mathbf{P}_{t-\tau}(\cdot|s'_t, a'_t)$ and $a'_{t+1} \sim \boldsymbol{\pi}_{t-\tau-1}(\cdot|s'_{t+1})$. Then, by definition of $J^{\boldsymbol{\pi}_{t-\tau-1}, \mathbf{P}_{t-\tau}, \mathbf{r}_t}$, we have

$$\mathbb{E}\left[J^{\boldsymbol{\pi}_{t-\tau-1}, \mathbf{P}_{t-\tau}, \mathbf{r}_t} - r_t(s'_t, a'_t)|\mathcal{F}_{t-\tau}\right] = 0.$$

Hence, we have

$$
\begin{aligned}
\mathbb{E}&\left[J^{\boldsymbol{\pi}_{t-\tau-1}, \mathbf{P}_{t-\tau}, \mathbf{r}_t} - r_t(\tilde{s}_t, \tilde{a}_t)|\mathcal{F}_{t-\tau}\right] \\
&= \mathbb{E}\left[J^{\boldsymbol{\pi}_{t-\tau-1}, \mathbf{P}_{t-\tau}, \mathbf{r}_t} - r_t(s'_t, a'_t) - r_t(\tilde{s}_t, \tilde{a}_t) + r_t(s'_t, a'_t)|\mathcal{F}_{t-\tau}\right] \\
&= \mathbb{E}\left[r_t(s'_t, a'_t) - r_t(\tilde{s}_t, \tilde{a}_t)|\mathcal{F}_{t-\tau}\right] \\
&\leq 2U_R \cdot 2d_{TV}\left(d^{\boldsymbol{\pi}_{t-\tau-1}, \mathbf{P}_{t-\tau}} \otimes \boldsymbol{\pi}_{t-\tau-1}, P((\tilde{s}_t, \tilde{a}_t) \in \cdot|\mathcal{F}_{t-\tau})\right) \\
&\overset{(a)}{\leq} 4U_R d_{TV}\left(d^{\boldsymbol{\pi}_{t-\tau-1}, \mathbf{P}_{t-\tau}}, P(\tilde{s}_t \in \cdot|\mathcal{F}_{t-\tau})\right) \\
&\overset{(b)}{\leq} 4U_R m\rho^\tau
\end{aligned}
$$

where $(a)$ follows from Lemma B.1 in [33] and $(b)$ is by Assumption 1. $\qquad\square$

**Lemma 18.** *For any* $\boldsymbol{\pi}, \boldsymbol{\pi}', \mathbf{P}, \mathbf{P}', \mathbf{r}, \eta$, *and* $O = (s, a, s', a')$, *we have*

$$|\Lambda(\boldsymbol{\pi}, \mathbf{P}, \mathbf{r}, \eta, O) - \Lambda(\boldsymbol{\pi}', \mathbf{P}', \mathbf{r}, \eta, O)| \leq F_5 L_{\boldsymbol{\pi}} \|\boldsymbol{\pi} - \boldsymbol{\pi}'\|_2 + F_5 L_P \|\mathbf{P} - \mathbf{P}'\|_\infty,$$

*where* $F_5 = 4U_R$.

*Proof.*

$$\begin{aligned}
&|\Lambda(\boldsymbol{\pi}, \mathbf{P}, \mathbf{r}, \eta, O) - \Lambda(\boldsymbol{\pi}', \mathbf{P}', \mathbf{r}, \eta, O)| \\
&\leq |(\eta - J^{\boldsymbol{\pi}, \mathbf{P}, \mathbf{r}})(r(s, a) - J^{\boldsymbol{\pi}, \mathbf{P}, \mathbf{r}}) - (\eta - J^{\boldsymbol{\pi}', \mathbf{P}', \mathbf{r}})(r(s, a) - J^{\boldsymbol{\pi}', \mathbf{P}', \mathbf{r}})| \\
&\leq |(\eta - J^{\boldsymbol{\pi}, \mathbf{P}, \mathbf{r}})(r(s, a) - J^{\boldsymbol{\pi}, \mathbf{P}, \mathbf{r}}) - (\eta - J^{\boldsymbol{\pi}, \mathbf{P}, \mathbf{r}})(r(s, a) - J^{\boldsymbol{\pi}', \mathbf{P}', \mathbf{r}})| \\
&\quad + |(\eta - J^{\boldsymbol{\pi}, \mathbf{P}, \mathbf{r}})(r(s, a) - J^{\boldsymbol{\pi}', \mathbf{P}', \mathbf{r}}) - (\eta - J^{\boldsymbol{\pi}', \mathbf{P}', \mathbf{r}})(r(s, a) - J^{\boldsymbol{\pi}', \mathbf{P}', \mathbf{r}})| \\
&\leq 4U_R |J^{\boldsymbol{\pi}, \mathbf{P}, \mathbf{r}} - J^{\boldsymbol{\pi}', \mathbf{P}', \mathbf{r}}| \overset{(a)}{\leq} 4U_R L_{\boldsymbol{\pi}} \|\boldsymbol{\pi} - \boldsymbol{\pi}'\|_2 + 4U_R L_P \|\mathbf{P} - \mathbf{P}'\|_\infty
\end{aligned}$$

where $(a)$ follows from Lemma 6. $\qquad\square$

**Lemma 19.** *For any* $\boldsymbol{\pi}, \mathbf{P}, \mathbf{r}, \eta, \eta'$ *and* $O = (s, a, s', a')$, *we have*

$$|\Lambda(\boldsymbol{\pi}, \mathbf{P}, \mathbf{r}, \eta, O) - \Lambda(\boldsymbol{\pi}, \mathbf{P}, \mathbf{r}, \eta', O)| \leq F_6 |\eta - \eta'|$$

*where* $F_6 = 2U_R$.

*Proof.* Recall the definition of $\Lambda(\cdot)$ in Section B. It is straightforward to see that

$$|\Lambda(\boldsymbol{\pi}, \mathbf{P}, \mathbf{r}, \eta, O) - \Lambda(\boldsymbol{\pi}, \mathbf{P}, \mathbf{r}, \eta', O)| \leq 2U_R |\eta - \eta'|$$

$\qquad\square$

**Lemma 20.** *Consider an observation from the original Markov chain by* $O_t = (s_t, a_t, s_{t+1}, a_{t+1})$ *and auxiliary Markov chain by* $\tilde{O}_t = (\tilde{s}_t, \tilde{a}_t, \tilde{s}_{t+1}, \tilde{a}_{t+1})$. *Conditioned on* $\mathcal{F}_{t-\tau} = \{s_{t-\tau}, \boldsymbol{\pi}_{t-\tau-1}, \mathbf{P}_{t-\tau}\}$, *we have*

$$\begin{aligned}
&\mathbb{E}\left[\Lambda(\boldsymbol{\pi}_{t-\tau-1}, \mathbf{P}_{t-\tau}, \mathbf{r}_t, \eta_{t-\tau}, O_t) - \Lambda(\boldsymbol{\pi}_{t-\tau-1}, \mathbf{P}_{t-\tau}, \mathbf{r}_t, \eta_{t-\tau}, \tilde{O}_t) \big| \mathcal{F}_{t-\tau}\right] \\
&\leq F_7 \sqrt{|\mathcal{S}||\mathcal{A}|} \mathbb{E}\left[\sum_{i=t-\tau}^t \|\boldsymbol{\pi}_i - \boldsymbol{\pi}_{t-\tau-1}\|_2 \Big| \mathcal{F}_{t-\tau}\right] + F_7 \sum_{i=t-\tau}^t \|\mathbf{P}_i - \mathbf{P}_{t-\tau}\|_\infty
\end{aligned}$$

*where* $F_7 = 8U_R^2$.

*Proof.*

$$\begin{aligned}
&\mathbb{E}\left[\Lambda(\boldsymbol{\pi}_{t-\tau-1}, \mathbf{P}_{t-\tau}, \mathbf{r}_t, \eta_{t-\tau}, O_t) - \Lambda(\boldsymbol{\pi}_{t-\tau-1}, \mathbf{P}_{t-\tau}, \mathbf{r}_t, \eta_{t-\tau}, \tilde{O}_t) \big| \mathcal{F}_{t-\tau}\right] \\
&= (\eta_{t-\tau} - J^{\boldsymbol{\pi}_{t-\tau-1}, \mathbf{P}_{t-\tau}, \mathbf{r}_t}) \mathbb{E}\left[r_t(s_t, a_t) - r_t(\tilde{s}_t, \tilde{a}_t) \big| \mathcal{F}_{t-\tau}\right] \\
&\leq 2U_R \cdot 4U_R d_{TV}\left(P(O_t \in \cdot | \mathcal{F}_{t-\tau}), P(\tilde{O}_t \in \cdot | \mathcal{F}_{t-\tau})\right) \\
&\overset{(a)}{\leq} F_7 \sqrt{|\mathcal{S}||\mathcal{A}|} \mathbb{E}\left[\sum_{i=t-\tau}^t \|\boldsymbol{\pi}_i - \boldsymbol{\pi}_{t-\tau-1}\|_2 \Big| \mathcal{F}_{t-\tau}\right] + F_7 \sum_{i=t-\tau}^t \|\mathbf{P}_i - \mathbf{P}_{t-\tau}\|_\infty
\end{aligned}$$

where $(a)$ follows from Lemma 23. $\qquad\square$

**Lemma 21.** *Consider an observation from the original Markov chain by* $O_t = (s_t, a_t, s_{t+1}, a_{t+1})$ *and auxiliary Markov chain by* $\tilde{O}_t = (\tilde{s}_t, \tilde{a}_t, \tilde{s}_{t+1}, \tilde{a}_{t+1})$. *Conditioned on* $\mathcal{F}_{t-\tau} = \{s_{t-\tau}, \boldsymbol{\pi}_{t-\tau-1}, \mathbf{P}_{t-\tau}\}$, *we have*

$$\mathbb{E}\left[\Lambda(\boldsymbol{\pi}_{t-\tau-1}, \mathbf{P}_{t-\tau}, \mathbf{r}_t, \eta_{t-\tau}, \tilde{O}_t) \big| \mathcal{F}_{t-\tau}\right] \leq F_8 m \rho^\tau$$

*where* $F_8 = 8U_R^2$.

*Proof.* Consider the observation tuple $O'_t = (s'_t, a'_t, s'_{t+1}, a'_{t+1})$ where $s'_t \sim d^{\boldsymbol{\pi}_{t-\tau-1}, \mathbf{P}_{t-\tau}}(\cdot)$, $a'_t \sim \boldsymbol{\pi}_{t-\tau-1}(\cdot|s'_t)$, $s'_{t+1} \sim \mathbf{P}_{t-\tau}(\cdot|s'_t, a'_t)$ and $a'_{t+1} \sim \boldsymbol{\pi}_{t-\tau-1}(\cdot|s'_{t+1})$.

We know

$$\mathbb{E}\left[\Lambda(\boldsymbol{\pi}_{t-\tau-1}, \mathbf{P}_{t-\tau}, \mathbf{r}_t, \eta_{t-\tau}, O'_t)\big|\mathcal{F}_{t-\tau}\right] = 0.$$

Hence, we have

$$\begin{aligned}
\mathbb{E}&\left[\Lambda(\boldsymbol{\pi}_{t-\tau-1}, \mathbf{P}_{t-\tau}, \mathbf{r}_t, \eta_{t-\tau}, \tilde{O}_t)\big|\mathcal{F}_{t-\tau}\right] \\
&= \mathbb{E}\left[\Lambda(\boldsymbol{\pi}_{t-\tau-1}, \mathbf{P}_{t-\tau}, \mathbf{r}_t, \eta_{t-\tau}, \tilde{O}_t)\big| - \Lambda(\boldsymbol{\pi}_{t-\tau-1}, \mathbf{P}_{t-\tau}, \mathbf{r}_t, \eta_{t-\tau}, O'_t)\big|\mathcal{F}_{t-\tau}\right] \\
&= \mathbb{E}\left[(\eta_{t-\tau} - J^{\boldsymbol{\pi}_{t-\tau-1}, \mathbf{P}_{t-\tau}, \mathbf{r}_t})(r_t(\tilde{s}_t, \tilde{a}_t) - r_t(s'_t, a'_t))\big|\mathcal{F}_{t-\tau}\right] \\
&\leq 2U_R \cdot 4U_R d_{TV}\left(d^{\boldsymbol{\pi}_{t-\tau-1}, \mathbf{P}_{t-\tau}} \otimes \boldsymbol{\pi}_{t-\tau-1}, P((\tilde{s}_t, \tilde{a}_t) \in \cdot|\mathcal{F}_{t-\tau})\right) \\
&\overset{(a)}{\leq} 2U_R \cdot 4U_R d_{TV}\left(d^{\boldsymbol{\pi}_{t-\tau-1}, \mathbf{P}_{t-\tau}}, P(\tilde{s}_t \in \cdot|\mathcal{F}_{t-\tau})\right) \\
&\overset{(b)}{\leq} 8U_R^2 m\rho^\tau
\end{aligned}$$

where $(a)$ follows from Lemma B.1 in [33] and $(b)$ is by Assumption 1. $\qquad\square$

### D.6 Preliminary Lemmas

**Lemma 22.** *For any policies $\boldsymbol{\pi}, \boldsymbol{\pi}'$ and transition probabilities matrices $\mathbf{P}, \mathbf{P}'$, it holds that*

$$d_{TV}\left(d^{\boldsymbol{\pi}, \mathbf{P}}, d^{\boldsymbol{\pi}', \mathbf{P}'}\right) \leq M\sqrt{|\mathcal{S}||\mathcal{A}|}\|\boldsymbol{\pi} - \boldsymbol{\pi}'\|_2 + M\|\mathbf{P} - \mathbf{P}'\|_\infty,$$

$$d_{TV}\left(d^{\boldsymbol{\pi}, \mathbf{P}} \otimes \boldsymbol{\pi}, d^{\boldsymbol{\pi}', \mathbf{P}'} \otimes \boldsymbol{\pi}'\right) \leq (M+1)\sqrt{|\mathcal{S}||\mathcal{A}|}\|\boldsymbol{\pi} - \boldsymbol{\pi}'\|_2 + M\|\mathbf{P} - \mathbf{P}'\|_\infty,$$

$$d_{TV}\left(d^{\boldsymbol{\pi}, \mathbf{P}} \otimes \boldsymbol{\pi} \otimes \mathbf{P}, d^{\boldsymbol{\pi}', \mathbf{P}'} \otimes \boldsymbol{\pi}' \otimes \mathbf{P}'\right) \leq (M+1)\sqrt{|\mathcal{S}||\mathcal{A}|}\|\boldsymbol{\pi} - \boldsymbol{\pi}'\|_2 + (M+1)\|\mathbf{P} - \mathbf{P}'\|_\infty,$$

$$d_{TV}\left(d^{\boldsymbol{\pi}, \mathbf{P}} \otimes \boldsymbol{\pi} \otimes \mathbf{P} \otimes \boldsymbol{\pi}, d^{\boldsymbol{\pi}', \mathbf{P}'} \otimes \boldsymbol{\pi}' \otimes \mathbf{P}' \otimes \boldsymbol{\pi}'\right) \leq (M+2)\sqrt{|\mathcal{S}||\mathcal{A}|}\|\boldsymbol{\pi} - \boldsymbol{\pi}'\|_2 + (M+1)\|\mathbf{P} - \mathbf{P}'\|_\infty$$

*where $\otimes$ denotes the Kronecker product, and $M := \left(\lceil\log_\rho m^{-1}\rceil + \frac{1}{1-\rho}\right)$.*

*Proof.* Recall that $d^{\boldsymbol{\pi}, \mathbf{P}}(\cdot)$ is the stationary distribution induced over the states by a Markov chain with transition probabilities $\mathbf{P}$ following policy $\boldsymbol{\pi}$. Define the matrices $\mathbf{K}, \mathbf{K}' \in \mathbb{R}^{|\mathcal{S}| \times |\mathcal{S}|}$ such that $\mathbf{K}(s, s') = \sum_{a \in \mathcal{A}} P(s'|s, a)\pi(a|s)$ and $\mathbf{K}'(s, s') = \sum_{a \in \mathcal{A}} P'(s'|s, a)\pi'(a|s)$. Further denote the total variation norm as $\|\cdot\|_{TV}$. Note that $\|\mathbf{P} - \mathbf{P}'\|_\infty = \max_{s,a} \sum_{s'} |P(s'|s, a) - P'(s'|s, a)|$.

From Theorem 3.1 in [56], we have,

$$\begin{aligned}
d_{TV}\left(d^{\boldsymbol{\pi}, \mathbf{P}}, d^{\boldsymbol{\pi}', \mathbf{P}'}\right) &\leq M \sup_{\|q\|_{TV}=1} \left\|\int_\mathcal{S} q(ds)(\mathbf{K} - \mathbf{K}')(s, \cdot)\right\|_{TV} \\
&\leq M \sup_{\|q\|_{TV}=1} \int_\mathcal{S} \left|\int_\mathcal{S} q(ds)(\mathbf{K} - \mathbf{K}')(s, ds')\right| \\
&\leq M \sup_{\|q\|_{TV}=1} \int_\mathcal{S} \int_\mathcal{S} |q(ds)| \left|\sum_{a \in \mathcal{A}} P(ds'|s, a)\pi(a|s) - P'(ds'|s, a)\pi'(a|s)\right| \\
&\leq M \sup_{\|q\|_{TV}=1} \int_\mathcal{S} \int_\mathcal{S} \sum_a |q(ds)| |P(ds'|s, a)\pi(a|s) - P(ds'|s, a)\pi'(a|s)| \\
&\quad + M \sup_{\|q\|_{TV}=1} \int_\mathcal{S} \int_\mathcal{S} \sum_a |q(ds)| |P(ds'|s, a)\pi'(a|s) - P'(ds'|s, a)\pi'(a|s)|
\end{aligned}$$

$$\leq M\sqrt{|\mathcal{S}||\mathcal{A}|}\|\boldsymbol{\pi} - \boldsymbol{\pi}'\|_2 + M\|\mathbf{P} - \mathbf{P}'\|_\infty.$$

For the second inequality, we have,

$$d_{TV}\left(d^{\boldsymbol{\pi},\mathbf{P}} \otimes \boldsymbol{\pi}, d^{\boldsymbol{\pi}',\mathbf{P}'} \otimes \boldsymbol{\pi}'\right) \leq \frac{1}{2}\int_\mathcal{S}\sum_a \left|d^{\boldsymbol{\pi},\mathbf{P}}(ds)\pi(a|s) - d^{\boldsymbol{\pi}',\mathbf{P}'}(ds)\pi'(a|s)\right|$$

$$\leq \frac{1}{2}\int_\mathcal{S}\sum_a \left|d^{\boldsymbol{\pi},\mathbf{P}}(ds)\pi(a|s) - d^{\boldsymbol{\pi},\mathbf{P}}(ds)\pi'(a|s)\right|$$

$$+ \frac{1}{2}\int_\mathcal{S}\sum_a \left|d^{\boldsymbol{\pi},\mathbf{P}}(ds)\pi'(a|s) - d^{\boldsymbol{\pi}',\mathbf{P}'}(ds)\pi'(a|s)\right|$$

$$\leq \sqrt{|\mathcal{S}||\mathcal{A}|}\|\boldsymbol{\pi} - \boldsymbol{\pi}'\|_2 + d_{TV}\left(d^{\boldsymbol{\pi},\mathbf{P}}, d^{\boldsymbol{\pi}',\mathbf{P}'}\right)$$

$$\leq (M+1)\sqrt{|\mathcal{S}||\mathcal{A}|}\|\boldsymbol{\pi} - \boldsymbol{\pi}'\|_2 + M\|\mathbf{P} - \mathbf{P}'\|_\infty.$$

The rest follow in a similar manner. $\qquad\square$

**Lemma 23.** *Consider observations $O_t = (s_t, a_t, s_{t+1}, a_{t+1})$ and $\tilde{O}_t = (\tilde{s}_t, \tilde{a}_t, \tilde{s}_{t+1}, \tilde{a}_{t+1})$ and define $\mathcal{F}_{t-\tau} := \{s_{t-\tau}, \boldsymbol{\pi}_{t-\tau-1}, \mathbf{P}_{t-\tau}\}$. We have*

$$d_{TV}\left(P(O_t \in \cdot|\mathcal{F}_{t-\tau}), P(\tilde{O}_t \in \cdot|\mathcal{F}_{t-\tau})\right) \leq \sqrt{|\mathcal{S}||\mathcal{A}|}\sum_{i=t-\tau}^t \mathbb{E}\left[\|\pi_i - \pi_{t-\tau-1}\|_2\Big|\mathcal{F}_{t-\tau}\right] + \|\mathbf{P}_i - \mathbf{P}_{t-\tau}\|_\infty.$$

*Proof.*

$$d_{TV}\left(P(O_t \in \cdot|\mathcal{F}_{t-\tau}), P(\tilde{O}_t \in \cdot|\mathcal{F}_{t-\tau})\right)$$

$$= \frac{1}{2}\sum_{s,a,s',a'}|P(\overbrace{s_t = s, a_t = a}^{\mathcal{H}_t}, s_{t+1} = s', a_{t+1} = a'|\mathcal{F}_{t-\tau}) - P(\tilde{s}_t = s, \tilde{a}_t = a, \tilde{s}_{t+1} = s', \tilde{a}_{t+1} = a'|\mathcal{F}_{t-\tau})|$$

$$= \frac{1}{2}\sum_{s,a,s',a'}|P(s_t = s, a_t = a|\mathcal{F}_{t-\tau})P_t(s'|s,a)\mathbb{E}\left[\pi_t(a'|s')|\mathcal{F}_{t-\tau}, \mathcal{H}_t\right]$$

$$- P(\tilde{s}_t = s, \tilde{a}_t = a|\mathcal{F}_{t-\tau})P_{t-\tau}(s'|s,a)\pi_{t-\tau-1}(a'|s')|$$

$$\leq \frac{1}{2}\sum_{s,a,s',a'}|P(s_t = s, a_t = a|\mathcal{F}_{t-\tau})P_t(s'|s,a)\mathbb{E}\left[\pi_t(a'|s')|\mathcal{F}_{t-\tau}, \mathcal{H}_t\right]$$

$$- P(\tilde{s}_t = s, \tilde{a}_t = a|\mathcal{F}_{t-\tau})P_t(s'|s,a)\pi_{t-\tau-1}(a'|s')|$$

$$+ \frac{1}{2}\sum_{s,a,s',a'}|P(\tilde{s}_t = s, \tilde{a} = a|\mathcal{F}_{t-\tau})P_t(s'|s,a)\pi_{t-\tau-1}(a'|s')$$

$$- P(\tilde{s}_t = s, \tilde{a}_t = a|\mathcal{F}_{t-\tau})P_{t-\tau}(s'|s,a)\pi_{t-\tau-1}(a'|s')|$$

$$= \frac{1}{2}\sum_{s,a,s',a'}P_t(s'|s,a)P(s_t = s, a_t = a|\mathcal{F}_{t-\tau})|\mathbb{E}\left[\pi_t(a'|s')|\mathcal{F}_{t-\tau}, \mathcal{H}_t\right] - \pi_{t-\tau-1}(a'|s')|$$

$$+ \frac{1}{2}\sum_{s,a}|P(s_t = s, a_t = a|\mathcal{F}_{t-\tau}) - P(\tilde{s}_t = s, \tilde{a}_t = a|\mathcal{F}_{t-\tau})|$$

$$+ \frac{1}{2}\sum_{s,a,s',a'}P(\tilde{s}_t = s, \tilde{a}_t = a|\mathcal{F}_{t-\tau})\pi_{t-\tau-1}(a'|s')|P_t(s'|s,a) - P_{t-\tau}(s'|s,a)|$$

$$\leq \sqrt{|\mathcal{S}||\mathcal{A}|}\mathbb{E}\left[\|\pi_t - \pi_{t-\tau-1}\|_2\Big|\mathcal{F}_{t-\tau}\right] + d_{TV}\left(P(O_{t-1} \in \cdot|\mathcal{F}_{t-\tau}), P(\tilde{O}_{t-1} \in \cdot|\mathcal{F}_{t-\tau})\right) + \|\mathbf{P}_t - \mathbf{P}_{t-\tau}\|_\infty.$$

Finally, recursing backwards until $\tau$ yields the result. $\qquad\square$

**Lemma 24.** *If an observation is denoted as $O = (s, a, s', a')$, then the following hold for all $t, t'$*

1. $\|Q_t^{\pi}\|_2 \le U_Q$; $\|Q_t\|_2 \le R_Q = U_Q$

2. $\|\mathbf{A}(O)\|_{\infty} \le 2$; $\|\mathbf{A}(O)\|_2 \le \sqrt{2}$

3. $\|\bar{\mathbf{A}}^{\boldsymbol{\pi},\mathbf{P}} - \bar{\mathbf{A}}^{\boldsymbol{\pi}',\mathbf{P}'}\|_{\infty} \le 2d_{TV}\left(d^{\boldsymbol{\pi},\mathbf{P}} \otimes \boldsymbol{\pi} \otimes \mathbf{P} \otimes \boldsymbol{\pi}, d^{\boldsymbol{\pi}',\mathbf{P}'} \otimes \boldsymbol{\pi}' \otimes \mathbf{P}' \otimes \boldsymbol{\pi}'\right)$

4. $\|\boldsymbol{\psi}_{t+1} - \boldsymbol{\psi}_t\|_2 \le \|\mathbf{Q}_{t+1} - \mathbf{Q}_t\|_2 + \|\mathbf{Q}_{t+1}^{\pi_{t+1}} - \mathbf{Q}_t^{\pi_t}\|_2$

*Proof.* We have the following.

1. See the projection operator $\Pi_{R_Q}(\cdot)$ used in Algorithm 1 and discussed further in Section 5.1.

2. Follows from the definition of $\mathbf{A}(O)$ in Section 5.1

3. Follows from the definition of $\bar{\mathbf{A}}^{\boldsymbol{\pi},\mathbf{P}}$ in Section 5.1 and

$$\|\bar{\mathbf{A}}^{\boldsymbol{\pi},\mathbf{P}} - \bar{\mathbf{A}}^{\boldsymbol{\pi}',\mathbf{P}'}\|_{\infty} = \max_{s,a} \sum_{s',a'} |d^{\boldsymbol{\pi},\mathbf{P}}(s,a)\boldsymbol{\pi}(a|s)\mathbf{P}(s'|s,a)\boldsymbol{\pi}(a'|s')$$
$$- d^{\boldsymbol{\pi}',\mathbf{P}'}(s,a)\boldsymbol{\pi}'(a|s)\mathbf{P}'(s'|s,a)\boldsymbol{\pi}'(a'|s')|$$

4. By the definition of $\boldsymbol{\psi}_t = \Pi_E\left[\mathbf{Q}_t - \mathbf{Q}_t^{\boldsymbol{\pi}_t}\right]$ and triangle inequality

$\square$

## D.7 Universal Dynamic Regret

While dynamic regret as defined in Equation (2) remains the predominant metric of performance in non-stationary RL literature [12, 47, 18, 16, 17], we additionally consider the universal dynamic regret often used in the adversarial learning literature [57, 58, 59] for completeness. We now present an upper bound on the universal dynamic regret defined as the difference between the optimal total reward that can be obtained in $T$ time-steps and the reward accumulated by our algorithm as follows

$$\text{U-Dyn-Reg}(\mathcal{M}, T) = \max_{\{\boldsymbol{\pi}_t^c\}_{t=0}^{T-1}} \mathbb{E}_{a_t^c \sim \boldsymbol{\pi}_t^c(\cdot|s_t^c)}\left[\sum_{t=0}^{T-1} r_t(s_t^c, a_t^c)\right] - \mathbb{E}\left[\sum_{t=0}^{T-1} r_t(s_t, a_t)\right]. \tag{35}$$

Observe that this notion of regret is slightly different from the dynamic regret defined previously in Equation (2) which compares against the sum of average rewards $J_t^{\boldsymbol{\pi}_t^{\star}}$ obtained by the policies $\boldsymbol{\pi}_t^{\star}$.

We make the following additional assumptions on the structure of the MDP. The condition number captures the sensitivity of the stationary distribution to the change in probabilities and the recurrent coefficient captures how often a state is visited.

**Assumption 2** (Bounded Condition Number, [60])**.** The condition number of a Markov chain following $\boldsymbol{\pi}$ induced by the transition probability matrix $\mathbf{P}^{\boldsymbol{\pi}} \in \mathbb{R}^{|\mathcal{S}| \times |\mathcal{S}|}$ is defined as the maximum norm of the Drazin inverse of $I - \mathbf{P}^{\boldsymbol{\pi}}$ as

$$\kappa(\mathbf{P}^{\boldsymbol{\pi}}) = \|(I - \mathbf{P}^{\boldsymbol{\pi}})^{\#}\|_{\max}$$

where $\mathbf{B}^{\#}$ represents the Drazin inverse of matrix $\mathbf{B}$ and $\|\mathbf{B}\|_{\max} = \max_{i,j} B_{i,j}$. We assume that for all environments $t \in [T]$, there exists a $\kappa > 0$, such that the condition numbers of Markov chain $\mathcal{M}_t$ following the optimal policy $\boldsymbol{\pi}_t^{\star}$ are bounded as

$$\kappa(\mathbf{P}_t^{\boldsymbol{\pi}_t^{\star}}) \le \kappa.$$

**Assumption 3** (Bounded Recurrent Coefficient, [61]). Denote the first passage time of state $j$ from state $i$ in an MDP $\mathcal{M}(\mathcal{S}, \mathcal{A}, \mathbf{P}, \mathbf{r})$ following policy $\boldsymbol{\pi}$ as $Y_{i,j}^{\boldsymbol{\pi}, \mathbf{P}}$. The recurrent coefficient of state $j$, denoted by $\nu_j$, is defined as the probability that the first passage time $Y_{i,j}^{\boldsymbol{\pi}}$ of state $j$ from any state $i$ following any policy $\boldsymbol{\pi}$ is smaller than the number of states as

$$\nu_j(\mathcal{M}) = \min_{i, \boldsymbol{\pi}} \mathrm{Prob}(Y_{i,j}^{\boldsymbol{\pi}} \leq |\mathcal{S}|).$$

We assume that for all environments $t \in [T]$ and all states $j \in \mathcal{S}$, there exists a $\nu$, such that the recurrent coefficients $\nu_j(\mathcal{M}_t)$ are bounded as

$$\nu_j(\mathcal{M}_t) \geq \nu.$$

**Corollary 1.** *If Assumption 1, Assumption 2, Assumption 3 are satisfied in Algorithm 1 with appropriate choice of parameters, then*

$$U\text{-}Dyn\text{-}Reg(\mathcal{M}, T) = \max_{\{\boldsymbol{\pi}_t^c\}_{t=0}^{T-1}} \mathbb{E}\left[\sum_{t=0}^{T-1} r_t(s_t^c, \boldsymbol{\pi}_t^c)\right] - \mathbb{E}\left[\sum_{t=0}^{T-1} r_t(s_t, a_t)\right] \leq \tilde{\mathcal{O}}\left(|\mathcal{S}|^{1/2}|\mathcal{A}|^{1/2}\Delta_T^{1/6}T^{5/6}\right).$$

*Proof.* We decompose the regret as

$$
\begin{aligned}
\text{U-Dyn-Reg}(\mathcal{M}, T) &= \max_{\{\boldsymbol{\pi}_t^c\}_{t=0}^{T-1}} \mathbb{E}\left[\sum_{t=0}^{T-1} r_t(s_t^c, \boldsymbol{\pi}_t^c)\right] - \mathbb{E}\left[\sum_{t=0}^{T-1} r_t(s_t, a_t)\right] \\
&= \left(\max_{\{\boldsymbol{\pi}_t^c\}_{t=0}^{T-1}} \mathbb{E}\left[\sum_{t=0}^{T-1} r_t(s_t^c, \boldsymbol{\pi}_t^c)\right] - \mathbb{E}\left[\sum_{t=0}^{T-1} J_t^{\boldsymbol{\pi}_t^\star}\right]\right) + \left(\mathbb{E}\left[\sum_{t=0}^{T-1} J_t^{\boldsymbol{\pi}_t^\star}\right] - \mathbb{E}\left[\sum_{t=0}^{T-1} r_t(s_t, a_t)\right]\right) \\
&\stackrel{(a)}{\leq} \tilde{\mathcal{O}}\left(\frac{\kappa}{\nu^2}\Delta_T\right) + \left(\mathbb{E}\left[\sum_{t=0}^{T-1} J_t^{\boldsymbol{\pi}_t^\star}\right] - \mathbb{E}\left[\sum_{t=0}^{T-1} r_t(s_t, a_t)\right]\right) \\
&\stackrel{(b)}{\leq} \tilde{\mathcal{O}}\left(\frac{\kappa}{\nu^2}\Delta_T\right) + \tilde{\mathcal{O}}\left(|\mathcal{S}|^{1/2}|\mathcal{A}|^{1/2}\Delta_T^{1/6}T^{5/6}\right)
\end{aligned}
$$

where $(a)$ follows from Lemma 1 (b), Corollary 2, Theorem 1 in [59] and $(b)$ follows from Theorem 4. $\qquad\square$

## E  NS-NAC with Function Approximation

In this section, we present the NS-NAC algorithm with function approximated policy and the state-action value function and the associated regret bound. Consider the policy $\boldsymbol{\pi}_{\boldsymbol{\theta}}$ parameterized by $\boldsymbol{\theta} \in \mathbb{R}^d$. Consider the state-action value function $\mathbf{Q}^{\boldsymbol{\pi}_{\boldsymbol{\theta}}}(s, a)$ approximated as a linear function $f_{\boldsymbol{\theta}}^T(s, a)\omega$ where $f_{\boldsymbol{\theta}}(s, a)$ denotes the feature vector and $\omega \in \mathbb{R}^d$. We assume the actor and the critic function approximations to be compatible as $f_{\boldsymbol{\theta}}(s, a) = \nabla_{\boldsymbol{\theta}} \log \boldsymbol{\pi}_{\boldsymbol{\theta}}(a|s)$ [62, 63]. The natural policy gradient [20] can hence be expressed as

$$\boldsymbol{\theta}_{t+1} \leftarrow \boldsymbol{\theta}_t + \beta F_{\boldsymbol{\theta}_t}^{-1} \mathbb{E}_{s,a}\left[f_{\boldsymbol{\theta}_t}(s, a)(f_{\boldsymbol{\theta}_t}^T(s, a)\omega_{\boldsymbol{\theta}_t}^\star)\right]$$

where $\omega_{\boldsymbol{\theta}_t}^\star = \arg\min_{\omega} \mathbb{E}\left[(\mathbf{Q}_t^{\boldsymbol{\pi}_{\boldsymbol{\theta}_t}}(s, a) - f_{\boldsymbol{\theta}_t}^T(s, a)\omega)^2\right]$. In the absence of the information of the exact gradient, the *actor* update step corresponding to line 10 thus becomes

$$\boldsymbol{\theta}_{t+1} \leftarrow \boldsymbol{\theta}_t + \beta\omega_t.$$

The TD update step of the *critic* in line 9 can be written as

$$\omega_{t+1} \leftarrow \omega_t + \alpha\left[r_t(s_t, a_t) - \eta_t + f_t^T(s_{t+1}, a_{t+1})\omega_t - f_t^T(s_t, a_t)\omega_t\right] f_t(s_t, a_t).$$

We now detail the assumptions under which the following upper bound on the dynamic regret of NS-NAC with function approximation holds.

**Assumption 4** (Uniform Ergodicity). A Markov chain generated by implementing policy $\boldsymbol{\pi_\theta}$ and transition probabilities $\mathbf{P}$ is called uniformly ergodic, if there exists $m > 0$ and $\rho \in (0,1)$ such that

$$d_{TV}\left(P(s_\tau \in \cdot | s_0 = s), d^{\boldsymbol{\pi_\theta}, \mathbf{P}}\right) \le m\rho^\tau \; \forall \tau \ge 0, s \in \mathcal{S},$$

where $d^{\boldsymbol{\pi_\theta}, \mathbf{P}}$ is the stationary distribution induced over the states. We assume Markov chains induced by all potential policies $\boldsymbol{\pi_{\theta_t}}$ in all environments $\mathbf{P}_t$, $t \in [T]$, are uniformly ergodic.

**Assumption 5.** For all potential parameters $\boldsymbol{\theta}_t$ in all environments $\mathbf{P}_t, t \in [T]$, the maximum eigenvalue of matrix $\bar{A}^{\boldsymbol{\pi_{\theta_t}}, \mathbf{P}_t} = \mathbb{E}_{s,a,s',a'}\left[f_t(s,a)(f_t(s',a') - f_t(s,a))^T\right]$ is $-\lambda$.

**Assumption 6** (Smoothness and Boundedness). For any $\boldsymbol{\theta}, \boldsymbol{\theta}' \in \mathbb{R}^d$ and any state-action pair $s \in \mathcal{S}, a \in \mathcal{A}$, there exist positive constants $L_A, L_C$ such that

1. $\|f_{\boldsymbol{\theta}}\|_2 \le 1$,

2. $\|f_{\boldsymbol{\theta}}(s,a) - f_{\boldsymbol{\theta}'}(s,a)\| \le L_C \|\boldsymbol{\theta} - \boldsymbol{\theta}'\|_2$, and

3. $\|\boldsymbol{\pi_\theta}(\cdot|s) - \boldsymbol{\pi_{\theta'}}(\cdot|s)\|_{TV} \le L_A \|\boldsymbol{\theta} - \boldsymbol{\theta}'\|_2$.

**Definition 1.** Define the compatible linear function approximation error as

$$\epsilon_{app} := \max_{\boldsymbol{\theta}} \min_{\omega} \mathbb{E}_{s \sim d^{\boldsymbol{\pi_\theta}, \mathbf{P}_t}, a \sim \boldsymbol{\pi_\theta}}\left[\|\mathbf{Q}_t^{\boldsymbol{\pi}}(s,a) - f_{\boldsymbol{\theta}}^T(s,a)\omega\|_2^2\right].$$

The dynamic regret achieved by NS-NAC with function approximation described above can be upper bounded as follows.

**Proposition 5.** *If assumptions 4, 5 and 6 are satisfied, $\epsilon_{app}$ is the function approximation error defined in 1 and the parameters of* NS-NAC *with $d$-dimensioned function approximation are chosen optimally, then*

$$Dyn\text{-}Reg(\mathcal{M}, T) = \mathbb{E}\left[\sum_{t=0}^{T-1} J_t^{\boldsymbol{\pi_t^\star}} - r_t(s_t, a_t)\right] = \tilde{\mathcal{O}}\left(d^{1/2}\Delta_T^{1/6}T^{5/6}\right) + \tilde{\mathcal{O}}\left(d^{1/2}\epsilon_{app}T\right).$$

*Proof.* Function approximation has been used commonly in actor-critic [34, 33] and natural actor-critic [43] algorithms in the infinite-horizon average reward setting. For the sake of brevity, we choose not to repeat the proof here and instead we point the readers to [43] for the technique to incorporate function approximation into our analysis of NS-NAC detailed in Section D-Section D.6 above. Note that the structure of the proof including the methods of actor, critic and average reward analyses remains the same with the only difference lying in accounting for the function approximation error $\epsilon_{app}$. $\square$

# F   Regret Analysis: BORL-NS-NAC

**Theorem 5.** *If Assumption 1 is satisfied, the time horizon $T$ is divided into epochs of length $W = \mathcal{O}(T^{2/3})$ in Algorithm 2, then we have for any $j^\dagger \in \{0, 1, \cdots, \lfloor \ln T \rfloor\}$*

$$Dyn\text{-}Reg(\mathcal{M}, T) \le \underbrace{\tilde{\mathcal{O}}\left(W\sqrt{\ln T \cdot \frac{T}{W}}\right)}_{cost\ of\ hedging\ by\ EXP3.P} + \underbrace{\tilde{\mathcal{O}}\left(\frac{TN^\dagger}{W\beta^\dagger}\right) + \tilde{\mathcal{O}}\left(\frac{T}{W}\sqrt{\frac{N^\dagger W}{\alpha^\dagger}}\right)}_{effect\ of\ initialization} + \underbrace{\tilde{\mathcal{O}}\left(\frac{\beta^\dagger T}{\alpha^\dagger}\right) + \tilde{\mathcal{O}}\left(T\sqrt{\beta^\dagger}\right)}_{\substack{cumulative\ change \\ in\ policy\ over\ horizon\ T}} \quad (36)$$

$$+ \underbrace{\tilde{\mathcal{O}}\left(\frac{\beta^\dagger T}{\gamma^\dagger}\right) + \tilde{\mathcal{O}}\left(T\sqrt{\gamma^\dagger}\right) + \tilde{\mathcal{O}}\left(\frac{T}{W}\sqrt{\frac{N^\dagger W}{\gamma}}\right)}_{error\ in\ average\ reward\ estimate\ at\ critic} + \underbrace{\tilde{\mathcal{O}}\left(T\sqrt{\alpha^\dagger}\right)}_{\substack{cumulative\ change \\ in\ critic\ estimates}} + \underbrace{\tilde{\mathcal{O}}\left(\frac{\Delta_T W}{N^\dagger}\right) + \tilde{\mathcal{O}}\left(\frac{\Delta_T^{1/3}T^{2/3}}{\sqrt{\alpha^\dagger}} + \frac{\Delta_T^{1/3}T^{2/3}}{\sqrt{\gamma^\dagger}}\right)}_{error\ due\ to\ non\text{-}stationarity}),$$

where $\Delta_T = \Delta_{R,T} + \Delta_{P,T}$, $\alpha^\dagger = \gamma^\dagger = \left(\frac{T^{j^\dagger/\lfloor \ln T \rfloor}}{T}\right)^{1/3}$, $\beta^\dagger = \left(\frac{T^{j^\dagger/\lfloor \ln T \rfloor}}{T}\right)^{1/2}$, $N^\dagger = W\left(\frac{T^{j^\dagger/\lfloor \ln T \rfloor}}{T}\right)^{5/6}$ and $\tilde{\mathcal{O}}(\cdot)$ *hides constants and logarithmic dependence on time horizon* $T$. *Choosing optimal value of* $j^\dagger$ *and resulting optimal parameters as* $\alpha^\star = \gamma^\star = \left(\frac{\Delta_T}{T}\right)^{1/3}$, $\beta^\star = \left(\frac{\Delta_T}{T}\right)^{1/2}$ *and* $\frac{N^\star T}{W} = \Delta_T^{5/6}T^{1/6}$, *we upper bound the regret as*

$$Dyn\text{-}Reg(\mathcal{M}, T) \leq \tilde{\mathcal{O}}\left(|\mathcal{S}|^{1/2}|\mathcal{A}|^{1/2}\Delta_T^{1/6}T^{5/6}\right).$$

*Proof.* We start by decomposing the regret, for any $j^\dagger \in \{0, 1, \cdots, \lfloor \ln T \rfloor\}$ and corresponding step-sizes $\alpha^\dagger, \gamma^\dagger, \beta^\dagger$ and number of restarts $N^\dagger$, as follows

$$\begin{aligned}
Dyn\text{-}Reg(\mathcal{M}, T) &= \sum_{i=0}^{\lfloor T/W \rfloor} \mathbb{E}\left[\sum_{t=iW}^{(i+1)W-1} J_t^{\pi_t^\star} - r_t(s_t, a_t)\right] \\
&= \sum_{i=0}^{\lfloor T/W \rfloor} \mathbb{E}\left[\left(\sum_{t=iW}^{(i+1)W-1} J_t^{\pi_t^\star}\right) - R_{i,j^\dagger}\right] + \sum_{i=0}^{\lfloor T/W \rfloor} \mathbb{E}\left[R_{i,j^\dagger} - R_{i,j_i}\right] \\
&\overset{(a)}{\leq} \sum_{i=0}^{\lfloor T/W \rfloor} \mathbb{E}\left[\left(\sum_{t=iW}^{(i+1)W-1} J_t^{\pi_t^\star}\right) - R_{i,j^\dagger}\right] + \tilde{\mathcal{O}}\left(W\sqrt{\ln T \cdot \frac{T}{W}}\right) \\
&\overset{(b)}{\leq} \left[\sum_{i=0}^{\lfloor T/W \rfloor} \tilde{\mathcal{O}}\left(\frac{N^\dagger}{\beta^\dagger}\right) + \tilde{\mathcal{O}}\left(\sqrt{\frac{N^\dagger W}{\alpha^\dagger}}\right) + \tilde{\mathcal{O}}\left(\frac{\beta^\dagger W}{\alpha^\dagger}\right) + \tilde{\mathcal{O}}\left(W\sqrt{\beta^\dagger}\right) + \tilde{\mathcal{O}}\left(\frac{\beta^\dagger W}{\alpha^\dagger}\right) + \tilde{\mathcal{O}}\left(W\sqrt{\gamma^\dagger}\right) \right. \\
&\quad + \tilde{\mathcal{O}}\left(\sqrt{\frac{N^\dagger W}{\gamma^\dagger}}\right) + \tilde{\mathcal{O}}\left(W\sqrt{\alpha^\dagger}\right) + \tilde{\mathcal{O}}\left(\frac{\Delta_{iW,(i+1)W}W}{N^\dagger}\right) + \tilde{\mathcal{O}}\left(\frac{\Delta_{iW,(i+1)W}^{1/3}W^{2/3}}{\sqrt{\alpha^\dagger}}\right) \\
&\quad \left. + \tilde{\mathcal{O}}\left(\frac{\Delta_{iW,(i+1)W}^{1/3}W^{2/3}}{\sqrt{\gamma^\dagger}}\right)\right] + \tilde{\mathcal{O}}\left(W\sqrt{\ln T \cdot \frac{T}{W}}\right) \\
&\overset{(c)}{\leq} \tilde{\mathcal{O}}\left(\frac{TN^\dagger}{W\beta^\dagger}\right) + \tilde{\mathcal{O}}\left(\frac{T}{W}\sqrt{\frac{N^\dagger W}{\alpha^\dagger}}\right) + \tilde{\mathcal{O}}\left(\frac{\beta^\dagger T}{\alpha^\dagger}\right) + \tilde{\mathcal{O}}\left(T\sqrt{\beta^\dagger}\right) + \tilde{\mathcal{O}}\left(\frac{\beta^\dagger T}{\gamma^\dagger}\right) + \tilde{\mathcal{O}}\left(T\sqrt{\gamma^\dagger}\right) \\
&\quad + \tilde{\mathcal{O}}\left(\frac{T}{W}\sqrt{\frac{N^\dagger W}{\gamma^\dagger}}\right) + \tilde{\mathcal{O}}\left(T\sqrt{\alpha^\dagger}\right) + \tilde{\mathcal{O}}\left(\frac{\Delta_T W}{N^\dagger}\right) + \tilde{\mathcal{O}}\left(\frac{\Delta_T^{1/3}T^{2/3}}{\sqrt{\alpha^\dagger}}\right) \\
&\quad + \tilde{\mathcal{O}}\left(\frac{\Delta_T^{1/3}T^{2/3}}{\sqrt{\gamma^\dagger}}\right) + \tilde{\mathcal{O}}\left(W\sqrt{\ln T \cdot \frac{T}{W}}\right)
\end{aligned}$$

where $(a)$ follows from the EXP3.P regret bound of an $\lceil \ln T \rceil$-armed bandit with rewards in $[0, W \cdot U_R]$ as detailed in Section 3.2 of [50], $(b)$ follows from Theorem 4 and $(c)$ follows from Jensen's inequality.

Further, there exists some $j^\dagger$ such that

$$\left(\frac{T^{j^\dagger/\lfloor \ln T \rfloor}}{T}\right)^{1/2} \leq \beta^\star = \left(\frac{\Delta_T}{T}\right)^{1/2} \leq \left(\frac{T^{(j^\dagger+1)/\lfloor \ln T \rfloor}}{T}\right)^{1/2},$$

$$\left(\frac{T^{j^\dagger/\lfloor \ln T \rfloor}}{T}\right)^{1/3} \leq \alpha^\star = \gamma^\star = \left(\frac{\Delta_T}{T}\right)^{1/3} \leq \left(\frac{T^{(j^\dagger+1)/\lfloor \ln T \rfloor}}{T}\right)^{1/3},$$

$$\left(T^{j^\dagger/\lfloor \ln T \rfloor}\right)^{5/6}T^{1/6} \leq \frac{N^\star T}{W} = \Delta_T^{5/6}T^{1/6} \leq \left(T^{(j^\dagger+1)/\lfloor \ln T \rfloor}\right)^{5/6}T^{1/6}.$$

We conclude the proof by adapting $\beta^\star, \alpha^\star, \gamma^\star, N^\star$ to $\beta^\dagger, \alpha^\dagger, \gamma^\dagger, N^\dagger$ in the above regret expression and observing that $T^{1/\lfloor \ln T \rfloor} = \mathcal{O}(1)$ results in the final upper bound presented in the theorem. $\qquad\square$

# G   Simulations

**Synthetic Environment.**   We empirically evaluate the performance of our algorithms on a synthetic non-stationary MDP, comparing it with three baseline algorithms: SW-UCRL2-CW [51], Var-UCRL2 [13], and RestartQ-UCB ([16]). The synthetic MDP environment simulates non-stationary dynamics by alternating between two sets of transition matrices and reward functions over the time horizon $T$. The switching frequency, controlled by $n_{\text{switches}}$, determines the degree of non-stationarity and the variation budget $\Delta_{P,T}$ for transitions and $\Delta_{R,T}$ for rewards. The MDP consists of $|\mathcal{S}|$ states and $|\mathcal{A}|$ actions per state, with two sets of transition probabilities and rewards sampled at initialization. Further, to benchmark the effect of the dynamic changes, the optimal policy is recalculated at each switching step $t_{switch}$ by solving a linear programming problem [55].

The environment alternates between these two sets of transitions and rewards, $(\mathbf{P}_1, \mathbf{r}_1)$ and $(\mathbf{P}_2, \mathbf{r}_2)$, every $T/n_{\text{switches}}$ steps. The transition probabilities, $\mathbf{P}_1$ and $\mathbf{P}_2$, are drawn from a Dirichlet distribution with a concentration parameter set to 0.5, ensuring a moderate degree of randomness in the state transitions. The first reward matrix $\mathbf{r}_1$ is drawn from a Beta distribution with shape parameters $\alpha = 0.5$ and $\beta = 0.5$, leading to rewards spread across the interval $[0, 1]$, with a higher probability near the extremes of 0 and 1. The second reward matrix $\mathbf{r}_2$ is sampled from a Beta distribution with shape parameters $\alpha = 0.2$ and $\beta = 0.9$, producing rewards skewed toward lower values, introducing diversity in the reward structure. We use 5 random seeds to initialize the matrices, with standard deviation capturing variability across these runs.

**Varying $T$.**   We evaluate the performance of different algorithms in the synthetic environment with $|\mathcal{S}| = 50$ and $|\mathcal{A}| = 4$ under varying time horizons $T$. Specifically, the time horizon $T$ is varied over the values $50 \times 10^3, 70 \times 10^3, 100 \times 10^3, 150 \times 10^3, 180 \times 10^3, 200 \times 10^3$, and $250 \times 10^3$. For each $T$, we set $n_{\text{switches}} = 1000$, resulting in a transition variation budget $\Delta_{P,T} = 303$, indicating significant environmental changes across the time horizon. The reward function is kept stationary (no switching between $\mathbf{r}_1$ and $\mathbf{r}_2$), and therefore $\Delta_{R,T} = 0$.

**Varying $\Delta_T$.**   We investigate the impact of changing variation budget by adjusting the number of switches $n_{\text{switches}}$ while keeping the number of states $|\mathcal{S}| = 50$, actions $|\mathcal{A}| = 4$, and the time horizon $T = 50 \times 10^3$ constant. The number of switches is varied across $10, 45, 100$, and $1000$, with both the reward function and the transition dynamics being non-stationary. The observed variation in rewards $\Delta_{R,T}$ is $9, 48, 98$, and $1000$, respectively, and the observed variation in transitions $\Delta_{P,T}$ is $4, 14, 30$, and $303$, respectively, corresponding to different levels of non-stationarity.

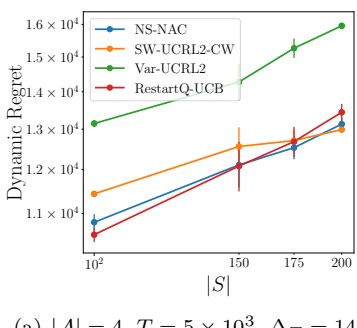

(a) $|\mathcal{A}| = 4$, $T = 5 \times 10^3$, $\Delta_T = 14$

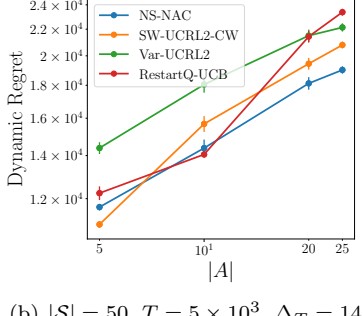

(b) $|\mathcal{S}| = 50$, $T = 5 \times 10^3$, $\Delta_T = 14$

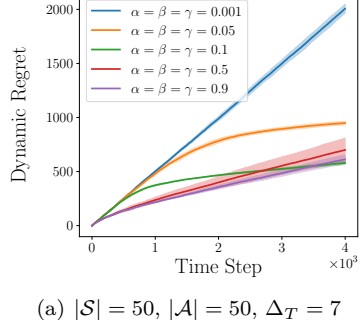

(a) $|\mathcal{S}| = 50$, $|\mathcal{A}| = 50$, $\Delta_T = 7$

Figure 2: Log-log plots showing the effect of varying: (a) number of states $|\mathcal{S}|$, and (b) number of actions $|\mathcal{A}|$.

Figure 3:   Performance of NS-NAC with different step-sizes in an environment with 17 abrupt, randomly scheduled switches over $T = 4 \times 10^3$ steps.

**Varying $|\mathcal{S}|$.**   We study the effect of varying the number of states while keeping the time horizon $T$, number of actions, and variation budget $\Delta_T$ constant. Specifically, the time horizon $T$ is fixed at $50 \times 10^3$ steps, and

the number of states is varied across the values $100, 150, 175$, and $200$, corresponding to environments with different state sizes while keeping the number of actions fixed at 4. The $n_{\text{switches}}$ is adjusted to $75, 100, 120$, and $150$, respectively, in order to maintain a consistent $\Delta_{P,T}$ of around 14 for all environments. The reward function is kept stationary with $\Delta_{R,T} = 0$ (no switching between $\mathbf{r}_1$ and $\mathbf{r}_2$).

**Varying $|\mathcal{A}|$.** We examine the effect of varying the number of actions while keeping the time horizon $T$, number of states, and variation budget $\Delta_T$ constant. Specifically, the time horizon $T$ is fixed at $50 \times 10^3$ steps, and the number of actions is varied across the values $5, 10, 20$, and $25$, corresponding to environments with different action sizes while keeping the number of states fixed at 50. The $n_{\text{switches}}$ is kept constant at 45 across all experiments to maintain a consistent variation budget $\Delta_{P,T}$ of around 14 for all environments. The reward function is kept stationary with $\Delta_{R,T} = 0$ (no switching between $\mathbf{r}_1$ and $\mathbf{r}_2$).

**Parameters.** The true variation budgets, $\Delta_{P,T}$ and $\Delta_{R,T}$, are provided to each algorithm, while the remaining hyperparameters are configured according to the optimal expressions derived in their respective papers. For SW-UCRL2-CW, the parameters include the window size $W_*$ and the confidence widening parameter $\eta_*$, both set using the optimal expressions given in the paper, and the confidence parameter $\delta = 0.05$. For Var-UCRL2, the true values of the variation budgets for transitions probabilities $\Delta_{P,T}$ and rewards $\Delta_{R,T}$, along with the confidence parameter $\delta = 0.05$, are used. In RestartQ-UCB, the ending times of the stages $L$, confidence parameter $\delta = 0.05$, initial number of samples $N_0$, and number of epochs $D$ are configured as described in the original paper with $H = 1$ (to adapt from episodic setting for which the algorithm is designed to infinite horizon setting in our work). For NS-NAC, we tune the step-sizes and number of restarts by grid search. The effect of different choices of step-sizes can be observed in Figure 3. Further, for BORL-NS-NAC, we set the number of epochs as $W = \lfloor T^{2/3} \rfloor$.

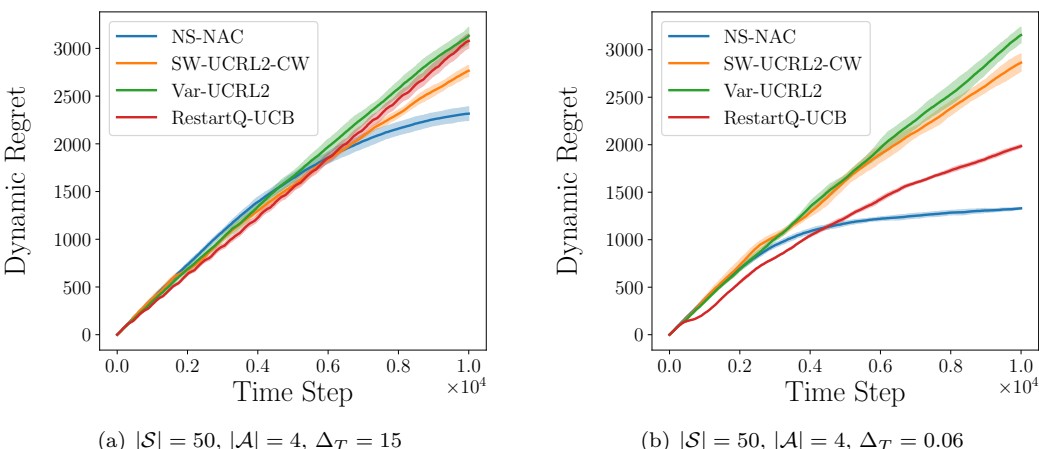

(a) $|\mathcal{S}| = 50$, $|\mathcal{A}| = 4$, $\Delta_T = 15$       (b) $|\mathcal{S}| = 50$, $|\mathcal{A}| = 4$, $\Delta_T = 0.06$

Figure 4: Performance of NS-NAC and baseline algorithms in various non-stationary settings. (a) Dynamic regret for a single instance over $T = 1 \times 10^4$ steps in an environment with 50 abrupt, randomly scheduled switches. (b) Dynamic regret for a single instance over $T = 1 \times 10^4$ steps in an environment with small, continuous changes.

**Additional Environments.**

We conducted further experiments to evaluate the adaptability of NS-NAC and baseline algorithms across diverse non-stationary settings. Figure 4(a) illustrates performance in an environment with 50 abrupt and randomly scheduled switches (between $\mathbf{P}_1$ and $\mathbf{P}_2$), simulating scenarios with non-periodic unpredictability. Figure 4(b) captures performance in a continuously changing environment, where the transition from $\mathbf{P}_1$ to $\mathbf{P}_2$ occurred gradually over $T = 10^5$ steps resulting $\Delta_T = 0.06$. This scenario reflects real-world conditions where systems experience smooth drift rather than abrupt changes. The results highlight NS-NAC's effectiveness in handling both abrupt and gradual changes, consistently matching the performance of baseline methods.

