# OpenReview forum: "Natural Policy Gradient for Average Reward Non-Stationary Reinforcement Learning"
_TMLR — Accepted by TMLR_

### Review · Reviewer_5B6M · 2025-10-23

**Summary Of Contributions:**

**Summary**

This work lies in the field of non-stationary (NS) reinforcement learning (RL), considering the infinite-horizon average-reward setting.
The authors claim to propose the first policy-based method for this setting, namely NS-NAC (non-stationary natural actor critic), which is a policy gradient method incorporating restart-based exploration for change. The latter means that the method divides the time horizon $T$ into $N$ segments of equal length at the beginning of which each NAC sub-routine is re-initialized. The authors show that selecting the step sizes as a function of the variation budget $\Delta_{T}$ and $N=\Delta_{T}^{5/6} T^{1/6}$, then NS-NAC suffers a dynamic regret of order $\widetilde{\mathcal{O}}(|\mathcal{S}|^{1/2} |\mathcal{A}|^{1/2} \Delta_{T}^{1/6} T^{5/6})$.  While the results are presented for tabular softmax policies, they are extended to the case of function approximation in appendix.
Furthermore, the authors introduce a bandit over RL (BORL) variant of NS-NAC, namely BORL-NS-NAC, which deletes the need for passing $\Delta_{T}$ as an input (required in NS-NAC to set the step sizes). Fundamentally, this variant runs the EXP3.P method over NS-NAC subroutines whose parameters are pulled by the overlying bandit method. The authors show that such a parameter-free variant suffers the same regret of the original version when the number of epochs is selected as $\mathcal{O}(T^{2/3})$.
The authors discuss their theoretical result against a lower bound for non-stationary undiscounted RL, conjecturing that the cause of the regret bound not to be tight is likely due to the analysis.
Finally, the proposed methods are validated in synthetic tabular settings in which they are compared against baselines, showing a sub-linear regret that aligns with the one exhibited by the competitors.

**Strengths**
1. The faced problem is relevant and, up to my knowledge, the proposed method(s) actually seems to be the first model-free policy-based method(s) for NS-RL. This opens to more practical usage of RL methods for NS environments, since policy-based methods come with notorious benefits for real-world-like problems.
2. In Section 3 the authors introduce all the necessary knowledge to understand the content in the main paper, making the manuscript almost self-contained.
3. Even if the authors state the main results for tabular environments, they provide an extension to function approximation
4. The authors take the time to present their results and they extensively compare the guarantees of their method with the ones of other baselines (even if thought for different settings) and with the lower bound for the undiscounted setting.

**Weaknesses**
1. I feel that the BORL-NS-NAC method is not introduced in the best possible way. The method, which is meant to remove from NS-NAC the need of passing $\Delta_{T}$ as input is not introduced, seems just an add-on in the abstract as well as in the original contribution. Moreover, when the reader gets to Section 6, it is very brief. Since I think that BORL-NS-NAC is an important extension of the main method (as also done in [12]), even because it presents the same guarantees, I suggest to introduce it properly, right from the beginning of the paper (abstract, original contribution, when presenting NS-NAC). To this end, the authors could stress deeply the importance of requiring $\Delta_{T}$ as an input parameter, report the pseudo-code of the method in the main paper, comment on how the step sizes of the underlying subroutines are selected, comment why and how the guarantees are the same (indeed there is no commentary after Theorem 6.1). I suggest to bring back in the main paper many information that are deferred to Appendix F.
2. Algorithm 1 does not show how to use $\Delta_{T}$. I suggest to briefly specify how it is employed in the step-sizes. Moreover, it seems that NS-NAC does not need actually the knowledge of the budget to work and to show guarantees on the dynamic regret, but just to select the optimal step sizes to obtain the final bound for the regret.
3. NS-NAC is based on fully restarting, meaning that after every amount of interaction, all the information collected are trashed, as commonly done in this setting [27]. However, do the authors believe that other approaches to forget the past (as sliding windows) may be well suited for this setting and possibly improving the results?
4. In Section 5.2, when introducing the result on the universal regret, I feel that this part is not very self-contained. I think that the paper would benefit from more commentary on this part (from Appendix D.7), especially regarding the assumptions needed and their meaning, since at the moment the authors just briefly mention the condition number and recurrent coefficient in the statement of Corollary 5.4.
5. The reported lower bound for the NS undiscounted setting from [16] lacks of the term $D^{2/3}$, being $D$ the maximum diameter of the MDPs. Is it just a typo, or is not important in this context (even if I guess it is)? Moreover, why the regret bounds for the method are not depending on the MDPs diameter?
6. The regret bound are not tight w.r.t. the lower bound and the authors comment this by conjecturing (Section 5.2) that there is a slack in the analysis of the underlying NAC which is adapted from the stationary setting. Do the authors conjecture that a different analysis could lead to a tighter bound? Or is it a problem inside the method itself? What about starting from simpler on-policy policy gradient methods?
7. I suggest to move Section 7 right after the presentation of the regret bound of NS-NAC.
8. In practice, the proposed methods quite align with baselines, not showing a huge practical advantage, except for NS-NAC in the experiments in the appendix (Figure 4). Moreover, the validation section in the main paper does not describe the settings in which the methods are tested. Furthermore, BORL-NS-NAC is not considered in the appendix. Even if I think that the amount of experiments is enough for a theoretical paper, I would have appreciated a sensitivity study of NS-NAC to $\Delta_{T}$ when compared to BORL-NS-NAC.

**References**
The same of the paper under revision.

**Additional Comments:**

**Minors**
 1. In every statements (in the main paper and in the appendix) the word "Theorem" is used instead of the word "Assumption".
 2. Lemma 5.2 does not seem to be the Lemma 2 of [47], but it is exactly Assumption 4.1 of [33]. If it is the case, consider to change it as an assumption too, otherwise, I kindly request to shed light on this fact.
 3. Typo at the top of page 21: "[...] and and [...]".

**References**
The same of the paper under revision.

**Audience:**

Yes

**Audience Explanation:**

The paper seems to be the first to propose model-free policy-based method(s) for NS-RL, a complex setting that matches many real-world scenarios. This works opens to more practical usage of RL methods for NS environments, since policy-based methods come with notorious benefits for real-world-like problems. Moreover, I think that this paper leaves some room for future work for policy-based methods in NS-RL.

**Broader Impact Concerns:**

I have no concerns on ethical implications of the work.

**Claims And Evidence:**

Yes

**Claims Explanation:**

For each theoretical result, the authors report the full proof if the result is new or a precise reference if the result is reported.
The method is also validated in synthetic settings, verifying that the methods show sublinear regret, aligning with the baselines.

**Requested Changes:**

**Critical Changes**
1. I suggest to elevate and integrate the BORL-NS-NAC Variant. Please see Weakness 1.
2. I suggest to clarify the role of $\Delta_T$ in Algorithm 1 in the case in which it has to be explicited. Please, see Weakness 2.
3. I suggest to clarify the lower bound comparison, especially regarding the missing term for the MDP diameter. Please see Weakness 5
4. I suggest to make the part of Section 5.2 on universal regret more self-contained (see Weakness 4). Alternatively, it can just be briefly mentioned and deferred to appendix.
5. I suggest to add more commentary for the experimental validation reported in the main paper, at least with a description of the environments employed (see Weakness 8).

**Recommended Changes**
1. Discuss a little bit more on restarting mechanism and compare it against other method for facing non-stationarity (see Weakness 3).
2. If it is the case, discuss whether tighter bound can be achieved by analyzing a simpler on-policy policy gradient method first (also as a conjecture, leaving it for future work), to isolate the slack introduced by the NAC analysis (see Weakness 6).
3. Please consider moving Section 7 to appear right after presenting the regret bound (after Section 5, see Weakness 7).
4. Please consider to add BORL-NS-NAC to the experiments in the appendix for a more complete validation, as well as considering to do a sensitivity analysis centered on the $\Delta_{T}$ parameter for NS-NAC, and compare it with BORL-NS-NAC (see Weakness 8).

---

> ### Author Response · Authors · 2025-12-05
>
> - We would like to thank the reviewer for appreciating the significance of our work in being the first policy-based algorithm for non-stationary RL, our results in both tabular and function approximation settings and the empirical evaluation we present.
>
> &nbsp;
>
> - Thank you for the suggestion on BORL-NS-NAC. We now have an expanded section in the main paper describing the algorithm in detail and a deeper integration across the paper.
> - **Role of Variation Budget in Algorithm 1:** We clarify the use of variation budget in Algorithm 1 and Section 4 in the revised version of the paper. The step-sizes corresponding to the actor $\beta$, critic $\alpha$ and the average reward $\gamma$ resulting in the optimal regret are a function of the variation budgets $\Delta_{R,T}, \Delta_{P, T}$. While NS-NAC does not need the knowledge of the budget to run in practice, poorly chosen step-sizes will lead to linear regret.
> - **Other Forgetting Mechanisms:** In our algorithm, the step-sizes act as forgetting factors or soft sliding windows. We observed that we needed an additional mechanism to explore for change (restarts in our case) and step-sizes as the soft sliding windows were not enough. Further, note that (hard) sliding windows as forgetting mechanism have been used in model based algorithms. While implementing such a mechanism alongside UCRL style algorithms require only remembering the observations i.e state-action-reward pairs in the trajectory, implementing a (hard) sliding window in model-free algorithms requires holding past gradients of actor and critic in memory which could become very expensive for high-dimensional parameterizations of policy and value function. We consider developing further efficient and practical forgetting mechanisms as future work.
> - We move the section on Universal Dynamic Regret entirely into Appendix D.7 for ease and clarity of presentation as suggested.
> - **Diameter and Mixing Time:** While the Diameter describes the complexity of the Markov chain in model-based and value-based algorithms, its counter-part that occurs naturally in policy gradient algorithms is the Mixing Time of the Markov chain. The mixing time can be defined similar to [4] as $\tau := \min \\{i \geq 0 | m\rho^{i-1} \leq \min\\{\alpha, \beta\\} \\}$ where $m, \rho$ are defined in Assumption 1. Further note that, $\lambda$ as defined in Lemma 1 also contains information on the mixing time of the Markov chain but it does not solely describe the mixing and contains additional information of the policy in use. Now, the regret upper bound in Theorem 1 with the explicit dependence on mixing time $\tau$ and $\lambda$ can be stated as $$\hbox{Dyn-Reg}(\mathcal{M}, T) \leq \mathcal{\tilde{O}}\left( |\mathcal{S}|^{1/2}|\mathcal{A}|^{1/2} \frac{\tau_{mix}}{\lambda} \Delta_T^{1/6} T^{5/6} \right).$$ Note that these quantities, $\tau$ and $\lambda$, are tracked explicitly in Appendix D. We choose not to include this explicit dependence in Theorem 1 in the main paper to maintain focus on the dependence of regret on the variation budget and prevent any distractions or confusion by comparing to the Diameter ($D$) in the literature.
> - **Gap between Bounds:** The slack and the gap between the upper and lower bounds arises due to the critic analyzing the norm-squared error $\Vert Q_t - Q_t^{\pi_t} \Vert^2$ (Proposition 2) while the actor needing the critic estimation error as $\Vert Q_t - Q_t^{\pi_t} \Vert$ (Proposition 1). The Cauchy-Schwarz inequality to relate the two causes the gap. The critic analysis uses squared-norm error instead of norm error because Markovian noise terms are more tractable to analyze in squared form using martingales. We conjecture that a different analysis technique that bridges this mismatch will close the gap between the upper and lower bounds. We note that this establishing this bridge continues to be an open problem to the best of our knowledge and all currently existing expected value estimation analyses in the actor-critic literature [1,2,3,4] provide guarantees on the norm-squared error by leveraging an appropriate decomposition and the auxiliary chain method. Further, moving to a simpler on-policy policy gradient algorithm will enable us to characterize only the convergence to a stationary point and not the gap between the average rewards induced by current and the optimal policies as required by our dynamic regret performance metric.

---

> > ### Author Response · Authors · 2025-12-05
> >
> > - **Simulations:**
> >   - We would like to draw to the reviewer's attention that the baselines we compare against in Section 8 and Appendix G namely SW-UCRL2-CW, Var-UCRL2 and RestartQ-UCB are model-based and value-based algorithm unlike our algorithms which are policy based methods. In our experiments, we show that both our algorithms NS-NAC and BORL-NS-NAC incur sub-linear regret and match the performance of the model-based and value-based baselines considered. It is out of scope of this work to compare or demonstrate that policy-based methods are better/worse than model-based and value-based methods.
> >   - We choose to skip presenting the details of the experimental setup in the main paper to ensure brevity and to maintain the reader's focus on the theoretical results which are the main contributions of this work.
> >   - The sensitivity of both NS-NAC and BORL-NS-NAC to changing variation budget $\Delta_T$ are presented in Figure 1 (c) in Section 8 of the main paper.
> >
> > &nbsp;
> >
> > **References:** \
> > [1] Wang, Y., Wang, Y., Zhou, Y., and Zou, S. Non-asymptotic analysis for single-loop (Natural) actor-critic with compatible function approximation. ICML 2024. \
> > [2] Chen, X. and Zhao, L. Finite-time analysis of single timescale actor-critic. NeurIPS 2023. \
> > [3] Khodadadian, S., Doan, T. T., Romberg, J., and Maguluri, S. T. Finite-sample analysis of two-time-scale natural actor–critic algorithm. IEEE Transactions on Automatic Control 2022. \
> > [4] Wu, Y. F., Zhang, W., Xu, P., and Gu, Q. A finite-time analysis of two time-scale actor-critic methods. NeurIPS 2020.

---

> > > ### Comment · Reviewer_5B6M · 2025-12-18
> > >
> > > I would like to thank the author for having answered my concerns and for the submitted revision of the manuscript. Here are my additional comments:
> > > 1.  **BORL-NS-NAC.** The additional commentary on BORL-NS-NAC has strengthened the paper. Moreover, I have appreciated that the authors highlighted the additional cost deriving from running EXP3.P.
> > > 2. **Complexity Term.** I thank the authors for the discussion on the mixing time. I think that it would be valuable to add it in the manuscript, as well as I believe that the lower bound result from [16] should be reported entirely, then commenting as done in the rebuttal.
> > > 3. **Lemma 1.** As written also in my preliminary review, I believe that is not that straightforward to translate Lemma 1 into Lemma 2 of [46] without some comments. Moreover, it seems to be very near to Assumption 4.1 of [33] (of course without considering to be under the uniform ergodicity).
> > > 4. **Simulations.** I apologize for having missed the sensitivity study. Yet, I believe that some more commentary (as done in Appendix G) on the simulations in the main paper will enrich the discussion.

---

> > > > ### Author Response · Authors · 2025-12-19
> > > >
> > > > Thank you for the discussion!
> > > >
> > > > - We add the discussion on the explicit dependence on mixing time to the revised version of the paper.
> > > > - While negative semi-definitiveness is an assumption (Assumption 4.1 of [33]) when there is function approximation involved, in the tabular case it no longer is an assumption but instead can be proved. Lemma 1 in our work follows from Eq 3.5, Lemma 2, [46] because in Eq 3.5, $\Phi = I$ in the tabular case and plugging in $\lambda=0$ for TD(0) results in an equivalence between $D(I-P) = -\bar{A}$.
> > > > - We now have an expanded section on the simulations.

---

### Review · Reviewer_PnUT · 2025-10-30

**Summary Of Contributions:**

The authors present 2 model-free policy-based algorithms for non stationary MDPs under the average reward setting.
- NS-NAC: a natural actor-critic algorithm that uses periodic restarts to handle the non stationary changes. However it requires to know the variation budgets.
- BORL-NS-NAC: a relaxed version of the first algorithm that uses a bandit-over-RL framework to eliminate the need for prior knowledge of the environment's total variation budget.
- Theoretical Analysis: a dynamic regret bound for both algorithms. The proof introduces a novel adaptation of Lyapunov analysis to a moving-target optimal policy.

Strengths:
- the paper fills a gap in the literature by providing the first theoretical analysis for a policy-based method in this setting
- the inclusion of a relaxation (BORL-NS-NAC) enhances the practical relevance of the work.

Weakness:
- it looks that experimentally BORL-NS-NAC is not better than the baselines
- there is a gap

**Additional Comments:**

Policy gradient is usually more interesting than value based methods in continuous action spaces, can your analysis be easily extended to this setup?

**Audience:**

Yes

**Audience Explanation:**

The work addresses a fundamental problem in reinforcement learning: non-stationary MDP, which is of broad interest. It provides the first theoretical justification for using policy gradient methods, which are widely used in practice. Researchers in RL theory, continual learning, and policy optimization could find the results valuable.

**Claims And Evidence:**

Yes

**Claims Explanation:**

The paper's central claims are theoretical, and they are supported by a detailed proof. The analysis adapts standard, sound techniques to the novel problem setting. The empirical results on a synthetic environment, while not extensive, corroborate the theoretical findings by demonstrating sub-linear regret.

**Requested Changes:**

In algorithm 1 and section 4, I can't find where ∆R,T and ∆P,T are needed. I do not think it is mentioned. Please make this clearer.

---

> ### Author Response · Authors · 2025-12-05
>
> - We would like to thank the reviewer for the constructive feedback and appreciating the gap in literature that this work fills by providing the first theoretical analysis of policy-based algorithms in the non-stationary settings and the practical relevance of BORL-NS-NAC.
>
> &nbsp;
>
> - **Empirical Performance of BORL-NS-NAC:** We would like to draw to the reviewer's attention that the baselines we compare against in Section 8 and Appendix G namely SW-UCRL2-CW, Var-UCRL2 and RestartQ-UCB are model-based and value-based algorithm unlike our algorithms which are policy based methods. In our experiments, we show that both our algorithms NS-NAC and BORL-NS-NAC incur sub-linear regret and match the performance of the model-based and value-based baselines considered. It is out of scope of this work to compare or demonstrate that policy-based methods are better/worse than model-based and value-based methods.
> - **Role of Variation Budget in Algorithm 1:** We clarify the use of variation budget in Algorithm 1 and Section 4 in the revised version of the paper. The step-sizes corresponding to the actor $\beta$, critic $\alpha$ and the average reward $\gamma$ resulting in the optimal regret are a function of the variation budgets $\Delta_{R,T}, \Delta_{P, T}$.
> - **Extension to Continuous Action Spaces:** It is indeed interesting to study policy gradient algorithm with varying dynamics in the continuous action space setting and we reserve this as future work that we would like to build upon our current work.

---

### Review · Reviewer_yZXp · 2025-11-28

**Summary Of Contributions:**

This paper studies non-stationary reinforcement learning in the infinite-horizon average-reward setting. In the considered non-stationary RL setting, the reward and transition functions may vary at any time step but subject to a $T$-step variation budget. Notably, the total variation budget, denoted by $\Delta_T$, is of primary concern. The performance is measured via the notion of dynamic regret, which is relevant for the considered non-stationary setting. Such a regret definition is indeed a form of variational regret. The main contribution of the paper is an algorithm called NS-NAC, abbreviating Non-Stationary Natural Actor-Critic, which is a two-time scale natural policy-gradient method. NS-NAC achieves a dynamic regret scaling (in expectation) as $|\mathcal S|^{1/2}|\mathcal A|^{1/2} \Delta_T^{1/6} T^{5/6}$, ignoring log-terms. In fact, the said regret bound for NS-NAC could be achieved only if the algorithm would know $\Delta_T$ (or an upper bound thereof) to set its step-sizes and the number of restarts. Besides, NS-NAC relies on the knowledge of a projection radius $R_Q$.
To relax the requirement of knowing $\Delta_T$, the authors introduce BORL-NS-NAC, an *almost* parameter-free variant of NS-NAC that employs a bandit-over-RL approach ---BORL-NS-NAC still needs to know the projection radius $R_Q$. BORL-NS-NAC is shown to achieve the same (in order) regret bound as NS-NAC.


**Strengths**

- The paper studies an important and challenging problem in the domain of regret minimization in MDPs, which is both theoretically and practically relevant. The two key components of the model (average-reward criterion and non-stationary subject to a variation budget) impose much technical challenge. In contrast to existing work that are mostly model-based, the work presents the first policy-based methods, which makes a nice addition to the literature.

- The regret analysis of NS-NAC appears to borrow its key technical elements from the analyses PG-style methods for (stationary) average-reward RL. Nevertheless, the analysis has its own complexity due to restart-based exploration and non-trivial learning rates as adapting factors. All these require requires some novel adaptations and tailoring in the analysis.

- Another key strength of the paper is to introduce and analyze its bandit-over-RL variant (BORL-NS-NAC), which removes the need to know $\Delta_T$ or a related upper bound. However, the algorithm is not fully parameter free due its dependence on $R_Q$. Yet another strength (in my view) is reporting an empirical evaluation and comparison to existing algorithms.

- The paper is written very well, admits a clear organization and presentation, and is overall a nice read. It offers a rich literature review, carefully positioning its contributions to the relevant work and citing most relevant papers I am aware of. Besides a few sentences that need some revision/polishing, there were only a few typos/grammar mistakes (listed later) that could be easily fixed.


**Weaknesses and Concerns**

- As a major concern, the regret analysis for NS-NAC is done under the very restrictive assumption that all policies induce uniformly ergodic Markov chains. Unless I am missing something, this essentially amounts to restricting to ergodic MDPs, which exclude many relevant domains in RL which are (weakly) communicating MDPs. If correct, this key fact must have been clearly stated earlier (e.g., Introduction) and must have been taken into account when positioning the algorithm relative to its model-based competitors that apply to wider classes (e.g., communicating).

- Related to the previous concern, all the presented algorithms here crucially rely on the knowledge of $R_Q$, which itself depends on $\lambda$, which might be related to mixing-type quantities. It's never discussed how crucial this dependence (i.e., knowing $R_Q$) is and how this can be interpreted.

- Regarding NS-NAC: in line 5, why one would sample $s_t$ uniformly? Shouldn’t it be decided by the environment?

- Introducing and analyzing the BORL variant of NS-NAC is nice. However, that it achieves the same regret bound --as the paper claims-- raises a key question as to whether the algorithm is parameter-free for free? Of course, the cost to pay could be multiplicative polylog-terms (in $T$) or numerical constants or even lower order terms. But this is neither discussed nor possible to understand from its regret analysis (see also the next comment).

- The regret analysis of BORL-NS-NAC appears to be rushed: a final *clean* bound in missing in the proof (in p. 48). As mentioned above, a clean, explicit final regret bound (i.e., without hiding stuff in big-O notations) is especially crucial here to get insight into what is paid relative to the regret bound of NS-NAC.

- Another concern (mostly presentational) is that the requirement of knowing $R_Q$ in BORL-NS-NAC must be made explicit in the introduction (and other relevant places).

**Additional Comments:**

**Some Detailed Comments and Questions**

- Is your reward function deterministic? Please clarify.
- As far as I see, the paper (41) is not related to NPG. Could you please elaborate further?
- In Section 2, first paragraph: Note that the ref (7) studies the discounted setting. It does not render correct to include it in a discussion about average-reward setting.
- I suggest citing some key papers for the assumptions used, especially for the structural results such as Equation (3).
- Please provide a ref for the first few lines of Section 5.1, since they appear to come from the stationary PG analysis literature.
- In Algorithm 1: The explanation for actor policy and $\theta_{t+1}$ do not appear to be connected. Please revise accordingly.
- In Corollary 5.4: What is condition number? What is recurrent coefficient?
- When deriving the bound on Markovian noise: Isn’t correct that we could restrict to having $s_{t-\tau-1}=\tilde s_{t-\tau-1}$? How such an adaptation is technically different from the stationary case?



**Potential Typos/Grammar Mistakes, Notation Issues, and Unclear Sentences**

- In Section 2, when introducing Advantage Actor-Critic, introduce the abbreviation A2C that is used later.
- Use of this style (number) for citing a paper is misleading as one may confuse it easily with equation numbers. Why not [number]? Or (author, year)?
- The abbreviation RL is readily introduced in the introduction, but then the authors switch to “Reinforcement Learning” (e.g., in Section 3.2, first line).
- In the definition of $J^\pi$, an expectation is missing.
- What is “Theorem 5.1”. Did you mean “Assumption 5.1”? It is used in a number of places.
- p. 4: The model of … and notion of … is standard  ==> … are standard
- p. 4: In Bellman equation for $Q$, there is some unnecessary space before $=$.
- p. 5: sub optimal ==> sub-optimal
- p. 8: i.e ==> i.e.
- p. 5: Both $\boldsymbol{\pi}_{\boldsymbol \theta_t}$ and $\boldsymbol{\pi}_t$ are used to denote the same thing.
- p. 8: … challenges that non-stationarity. ==> … challenges that non-stationarity poses OR … challenges due to non-stationarity. [Otherwise, the sentence appears incomplete.]
- p. 9: time step ==> timestep [To ensure consistency with earlier uses.]
- p. 9: $\widetilde O_t$ is not defined.
- p. 1: UCRL ==> UCRL2 (assuming that you are referring to the (meta-)algorithm of Jaksch et al., 2010)
- The paragraph “Function Approximation” in p. 7 seems to be redundant as it is already discussed in p. 6 (before beginning of Section 5).

**Audience:**

Yes

**Audience Explanation:**

The paper studies an important and interesting problem in the area of theoretical RL. It will attract TMLR readers because:
- the considered setting (non-stationary RL in the average-reward setting) is technically challenging and interesting;
- the considered performance criterion (regret) is relevant to a wide-range of audience;
- the developed algorithmic tools and analyses are interesting.

**Broader Impact Concerns:**

The paper studies a theoretical problem, and its presented empirical evaluation is based on synthetic data. Therefore, I believe there is no concern associated to the developments and results reported in the paper.

**Claims And Evidence:**

Yes

**Claims Explanation:**

The key statements are regret bounds as performance certificate of the presented algorithms, which are supported by the proofs that appear overall correct to me, even though I did not carefully check math details.

In terms of presentation, I raised some concerns regarding:
- How the setting is advertised: the assumption of ergodicity is key and must have been emphasized earlier in the paper, especially when positioning the paper relative to the model-based competitors.
- How the BORL-NS-NAC is advertised: it still depends on some input parameter, while the text induces this feeling that it is fully parameter-free.
- Need to have a final, explicit regret bound for BORL-NS-NAC in its proof, which allows for comparison between NS-NAC and BORL-NS-NAC.

**Requested Changes:**

Please address the weaknesses mentioned above as well as my detailed comments under "Additional Comments".

The parameter-free extension of NS-NAC is indeed an important development from the standpoint of the method applicability, and I therefore think that it must be included in the main text, especially considering that there is some space for it.

---

> ### Author Response · Authors · 2025-12-05
>
> - We would like to thank the reviewer for appreciating the importance of problem considered, the challenge in the technical analysis of NS-NAC, and the practical relevance of BORL-NS-NAC. We also thank the reviewer for the detailed feedback which we address as follows.
>
> &nbsp;
>
> - **Assumption 1, Uniform Ergodicity:**
>   - The assumption of uniform ergodicity is necessary and regularly used in the policy gradient literature in stationary environments [Murthy \& Srikant, 2023; Wu et al., 2020; Zou et al., 2019]. Note that in these works, the Markov chain is assumed to be uniformly ergodic for any policy $\pi$ that is encountered in the learning process at any time-step $t \in [T]$.
>   - Further, the notion of uniform ergodicity for any policy $\pi$ in any environment $\mathcal{M}$ has also been standardly considered in the online MDP literature with policy-based algorithms with changing reward functions [1, 2, 3]. This assumption is critical in our non-stationary actor-critic algorithm analysis to enable critic estimation which requires characterizing state-action distribution (see the term on Markovian noise, $I_6$ in Eq. 7).
>   - While we consider Assumption 1 in our work, we acknowledge that practical MDPs might not always be ergodic (finite Markov chains are always uniformly ergodic/fast mixing). In the absence of ergodicity, in the stationary RL problem, [4] proposes a modification to the MDP in Assumption 1/Section 3 and establishes guarantees as a function of mixing time with only a small loss in performance in Corollary 4.5/Appendix 1. The modification ($\epsilon$-exploration, Schweitzer transformation) to the original MDP converts it to an ergodic MDP and thereby allows for proof methods based on ergodicity. We conjecture that this method can be extended to the non-stationary setting and consider a formal analysis of relaxing the ergodicity assumption as future work.
> - **Projection Radius:** The projection step in the critic update is a standard step in policy gradient and the broader stochastic approximation literature [5,6,7] and are a necessary tool for tractability in analysis of the critic and noise arising from their stochastic updates. While we consider setting $R_Q = 2U_R \lambda^{-1}$ similar to [6, 7], note that $2U_R \lambda^{-1}$ is merely a sufficient upper bound. Further, in practice, the projection radius i.e. the radius of the ball in which the TD limit point lies can be estimated as described in Section 8.2 of [5] and has no crucial dependence on mixing related quantities.
> - **Starting State Distribution:** We apologize for the typo in Line 5, Algorithm. We meant $s_{t=0} \sim \text{Unif}\{|\mathcal{S}|\}$ which states that the MDP at time $t=0$ starts from a state chosen uniformly at random. While the starting state in an average reward, uniformly ergodic MDP does not really matter, to prevent any confusion we edit it to $s_0 \sim \text{any starting distribution}$.
> - **Detailed Section on BORL-NS-NAC:** Thank you for the suggestion on BORL-NS-NAC. We now have an expanded section in the main paper describing the algorithm in detail and a discussion comparing regret of NS-NAC and BORL-NS-NAC. Theorem 3 in the revised version now includes all the explicit terms of the regret upper bound of BORL-NS-NAC. We would like to note that while BORL-NS-NAC suffers an additional cost of hedging in comparison to NS-NAC, it continues to remain order-wise the same.
>
> &nbsp;
>
> - Yes, the rewards we consider in our work are a deterministic function of the state and action and this is a standard notion in theoretical RL literature.
> - Apologies for the typo, paper [41] is indeed not on NPG. We now remove a reference to reference [7] to keep our focus on the average-reward setting. Added citations for Eq 3 / Sec 5.1. Added references for the connection between the $\theta_{t}$ and $\pi_t$ equations of the actor update equation.
> - Corollary on Universal Dynamic Regret: We move the entire discussion on Universal Dynamic Regret to Appendix D.7 for ease and clarity of presentation as suggested by reviewer 5B6M.
> - The auxiliary Markov chain is constructed by starting from $s_{t-\tau}$ and conditioning on $s_{t-\tau}, \pi_{t-\tau-1}, P_{t-\tau}$ and rolled out by applying $\pi_{t-\tau-1}, P_{t-\tau}$. It is indeed true that $s_{t-\tau-1} = \tilde{s}\_{t-\tau-1}$ and $s_{t-\tau} = \tilde{s}\_{t-\tau}$ by our choice of construction. The technical adaptation of the auxiliary chain in our case is different from that of the stationary case because now the transition probabilities $P_t$ change with time as well in the original Markov chain in addition to the policy updates. These time varying transition probabilities add an extra layer of complexity to the martingale argument as can be seen in Lemmas 10, 11.
>
> &nbsp;
>
> - Thank you, we have fixed the typos and made notation clearer!

---

> > ### Author Response · Authors · 2025-12-05
> >
> > **References:** \
> > [1] P. Zhao, L-F. Li, Z-H. Zhou. Dynamic Regret of Online Markov Decision Processes. ICML 2022. \
> > [2] G. Neu, A. Gyorgy, C. Szepesvari, A. Antos. Online Markov Decision Processes under Bandit Feedback. NeurIPS 2010. \
> > [3] E. Even-Dar, S. M. Kakade, and Y. Mansour. Online Markov decision processes. Mathematics of Operations Research 2009. \
> > [4] Y. Murthy, M. Moharrami, R. Srikant. Performance Bounds for Policy-Based Average Reward Reinforcement Learning Algorithms. NeurIPS 2023. \
> > [5] Bhandari, Jalaj and Russo, Daniel and Singal, Raghav. A Finite Time Analysis of Temporal Difference Learning With Linear Function Approximation. COLT 2018. \
> > [6] Tengyu Xu and Shaofeng Zou and Yingbin Liang. Two Time-scale Off-Policy TD Learning: Non-asymptotic Analysis over Markovian Samples. NeurIPS 2019. \
> > [7] Wu, Y. F., Zhang, W., Xu, P., and Gu, Q. A finite-time analysis of two time-scale actor-critic methods. NeurIPS 2020.

---

> > > ### Comment · Reviewer_yZXp · 2025-12-13
> > >
> > > I would like to thank the authors for the responses and the revision, which address my major concerns. However, the following key questions/concerns remain:
> > >
> > > 1. Moving the description and discussions of BORL-NS-NAC to the main text improves the readability and quality of the paper substantially. However, it is not crystal clear yet, in my opinion, how the regret bound in Theorem 3 compares to that in Theorem 1. Does use of the BORL technique impact the multiplicative poly-log (in $T$ or other parameters) term in the bound or its cost is merely larger universal constants? In the current execution, the use of $\tilde {\mathcal O}$ that hide poly-log factors hinders one to uncover this.
> > >
> > > 2. I agree with the authors the ergodicity assumption is rather standard in the literature on PG. Nevertheless, one anticipate that this key assumption is mentioned in the introduction, when you list your contributions.
> > >
> > > 3. Note that the model-based approaches for the considered problem do not necessarily require the ergodicity assumptions you make. In my view, this important point is deserved to be discussed, albeit briefly.
> > >
> > > 4. Some typos/notations mistakes are not corrected despite your promise; e.g.:
> > > - In Section 3.1, in the definition of $J^\pi$, $\mathbb E$ is missing. It must be $\lim_{T\to\infty} \frac{\sum_{t=0}^{T-1}\mathbb E[r(s_t,a_t)]}{T} = ...$
> > > - [NEW] ... BORL-NS-NAC continues have ==> ... BORL-NS-NAC continues to have OR ... continues having

---

> > > > ### Author Response · Authors · 2025-12-14
> > > >
> > > > We would like to thank the reviewer for the discussion.
> > > >
> > > > - The use of BORL over NS-NAC does not impact the poly-log terms in the bound. It introduces the additional term $\tilde{\mathcal{O}}\left(W \sqrt{\ln T \cdot \frac{T}{W}}\right)$ and only an extra $O(1)$ constant factor in the other terms as tracked in the proof in Section F. We have now made this clearer in the revised manuscript.
> > > > - We now make the presentation clearer by including that our results hold under the standard assumption of uniform ergodicity in the Contributions part of the Introduction.
> > > > - In Section 3.1, in the definition of $J^\pi$, we don't need the expectation since the rewards are a deterministic function of state and action and are not stochastic. This is a standard notion in RL.
> > > > - Thank you, we have fixed the typo!

---

> > > > > ### Comment · Reviewer_yZXp · 2025-12-14
> > > > >
> > > > > Thanks for the revised version.
> > > > >
> > > > > - I am not sure if I can agree with you regarding dropping $\mathbb E$ in the definition of $J^\pi$. Even though the reward function is deterministic, $(s_t,a_t)$ are not. (i) $a_t$ is sampled from $\pi$ (a potentially randomized policy), and (ii) $s_t$ is generated by the transition kernel $\mathbf P$. Thus, you need $\mathbb E$ to integrate such randomnesses, as used in the literature with a similar setting (see, e.g., http://proceedings.mlr.press/v119/wei20c/wei20c.pdf).
> > > > >
> > > > > - Minor typo in p. 2: BORL-NS-NACthat ==> BORL-NS-NAC that

---

> > > > > > ### Author Response · Authors · 2025-12-17
> > > > > >
> > > > > > Thank you for the further discussion! Since we consider ergodicity in our work, the limit converges to the expected value under the stationary distribution. This is a common way of representation in the policy gradient literature such as Wu et al., 2020. Nevertheless, to prevent any confusion, we modify the definition of our average reward in the revised version to make it generally applicable in all cases.

---

### Decision · Action_Editor_j6nE · 2026-01-08

**Recommendation:** Accept as is

**Audience:**

Yes

**Audience Explanation:**

Yes. The paper addresses non-stationary reinforcement learning in the infinite-horizon average-reward setting, a problem of both theoretical and practical importance. Its focus on policy-based methods, combined with rigorous analysis, makes it particularly relevant to TMLR readers interested in reinforcement learning theory and applications in dynamic environments.

**Claims And Evidence:**

Yes

**Claims Explanation:**

Yes. The paper provides theoretical analysis of the proposed algorithms (NS-NAC and BORL-NS-NAC), with clearly stated assumptions and proofs that substantiate the main claims. The evidence is presented in a convincing manner, and revisions in response to reviewer feedback have further clarified algorithmic details and strengthened the exposition.